# Longitudinal plasma proteome profiling reveals the diversity of biomarkers for diagnosis and cetuximab therapy response of colorectal cancer

Yan Li [1,4], Bing Wang [1,4], Wentao Yang[2,3,4], Fahan Ma [1,4], Jianling Zou [2,3,4], Kai Li [1], Subei Tan [1], Jinwen Feng [1], Yunzhi Wang [1], Zhaoyu Qin [1], Zhiyu Chen [2,3] ✉ & Chen Ding [1] ✉

Cetuximab therapy is the major treatment for colorectal cancer (CRC), but drug resistance limits its effectiveness. Here, we perform longitudinal and deep proteomic profiling of 641 plasma samples originated from 147 CRC patients (CRCs) undergoing cetuximab therapy with multi-course treatment, and 90 healthy controls (HCs). COL12A1, THBS2, S100A8, and S100A9 are screened as potential proteins to distinguish CRCs from HCs both in plasma and tissue validation cohorts. We identify the potential biomarkers (RRAS2, MMP8, FBLN1, RPTOR, and IMPDH2) for the initial response prediction. In a longitudinal setting, we identify two clusters with distinct fluctuations and construct the model with high accuracy to predict the longitudinal response, further validated in the independent cohort. This study reveals the hetero-geneity of different biomarkers for tumor diagnosis, the initial and long-itudinal response prediction respectively in the first course and multi-course cetuximab treatment, may ultimately be useful in monitoring and intervention strategies for CRC.

Colorectal cancer (CRC) is the third most common cancer worldwide and the second most frequent cause of cancer deaths[1]. For CRC patients, screening has been proven to reduce cancer mortality in average-risk women and average-risk men[2]. Screening approaches includes noninvasive fecal occult blood tests, faecal immunochemical tests, carcinoembryonic antigen test, and colonoscopy[3–5]. However, the slow and asymptomatic nature of disease progression renders diagnosis at an early stage challenging, which severely reduce opportunities for timely disease detection and intervention[6]. Therefore, the pressing need is to explore the potential biomarkers to distinguish the CRC patients

from healthy controls, thus improving the noninvasive accurate diagnostic strategies.

Approximately half of patients with CRC die of metastatic disease; the overall five-year survival of these patients is less than 10%[7]. Current first-line standard of care for patients with unresectable CRC is che-motherapy in combination with monoclonal antibodies (mAbs), including anti–epidermal growth factor receptor (EGFR) mAbs or the anti–vascular endothelial growth factor (VEGF) mAb. After more than 20 years of translational and clinical investigation, EGFR family and its intracellular signaling pathways still represents the most relevant keystone for the targeted molecular treatment of CRC[8]. EGFR (HER1) is

[1]State Key Laboratory of Genetic Engineering and Collaborative Innovation Center for Genetics and Development, School of Life Sciences, Institutes of Biomedical Sciences, Human Phenome Institute, Zhongshan Hospital, Fudan University, Shanghai, China. [2]Department of Gastrointestinal Medical Oncology, Fudan University Shanghai Cancer Center, Shanghai, China. [3]Department of Oncology, Shanghai Medical College, Fudan University, Shanghai, China. [4]These authors contributed equally: Yan Li, Bing Wang, Wentao Yang, Fahan Ma, Jianling Zou. ✉e-mail: chanhj75@aliyun.com; chend@fudan.edu.cn

a growth factor receptor belonging to a family of cell membrane growth factor receptors with tyrosine kinase enzymatic activity, including HER2/c-neu (ERBB2), HER3 (ERBB3), and HER4 (ERBB4)[9]. The specific ligand binding to the EGFR extracellular domain induces receptor homodimerization or heterodimerization, which, in turn, triggers the phosphorylation of specific tyrosine residues in the EGFR intracellular domain. This activates a complex intracellular signaling cascade, which regulate cancer cell proliferation, survival, invasiveness, metastatic spread, and tumor-induced angiogenesis[10].

Cetuximab, a chimerized monoclonal antibody to EGFR, specifically bind to the EGFR with high affinity by occluding the ligand-binding region, blocking ligand-induced EGFR tyrosine kinase activation and subsequent signal-transduction events leading to cancer cell proliferation[9,11,12]. A randomized trial (CO.17) showed that among patients with CRC that had not responded to advanced chemotherapy, cetuximab improved overall survival and progression-free survival and preserved the quality of life better than did best supportive care alone[13]. Although EGFR is expressed in approximately 85% of patients with CRC, resistance to cetuximab was common that the disease had progressed in more than 50% of treated patients at the first assessment[13,14]. Therefore, it is necessary to find potential predictive biomarkers of the first treatment response to cetuximab. Activating mutations in *KRAS* are found in approximately 35–40% of CRC patients, which could result in EGFR-independent intracellular signal transduction activation, thus rendering EGFR inhibitors ineffective[10,15,16]. The presence of *KRAS*-activating mutations was discovered as the first predictive negative biomarker for anti-EGFR treatment in CRC[17]. A series of potential biomarkers that could be useful in predicting response to EGFR inhibitors has been investigated[18–20]. Because tumor heterogeneity is strongly associated with anti-EGFR drug resistance[21], a single biomarker is unlikely to satisfy the sensitivity and specificity required for most applications in predicting therapeutic response. In addition, acquired resistance inevitably emerges, disease progression occurs in all patients which limits the clinical efficacy of the drug[8,22].

The emergence of acquired resistance makes patients unable to benefit in the long term; however, the early warning of acquired resistance remains unclear. Therefore, the dynamic sampling at each therapy course of CRC patients is necessary, and contribute to the well-understanding of the dynamic trajectories related to therapy response during cetuximab treatment. Clinical analysis of tissue and blood is the most commonly used method to support treatment decisions for patients. Due to the invasiveness and risk for the patient, tissue biopsies could hardly monitor dynamic temporal and spatial changes that occur during CRC disease. The dynamic nature of the body fluid circulatory system and its constituent reflects diverse physiological or pathological states. Proteins circulating in the blood can be both mediators of organ cross talk and markers of whole-body states, and the ease of which in blood can be used for biomarker applications[23,24]. Mass spectrometry (MS)-based plasma proteomics has been successfully applied to disease diagnosis and characterization of protein change trajectories in disease, indicating that it is an optimal technology for biomarker discovery in this easily accessible plasma samples[25–29].

Here, we collected 540 plasma samples from 116 CRC patients (CRCs) undergoing cetuximab therapy with continuous multiple treatment courses and 66 healthy controls (HCs) in the plasma discovery cohort, 101 plasma samples from 31 CRCs and 24 HCs in the plasma validation cohort, as well as 31 tumor tissue samples and 27 paired normal-adjacent tissues (NATs) originated from 31 therapy-naïve CRCs in the tissue validation cohort in this study. We proposed an unbiased comprehensive analysis of plasma proteome to search for biomarker panels for CRC diagnosis and therapy response to cetuximab therapy. We firstly performed comparative proteomic analysis of proteome profiles between pre-treated CRC patients and healthy controls. The results showed that pre-treated CRC patients were featured by

extracellular matrix organization (ECM). We identified COL12A1, THBS2, S100A8, and S100A9 proteins, mainly involved in ECM pathway, could be used as effective biomarkers for distinguishing CRCs from HCs with high accuracy in our plasma discovery cohort, which was further validated in the independent tissue and plasma validation cohorts. In addition, we validated the predictive effect of the four proteins on a wider population scale of CRC patients, but not limited to the *RAS* wild-type CRC patients or metastatic CRC patients. We also explored the predictive efficacy of the four proteins (COL12A1, THBS2, S100A8, and S100A9) in multi-cancer cohort composed of 115 plasma samples from 95 patients with treatment-naive patients with various cancer types and 20 HCs, and two public cohorts related to the non-malignant diseases. The results showed four proteins could well distinguish CRCs from HCs, but could not distinguish the patients with other cancers and the non-malignant diseases (such as autoimmune diseases and infections), including gastric cancer (GC), esophageal cancer (EC), breast carcinoma (BRCA), lung cancer (LC), ulcerative colitis (UC), bladder cancer (BLCA), malignant lymphoma (ML), and SARS-CoV-2 infection, demonstrating the relative specificity of the four proteins for the CRC diagnosis. Then, we performed consensus clustering analysis on the proteome profiles of the pre-treated CRC patients, and identified three proteomic subtypes featured by distinct bioprocesses and associated with diverse therapeutic responses. We found that immune activation signaling and RRAS/RRAS2-mediated ECM pathways were associated with different responses to the initial cetuximab treatment. We proposed the key regulators involved in these pathways, including RRAS2, MMP8, FBLN1, RPTOR, and IMPDH2, could be used as the biomarkers for the initial response prediction of the first cetuximab treatment, in distinguishing sensitive patients from non-sensitive patients, which was also validated in the independent tissue and plasma validation cohorts. Furthermore, we analyzed a longitudinal cohort to identify the proteins with dynamic change associated with the treatment response to cetuximab. Finally, we used the logistic regression strategy to construct the predictive model composed of IDH3G, MDN1, KLC4, MYL9, SBF1, and HTRA3, which achieved high accuracy to predict response in multi-course cetuximab treatment. The robustness of this predictive model was further validated in the independent plasma cohorts. Overall, this study provides plasma protein dynamic changes throughout the cetuximab therapy course at an individual patient level, and identifies the potential biomarkers for tumor diagnosis, the initial response prediction and longitudinal response prediction, revealing the heterogeneity of these different biomarkers in the clinical management of CRC.

## Results

### The characteristics of CRC anti-EGFR therapy cohorts for plasma and tissue proteome profiling

To investigate the proteomic patterns of CRC and the association with response to cetuximab therapy, we collected 641 plasma samples from the plasma cohort composed of two independent cohorts including plasma discovery cohort (CRC patients (CRCs), $N = 116$; and healthy controls (HCs), $N = 66$) and plasma validation cohort (CRCs, $N = 31$; and HCs, $N = 24$). In the plasma cohort, we collected 89 pre-treatment plasma samples and 385 post-treatment plasma samples from CRC patients during continuous multiple courses anti-EGFR therapy in the plasma discovery cohort; and 31 pre-treatment plasma samples and 46 post-treatment plasma samples were collected in the plasma validation cohort. In addition, we included 31 tumor tissues and 27 paired NATs originated from 31 therapy-naïve CRC patients matched with the plasma samples in the independent tissue validation cohort. For CRC patients, patients pathologically diagnosed with CRC at Fudan University Shanghai Cancer Center (Shanghai, China) were included in this study retrospectively. The anti-EGFR therapy regimen was given at standard dosing as described in previous studies, of which patients were given 500 mg/m² cetuximab once-every-2-weeks combined with FOLFOX/FOLFIRI/irinotecan[30–32]. The post-treatment sampling was

acquired after each course of the treatment (up to eight weeks), and this longitudinal cohort included in this study covered up to 9 sampling time points during the treatment period. The inclusion criteria were as follows: diagnosis of CRC; presence of at least one measurable or unmeasurable but evaluable lesion (described according to Response Evaluation Criteria in Solid Tumors [RECIST] 1.1); presence of polymerase chain reaction (PCR)-confirmed wild-type *KRAS* (exon 2/3/4), *NRAS* (exon 2/3/4), and *BRAF* (exon 15) genotypes in tumor tissue before the receipt of anti-EGFR therapy; no history of severe heart or liver disease, psychiatric disorders, hemorrhage, or perforation of the digestive tract; and an Eastern Cooperative Oncology Group (ECOG) performance status of 0/1 at 3 days before treatment[33] (Methods). According to the National Comprehensive Cancer Network® (NCCN®) stated the NCCN Clinical Practice Guidelines in Oncology (NCCN Guidelines®) for available at NCCN.org, all the CRC patients receiving cetuximab therapy included in this study were left-sided *RAS* wild-type metastatic CRC. All the baseline clinical characteristics of the individuals included in this study were collected, including age, gender, degree of tumor differentiation, ECOG performance-status score, and other biochemical indicators (such as lactate dehydrogenase (LDH) level, white blood cell (WBC) count, lymphocyte number (LYMPHN), hemoglobin (HB), and platelet count) (Supplementary Data 1). Statistical analysis revealed that age and sex distributions were balanced among the CRC patients (CRCs) and healthy controls (HCs) both in plasma discovery cohort, plasma validation cohort and tissue validation cohort, showing there was no selection bias among these individuals in this study. In addition, there was no statistically significant difference in the clinical parameters of serum LDH level, WBC count, lymphocyte number, hemoglobin, and platelet count among the CRC groups (Table 1). An overview of proteomics workflow was shown in Fig. 1A.

## The plasma proteomic biomarkers to distinguish CRC patients from healthy controls

For the quality control of the performance of MS, the mixture of all plasma samples was measured every twenty samples, which was

adopted in proteomic studies[34–36]. A Pearson's correlation coefficient was calculated for all the quality-control runs, and the results were shown in Supplementary Fig. 1A. The average correlation coefficient among the control samples was 0.978, demonstrating the consistent stability of the MS platform. Proteomics measurement resulted in 1587–2502 gene products (GPs) in each sample (Supplementary Fig. 1B). A total of 9852 gene products (GPs) were identified in all plasma samples of the plasma discovery cohort, of which 9714 GPs were identified in CRC patients (CRCs, $N = 474$), and 7512 GPs were identified in healthy controls (HCs, $N = 66$) (Fig. 1B and Supplementary Fig. 1B; Supplementary Data 2). To explore the molecular difference of CRC patients from healthy controls, we then performed the comparative proteomic analysis of the pre-treatment CRCs ($N = 89$) and HCs. Firstly, we found no major differences in the proteomic coverage between the pre-treatment CRCs (7967 GPs) and HCs (7512 GPs) (two-sided Student's $t$ test, $P > 0.05$) (Fig. 1C). We observed that 78.9% GPs (6829 GPs) were commonly identified in the pre-treatment CRCs and HCs, which were majorly grouped into secreted proteins according to the protein classes in the Human Protein Atlas (HPA) database. Importantly, 13.2% proteins (1138 GPs) were exclusively identified in the pre-treatment CRCs and HCs; among these proteins, CRC related proteins ranked the higher proportion, representing plasma proteome could mostly reflected molecular alteration of CRC (Fig. 1C, D). Our study has so far presented a comprehensive view of the plasma proteomic landscape of the CRC cohort.

To detect changes in the plasma proteome between the pre-treatment CRC patients and healthy controls, we compared the proteome profiles of CRC and HC groups and identified a total of 1269 differentially expressed proteins (DEPs), of which 745 proteins up-regulated in the CRC group and 524 proteins up-regulated in the HC group (adj $P$ value < 0.05, fold change >2) (Fig. 1E and Supplementary Data 3). We then performed pathway enrichment analysis based on the DEPs according to the ConsensusPathDB (CPDB) molecular interaction data obtained from 31 different public repositories[37]. As shown in Fig. 1F, G, we found that glycolysis/gluconeogenesis, signaling by ERBB2, cellular response to chemical stress, MAPK activation, and

## Table 1 | Baseline characteristics in the discovery and validation cohorts

|  | Plasma discovery cohort | | Plasma validation cohort | | Tissue validation cohort | P-value |
|---|---|---|---|---|---|---|
|  | Healthy control (N = 66) | CRC (N = 116) | Healthy control (N = 24) | CRC (N = 31) | CRC (N = 31) |  |
| Age (years), median (range) | 62 (57–63) | 55.5 (21–76) | 55 (25–68) | 56 (29–77) | 57 (25–76) | 0.660[a] |
| Gender |  |  |  |  |  | 0.334[b] |
| Female | 44 (66.7%) | 36 (31.0%) | 12 (50.0%) | 11 (34.4%) | 14 (45.2%) |  |
| Male | 22 (33.3%) | 80 (69.0%) | 12 (50.0%) | 20 (65.6%) | 17 (54.8%) |  |
| Degree of tumor differentiation |  |  |  |  |  | 0.444[b] |
| Poorly differentiated | - | 29 (25.0%) | - | 5 (15.6%) | 11 (35.5%) |  |
| Moderately differentiated | - | 54 (46.6%) | - | 17 (56.2%) | 16 (51.6%) |  |
| Well differentiated | - | 2 (1.7%) | - | 0 (0.0%) | 0 (0.0%) |  |
| NA | - | 31 (26.7%) | - | 9 (28.1%) | 4 (12.9%) |  |
| ECOG performance status |  |  |  |  |  | 0.511[b] |
| 1 | - | 111 (95.7%) | - | 31 (100%) | 31 (100%) |  |
| 0 | - | 5 (4.3%) | - | 0 (0.0%) | 0 (0.0%) |  |
| Lactate dehydrogenase level (U/L), median (range) | - | 237 (97–3000) | - | 184 (118–2231) | 235 (118–2231) | 0.262[a] |
| White Blood Cell count (10⁹/L), median (range) | - | 5.75 (1.7–15.2) | - | 6.1 (2.9–11.9) | 6.5 (2.9–14.8) | 0.529[a] |
| Lymphocyte number (10⁹/L), median (range) | - | 1.4 (0.3–3.1) | - | 1.7 (0.3–107) | 1.6 (0.9–107) | 0.139[a] |
| Hemoglobin (g/L), median (range) | - | 126 (39–163) | - | 127 (97–155) | 126 (39–152) | 0.365[a] |
| Platelet Count (10⁹/L), median (range) | - | 198.5 (48–561) | - | 206 (105–417) | 205 (112–506) | 0.692[a] |

[a] one-way ANOVA analysis was applied for continuous variables.
[b] two-sided Fisher's exact test was used for categorical variables.

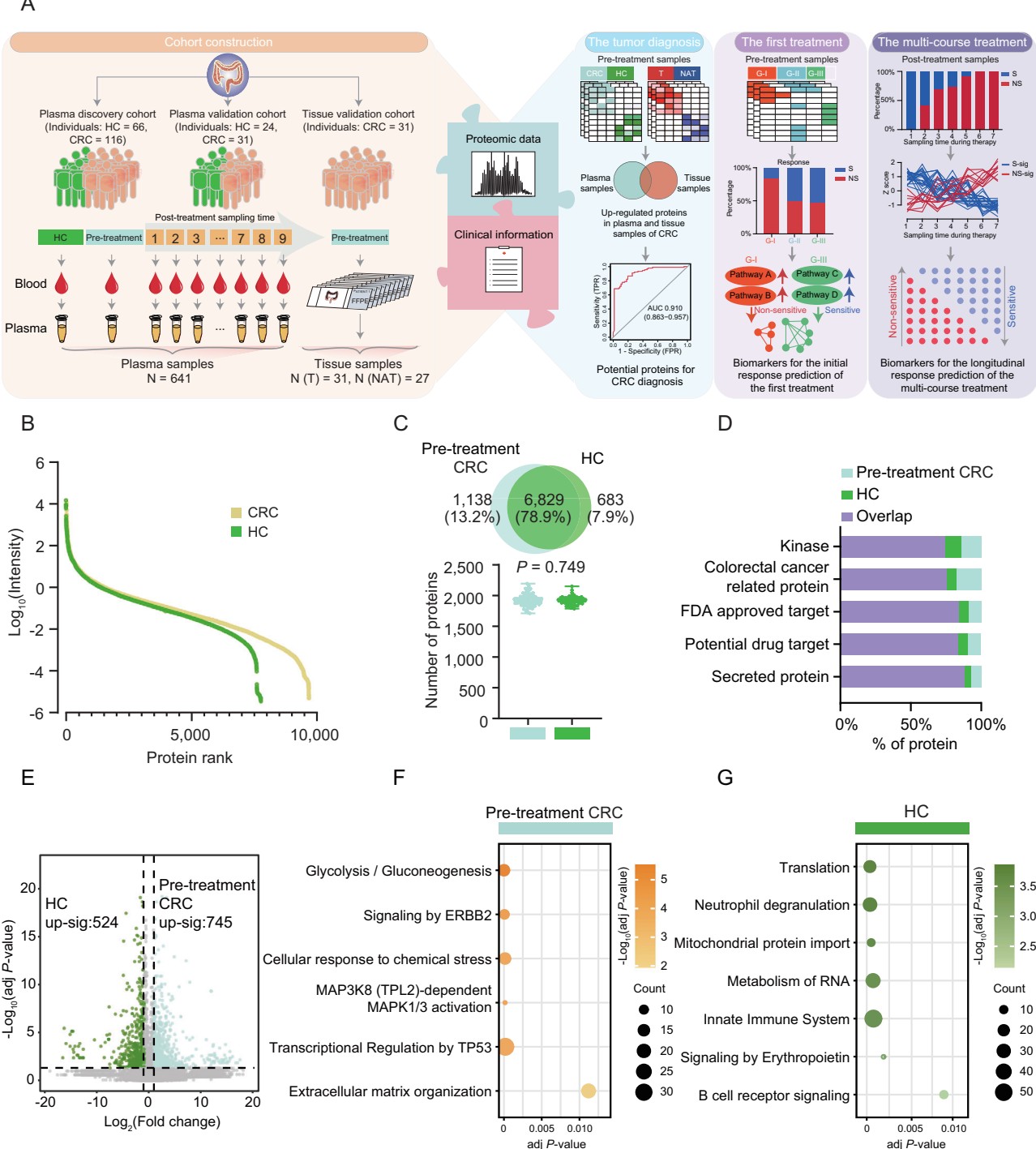

**Fig. 1 | Summary of the plasma proteomic analysis of CRC cohort and healthy control. A** Overview of the plasma proteomic workflow, including cohort construction (including plasma discovery cohort (CRC: *N* = 116 and Healthy control (HC): *N* = 66), plasma validation cohort (CRC: *N* = 31 and HC: *N* = 24), and tissue validation cohort composed of 31 CRC patients), proteomic profiling (data-independent acquisition (DIA), data-dependent acquisition (DDA), and parallel reaction monitoring (PRM)), and data analysis (proteomic dada and clinical information). For the plasma samples, the pre-treatment plasma samples and post-treatment plasma samples covering multi-course treatment were collected; for tissue samples, the pre-treatment samples were collected from the therapy-naïve CRC patients. **B** The dynamic range of the protein identification of CRC cohort and healthy control according to the descending sort of protein abundance in CRC and HC. Proteins are quantified as $\log_{10}$ transformed intensity. **C** Venn diagram showing the protein overlap of pre-treatment CRC and HC. The number of proteins is quantified in pre-treatment CRC patients (*N* = 89) and healthy controls (*N* = 66) (two-sided Student's *t* test, *P* = 0.749). Boxplots show median (central line), upper and lower quartiles (box limits), 1.5×interquartile range (whiskers). **D** The barplot showing the proportion of proteins in pre-treatment CRC patients and healthy controls (HCs). **E** Volcano plot showing the differential expression of pre-treatment CRC cohort and healthy control (two-sided Wilcoxon rank-sum test). The adj *P* < 0.05 is considered statistically significant. Blue, upregulated proteins in pre-treatment CRC patients; green, upregulated proteins in HCs. Bubble plots showing the CPDB pathway enrichment (two-sided Fisher's exact test) of pre-treatment CRC (**F**) and HC (**G**) groups. The adj *P* < 0.05 is considered statistically significant. Source data are provided as a Source Data file.

extracellular matrix organization pathways were enriched in the plasma samples of CRC patients. The plasma samples of healthy individuals were featured by neutrophil degranulation, innate immune system, etc. (adj *P* value < 0.05) (Fig. 1F, G; Supplementary Data 3). To explore how much of the alteration of these proteins and pathways in plasma could be reflected in the tumor, we reviewed all the archival formalin-fixed paraffin-embedded (FFPE) tissues from the therapy-naïve CRC patients included in this study. Then, we collected 31 tumor tissues and 27 paired NATs of CRC patients matched with the plasma samples, as the tissue validation cohort, for MS-based proteomic profiling. Proteomic analysis of tumor tissues and NATs identified 9258 GPs and 7452 GPs, respectively, of which 2232 GPs were specifically detected in tumor tissues (Supplementary Fig. 1D). In addition, we found the 6927 proteins were detected both in plasma and tissue samples (Supplementary Fig. 1E). We applied the same differential expression analysis in the tissue validation cohort, resulting in 2573 up-regulated proteins and 251 down-regulated proteins in tumor tissues (fold change (Tumor vs NAT) >2, adj *P* value < 0.05) (Supplementary Fig. 1F). Further comparison of the DEPs identified in plasma samples and tissue samples showed that, among the 745 proteins up-regulated in the plasma samples from CRC patients, 235 (31.54%) proteins were also up-regulated in tumor tissues from CRC patients (Supplementary Fig. 1G), which demonstrated a relatively high overlapped proportion of the DEPs in the plasma samples and tumor tissue samples of the CRC patients. To further explore the alteration of the pathways identified in the CRC plasma, we performed the pathway enrichment analysis based on these DEPs of the tissue validation cohort. Consistent with the enrichment results in the plasma proteome, we found that glycolysis and gluconeogenesis, adherens junction, TP53 transcriptional regulation, cellular response to chemical stress, and oncogenic MAPK signaling pathways were enriched in the tumor tissues; while immune regulation pathways such as complement system were enriched in the NATs (Supplementary Fig. 1H). These results revealed the alteration of proteins and pathways in plasma proteome had a relatively well reflection in the tissue samples. Therefore, a combined analysis of plasma and tissue proteome profile will identify robust biomarkers with potential clinical utility.

To search for the potential proteins applied for plasma and tissue samples that could be used as biomarkers for distinguishing CRC patients from healthy controls, we adopted strict screening strategy in the plasma and tissue samples as follows (Fig. 2A): (1) The candidate proteins were expressed in at least 50% of the samples; (2) The candidates were significantly increased in tumor samples than normal samples (two-sided Wilcoxon rank-sum test, adj *P* value < 0.01); (3) The candidates were identified with at least 2-fold increase in CRC samples than normal samples or NATs. At a result, we screened 148 proteins and 797 proteins significantly and stably overexpressed in the plasma samples and tumor tissues of CRC patients, respectively; among them, an overlapped 15 signatures (CPT1A, NUP205, CDC37, MAT2A, RPN1, GMPS, PSMA1, CDH1, SRSF7, FUBP3, PIGR, S100A8, S100A9, THBS2, and COL12A1) were elevated both in plasma and tissue samples of CRC patients. To further refine a protein panel to distinguish the CRC patients from healthy controls, we included the other independent CRC cohort in the Clinical Proteomic Tumor Analysis Consortium (CPTAC) study[38]. We applied the same criteria in the pairwise comparison between tumor tissue and matched non-tumor adjacent tissue proteomic data of the CRC cohort from CPTAC, which resulted in 31 proteins that were significantly overexpressed in tumor tissues of CRC patients. Combined the three independent cohorts, we ultimately narrowed down a group of four proteins, COL12A1, THBS2, S100A8, and S100A9, significantly increased both in plasma and tissues of CRC patients in our study and the CPTAC study (Fig. 2B). When the differential expression was dropped from 2-fold to 1.5-fold change, there were still only the four proteins (COL12A1, THBS2, S100A8, and S100A9) were screened among the three independent cohorts

(Supplementary Fig. 2A), indicating the four proteins could be used as the potential biomarkers to distinguish CRC patients from healthy controls. In addition, transcriptomic data from the public TCGA CRC cohort of TIMER2.0 database and immunohistochemical (IHC) staining data from the HPA database further confirmed the high expression of COL12A1, THBS2, S100A8, and S100A9 in CRC (Supplementary Fig. 2B, C). In a conclusion, the four proteins showed a consistent elevation both in plasma and tumor tissues of CRC patients, suggesting the potential to distinguish CRC patients from healthy controls.

Then, we performed gene set enrichment analysis (GSEA) for pathway enrichment analysis[39] to explore the potential biological association of the four proteins (COL12A1, THBS2, S100A8 and S100A9). The GSEA results demonstrated that the four proteins COL12A1, THBS2, S100A8, and S100A9 were enriched in collagen-containing extracellular matrix (GO:0062023) based on the cellular component database according to the Gene Ontology annotation (FDR = 0.03, NES = 1.34) (Fig. 2C). To validate this association, we performed single sample gene set enrichment analysis (ssGSEA)[40]. Further correlation analysis revealed that COL12A1, THBS2, S100A8, and S100A9 showed significantly positive correlation with extracellular matrix organization (ECM) ssGSEA score (Supplementary Fig. 2D). The protein-protein interaction network revealed a close connection among these proteins involving in ECM, especially for the four proteins including COL12A1, THBS2, S100A8, and S100A9 (Fig. 2C), consistent with the previous related reports[41–44]. Overall, the four biomarkers had the potential biological association, which showed significant enrichment to the collagen-containing extracellular matrix. Additionally, we also explored the association of the four proteins with clinical prognosis. We found a significantly negative correlation with clinical outcomes of the four proteins' expression of the TCGA CRC cohort in the CPTAC study[45] (two-sided log rank test, *P* < 0.05; hazard ratio (HR) >1). According to the HPA annotation, the four proteins belonged to secreted proteins and were identified in plasma; among them, S100A8 and S100A9 have known clinical utilities as potential markers and inhibitors (Fig. 2D).

To further determine whether these four proteins could effectively distinguish CRC patients from healthy controls, we used machine learning algorithms to assess the predictive capabilities of these proteins. We calculated the area under the curve (AUC) of the receiver operating characteristic (ROC) curve plot for the four proteins in the plasma discovery cohort, and found the AUC of at least 0.674 for a single marker (range: 0.674–0.843); and the AUC of the combined plasma markers was increased to 0.910 (95%CI: 0.863–0.957), exhibiting the better performance to distinguish CRC patients from healthy controls (Fig. 2E). To assess the predictive performance of the four proteins, 60% of samples were used as a training set, and the remaining 40% represented the independent testing set. Finally, the predictive model with a 10-fold cross validation yielded with high sensitivity (true positive rate) (90%) and specificity (true negative rate) (93.02%) (Fig. 2F). When applied to the independent testing set samples, the predictive model also achieved 72.6% accuracy (Fig. 2F). More importantly, the combined prediction of the four proteins achieved good performance with an AUC of 0.945 in the independent tissue validation cohort (Fig. 2G and Supplementary Fig. 2E). To further validate the prediction effect of the four proteins to distinguish CRC patients from healthy controls, we included an independent plasma validation cohort composed of 31 CRC patients and 24 healthy controls. As a result, the four proteins demonstrated a well distinguish of CRC patients from healthy controls with an AUC of 0.952 in the independent plasma validation cohort (Fig. 2G and Supplementary Fig. 2F).

Due to the four proteins were screened in the CRC patients receiving cetuximab therapy (*RAS* wild-type metastatic CRC patients), there was possible limitation of the clinical validity of the four proteins in CRC diagnosis. To explore whether the four proteins could also be applied on a wider population scale of CRC patients, we included a

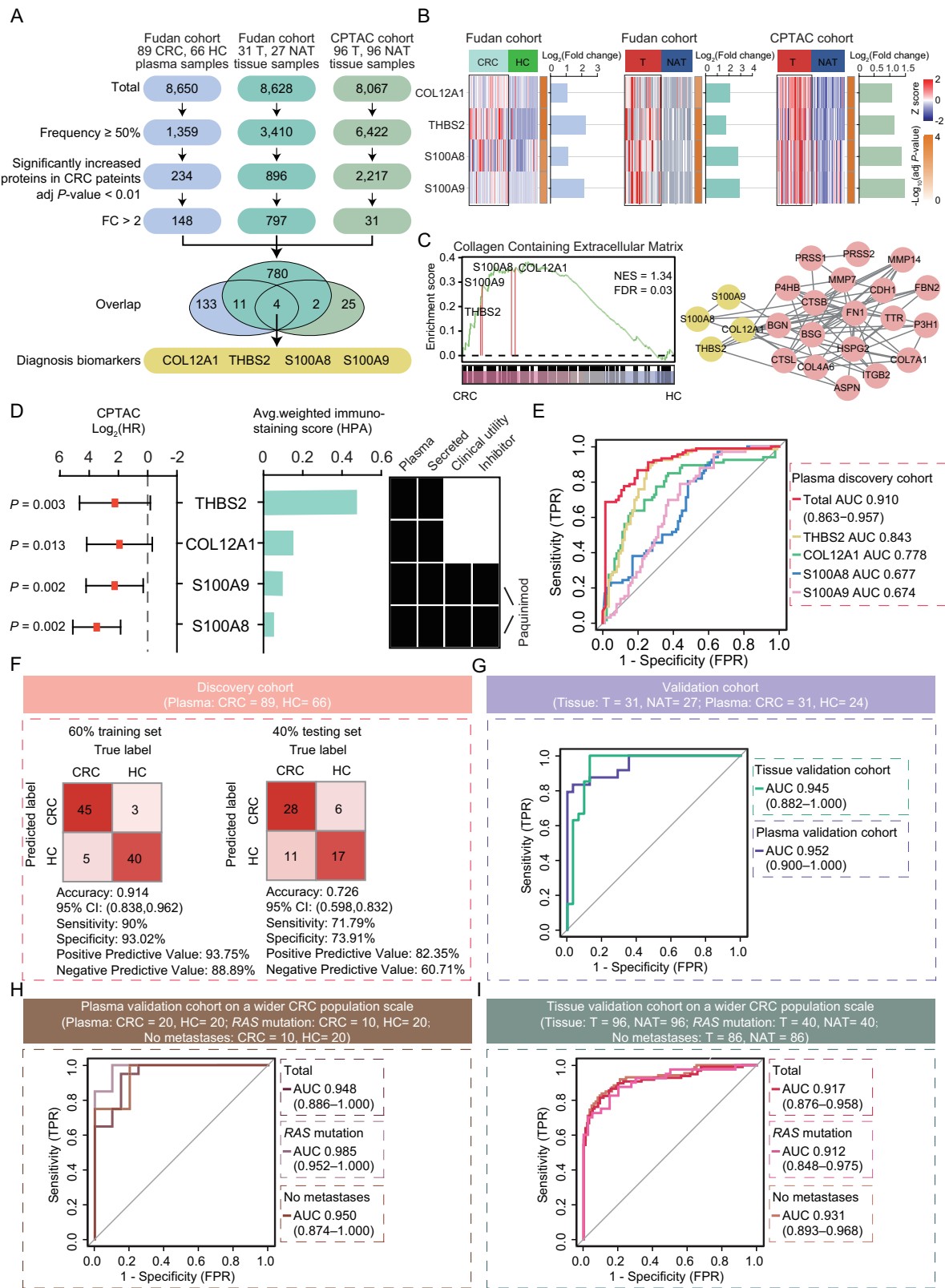

plasma validation cohort composed of patients of CRC ($N = 20$), of which 50% ($N = 10$) patients with CRC had *RAS* mutation and 50% ($N = 10$) patients with CRC had no tumor metastases, and healthy controls (HCs, $N = 20$). To eliminate the possible limitations that the four proteins could not distinguish CRC patients with *RAS* mutation and no metastases, we performed the differential analysis and ROC analysis based on the plasma proteomic data, respectively in total 20

CRC patients, CRC patients with *RAS* mutation, and CRC patients with no metastases. The differential analysis identified the four proteins (COL12A1, THBS2, S100A8, and S100A9) showed significant up-regulation in the total 20 CRC patients, and the ROC analysis demonstrated the four proteins showed a well distinguish of CRCs from HCs with an AUC of 0.948 (Fig. 2H and Supplementary Fig. 2G). As for the CRC patients with *RAS* mutation, we also found the four proteins

**Fig. 2 | Plasma protein biomarkers for diagnosis of CRC patients. A** The screening criteria of biomarkers for CRC diagnosis applied in Fudan cohort and the CPTAC CRC cohort. **B** The heatmaps showing the relative abundance (Z score) of the four proteins in Fudan cohort and the CPTAC CRC cohort. The little heatmaps and barplots showing the differential expression (two-sided Wilcoxon rank-sum test, adj P value < 0.05). **C** Left: the GSEA shows collagen containing extracellular matrix are enriched in CRC patients. Right: the protein-protein interaction network. **D** Left: hazard ratio (HR) of overall survival in CPTAC cohort (*N* = 95; two-sided log rank test, *P* value < 0.05) and immunohistochemistry (IHC) staining scores defined by the Human Protein Atlas (HPA) of these diagnostic biomarkers. Data are presented as median values (HR) with range (95%CI). Right: Overlap with plasma proteins and secreted proteins annotated by HPA, as well as clinical utilities and inhibitors. **E** The receiver operating characteristic (ROC) curves of the four proteins to distinguish CRCs from HCs. **F** Classification error matrix using logistic regression classifier of 60% training set and 40% testing set in distinguishing CRCs from HCs in the plasma discovery cohort. The number of samples identified is noted in each box. **G** The ROC curves of the four proteins in the independent tissue validation cohort and plasma validation cohort. The ROC curves of the four proteins to distinguish CRC patients from HCs or NATs, CRC patients with *RAS* mutation from HCs or NATs, and CRC patients with no metastases from HCs or NATs in the Fudan plasma validation cohort (**H**) and CPTAC tissue validation cohort (**I**) on a wider CRC population scale. Source data are provided as a Source Data file.

showed a consistent up-regulation in the CRC patients with *RAS* mutation; meanwhile, the four proteins achieved a high prediction with an AUC of 0.985 (Fig. 2H and Supplementary Fig. 2G). As for the CRC patients with no metastases, the differential analysis also showed a consistently significant up-regulation of the four proteins in the CRC patients with no metastases, with achieving a good performance with an AUC of 0.950 in distinguishing CRC patients with no metastases from HCs (Fig. 2H and Supplementary Fig. 2G).

In addition, to further verify whether the four proteins (COL12A1, THBS2, S100A8, and S100A9) showed consistent up-regulation in the tissue samples respectively from CRC patients with *RAS* mutation and no metastases, we also searched the public cohort from the CPTAC, which composed of a total of 96 tumor and matched normal adjacent tissues (NATs) pairs from CRC patients[38]. In the CPTAC CRC cohort, we found the four proteins showed a consistent up-regulation in the tissue samples from CRC patients, which suggested that the four proteins were indeed up-regulated in the tumor samples from CRC patients with *RAS* mutation and no metastases (Supplementary Fig. 2H). The ROC analysis demonstrated that the four proteins combined prediction could achieve a high prediction with an AUC of 0.917 for the CRC diagnosis, suggesting the stability and universality of the four proteins in CRC diagnosis (Fig. 2I). In the CPTAC CRC cohort, 42% (*N* = 40) patients with CRC had *RAS* mutation, and 90% (*N* = 86) patients with CRC had no tumor metastases. To further demonstrate the predictive efficient of the four proteins was not limited to the patients with *RAS* wild-type metastatic CRC, we further stratified 40 CRC patients with *RAS* mutation, of which ROC analysis showed a well distinguish from tumor and NATs with an AUC of 0.912 (Fig. 2I). In addition, we also stratified 86 non-metastatic CRC patients, of which ROC analysis also showed a good prediction with an AUC of 0.931 (Fig. 2I). Therefore, we validated the predictive effect of the four proteins on a wider population scale of CRC patients, but not limited to the *RAS* wild-type CRC patients or metastatic CRC patients.

To explore whether the four proteins were specific to CRC diagnosis rather than other cancers, we further enrolled a multi-cancer (including other six cancer types) plasma independent cohort composed of a total of 115 plasma samples, including 95 plasma samples from treatment-naïve patients with various cancer types (including colorectal cancer (CRC, *N* = 20), lung cancer (LC, *N* = 15), malignant lymphoma (ML, *N* = 10), bladder cancer (BLCA, *N* = 10), breast carcinoma (BRCA, *N* = 15), gastric cancer (GC, *N* = 10), esophageal cancer (EC, *N* = 15)), and 20 plasma samples from healthy controls (HCs, *N* = 20). The proteomic measurement showed the four proteins (COL12A1, THBS2, S100A8, and S100A9) were consistently up-regulated in the patients with CRC compared with HCs, while the four proteins didn't exhibited the consistency of up-regulation in patients with other cancers (including GC, EC, BRCA, LC, BLCA, and ML), suggesting the consistency of up-regulation of the four proteins only in CRC but not in other cancers. To further verify the predictive efficacy of the four proteins in distinguishing patients with CRC, but not other cancers, from HCs, we also performed the ROC analysis to evaluate the predictive effect. The ROC analysis revealed the four proteins combined prediction achieved good performance with an

AUC of 0.910 in distinguishing CRC patients from HCs, while the four proteins had poor performance with AUC values no more than 0.685 in distinguishing patients with other cancers from HCs (GC: 0.685, EC: 0.663, BRCA: 0.673, LC: 0.670, BLCA: 0.440, ML: 0.655) (Supplementary Fig. 2I). Overall, we validated the consistency of up-regulation of the four proteins (COL12A1, THBS2, S100A8, and S100A9) was only observed in CRC but not in other cancer types; and the four proteins could achieve good performance in the diagnosis of CRC, but not other cancer types.

In addition, to explore the predictive efficacy of the four proteins to distinguish patients with some non-malignant conditions, we also searched for the public datasets related to ulcerative colitis (an inflammatory bowel disease) and infection disease (such as SARS-CoV-2 infection). The GSE11223 dataset was composed of ulcerative colitis patients (UC, *N* = 129) and healthy controls (HCs, *N* = 73), and the GSE207015 dataset was composed of SARS-CoV-2 infected patients (COVID-19, *N* = 124) and non-infected healthy controls (*N* = 70). Then, the differential expression analysis showed the four proteins (COL12A1, THBS2, S100A8, and S100A9) showed no obvious change in the UC patients or the SARS-CoV-2 infected patients. To further investigate whether the four proteins could distinguish the patients with these non-malignant conditions, we performed the ROC analysis. The results revealed the four proteins combination could not distinguish the patients with the non-malignant diseases such as inflammatory diseases and infections from healthy controls with AUC values no more than 0.660 (UC: 0.659, SARS-CoV-2 infection: 0.612) (Supplementary Fig. 2I).

Overall, in this study, we validated that the combination of four proteins could well distinguish the CRC from HC, but not limited to the *RAS* wild-type CRC patients or metastatic CRC patients, validating the clinical validity of the four proteins on a wider population scale of CRC patients. In addition, we also validated that the four proteins could not distinguish the patients with other cancer types and the non-malignant diseases, demonstrating the relative specificity of the four proteins for the CRC diagnosis. These results demonstrated the robustness of the four proteins in distinguishing CRC patients from healthy controls. For the clinical validity of the potential biomarkers for CRC diagnosis, we recommended that it is deserved to apply these biomarkers in the consecutive patients in a colonoscopy cohort for the CRC diagnosis during several years in the future.

## The potential molecular features and biomarkers for the initial response to cetuximab therapy

Cetuximab therapy is the major targeted treatment of CRC, but drug resistance limits its effectiveness. As reported, the disease had progressed in more than 50% of treated patients at the first assessment of the CRC patients, which might due to tumor heterogeneity. To explore the potential biomarkers for predicting the therapy response, we focused on the baseline plasma proteome profiles of the 89 therapy-naïve CRC patients, and evaluated the initial therapy response after the first cetuximab treatment among these patients according to the widely accepted RECIST (version 1.1). Here, the objective response rate (ORR), defined as partial response (PR) plus complete response (CR),

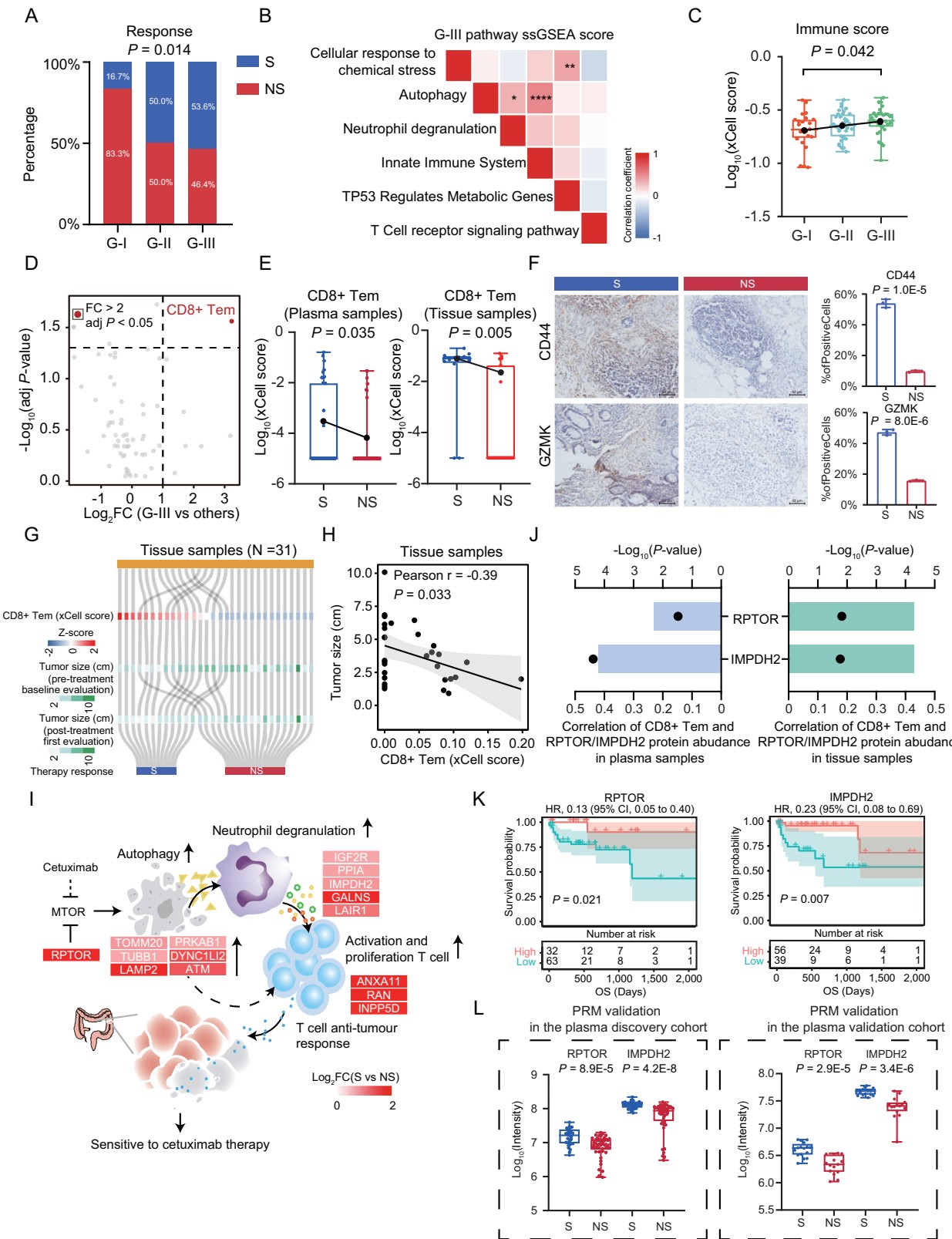

was selected for the efficacy evaluation; patients with CR and PR were defined as sensitive (S) and those with stable disease (SD) and progressive disease (PD) were defined as non-sensitive (NS), consistent with the previous research[46]. We preliminarily explored the association of diagnostic biomarkers (COL12A1, THBS2, S100A8, and S100A9) identified above with therapy response. As a result, the differential analysis of the diagnostic biomarkers showed no significant difference

between S and NS groups, and ROC analysis also indicated a poor prediction of the four biomarkers in initial therapy response (Supplementary Fig. 2J). Therefore, the diagnostic biomarkers, used for distinguishing the CRC patients from healthy controls, could not be regarded as the indicator for the initial therapy response prediction.

Although proteins have been regarded as the "executors of life", we have no direct evidence for that whether the proteome pattern can

**Fig. 3 | The potential mechanism and biomarkers for cetuximab sensitivity.**
**A** Association of proteomic subtypes with therapy response (two-sided Fisher's exact test). **B** Correlation of the pathways of the G-III subtype (two-sided Pearson's correlation test). **C, E** Boxplots for immune score among G-I ($N$ = 24), G-II ($N$ = 34), and G-III ($N$ = 31) ($P$ = 0.042) (**C**), and CD8+Tem score between S and NS groups in plasma samples (N (S) = 16, N (NS) = 15; $P$ = 0.035) and tissue samples (N (S) = 12, N (NS) = 19; $P$ = 0.005) (two-sided Student's $t$ test). **D** Differential cell types between G-III and other subtypes (two-sided Student's $t$ test). **F** Qualification of CD44 ($P$ = 1.0E-5) and GZMK ($P$ = 8.0E-6) stained by IHC in representative examples (two-sided Student's $t$ test). Data are shown as mean ± SD ($n$ = 3 independent experiments). **G** Association between CD8+Tem score (Z score) with tumor size (cm), and

S/NS group. **H** Correlation of CD8+Tem score and tumor size (two-sided Pearson's correlation test). Data are presented as Pearson r with 95%CI. **I** Diagram showing the potential mechanism of cetuximab sensitivity. The little heatmap depicted the log-transformed fold-change of S versus NS groups. **J** Correlation of CD8+Tem score and RPTOR/IMPDH2 (two-sided Pearson's correlation test). **K** Kaplan–Meier curves of Overall survival (OS) in the CPTAC cohort (two-sided log rank test). **L** Differential expression of RPTOR ($P$ = 8.9E-5, 2.9E-5) and IMPDH2 ($P$ = 4.2E-8, 3.4E-6) in discovery cohort (N (S) = 31, N (NS) = 45) and validation cohort (N (S) = 16, N (NS) = 15) (two-sided Wilcoxon rank-sum test). PRM, parallel reaction monitoring. Boxplots show median (central line), upper and lower quartiles (box limits), 1.5×interquartile range (whiskers) (**C**, **E**, and **L**). Source data are provided as a Source Data file.

distinguish different therapy responses. Therefore, herein, we firstly performed consensus clustering analysis, an unsupervised clustering method, on the 89 therapy-naïve plasma samples based on 1500 most variable proteins (Methods), which resulted in three subtypes: G-I ($N$ = 24), G-II ($N$ = 34), and G-III ($N$ = 31) (Supplementary Fig. 3A, B, and Supplementary Data 4). Further statistical analysis of all these clinical characteristics among the three proteomic subtypes revealed significant association of proteomic subtypes with therapy response (Fisher's exact test, $P$ = 0.014) (Fig. 3A), but this association was not observed with either grade, degree of tumor differentiation, ECOG, or any other biochemical indicators ($P$ > 0.05) (Table 2). These results demonstrated that the proteomic subtype could reflect the strong association with therapy response, which was irrespective of other clinical parameters. The therapy response exhibited a gradual decreasing resistance phenomenon from G-I to G-III, as the percentage of sensitive patients (S: CR and PR) dramatically increased from 16.7% in G-I to 53.6% in G-III, while the percentage of non-sensitive patients (NS: SD and PD) dramatically decreased from 83.3% in G-I to 46.4% in G-III (Fisher's exact test, $P$ = 0.014) (Fig. 3A). Comparative analysis of proteomic profiling resulted in 313 (G-I), 403 (G-II), and 416 (G-III) GPs (adj $P$ value < 0.05) among the three subtypes (Supplementary Data 4). Based on these differential proteins, we performed a functional enrichment analysis according to the CPDB database[37], and determined the dominant bioprocesses of each subtype: the G-I subtype, as the non-sensitive subtype, was characterized with PPAR-alpha pathway, integrin-linked kinase signaling, metabolism of vitamins and cofactors, extracellular matrix organization (ECM), RHO GTPase cycle and MAPK signaling pathway (adj $P$ value < 0.05); the G-III subtype, as the sensitive subtype, was featured by cellular response to chemical stress (adj $P$ value = 8.1E-6), neutrophil degranulation (adj $P$ value = 3.1E-5), autophagy (adj $P$ value = 4.9E-5), innate immune system (adj $P$ value = 1.2E-4), TP53 regulates metabolic genes (adj $P$ value = 8.2E-4), and T cell receptor signaling pathway (adj $P$ value = 2.1E-2); while for the G-II subtype, as the mixed subtype, was featured by DNA Double Strand Break Response (adj $P$ value = 1.2E-6), protein ubiquitination (adj $P$ value = 2.6E-5), protein processing in endoplasmic reticulum (adj $P$ value = 2.6E-5), spliceosome (adj $P$ value = 7.8E-4), hemostasis (adj $P$ value = 7.8E-4), and ribosome (adj $P$ value = 4.7E-2), in which the proteins involved showed no significant association with therapy response (Supplementary Fig. 3B, C). This preliminarily determined the association between proteome pattern and therapy response, which provided the direct basis for further comparison between sensitive group and non-sensitive group.

Next, we explored the potential resistant/sensitive mechanism and biomarkers for the initial therapy response based on the molecular features revealed by consensus clustering analysis. As a sensitive subtype, the overrepresented pathways of G-III subtype were validated by GSEA with significant enrichment of these pathways in G-III subtype compared with the other subtypes (FDR < 0.05) (Supplementary Fig. 3D). Consistently, we found the immune pathways, such as antigen processing, TCR signaling, and autophagy were enriched in the sensitive group in the tissue validation cohort (Supplementary Fig. 3E).

Further analysis based on the ssGSEA scores showed that among these pathways enriched in G-III subtype, autophagy showed significant association with neutrophil degranulation (Pearson $r$ = 0.26, $P$ = 0.014) or innate immune system (Pearson $r$ = 0.44, $P$ = 1.71E-5), indicating the biological association of autophagy with neutrophil degranulation and innate immune system (Fig. 3B). As reported, the tumor microenvironment infiltration estimated by proteomic data had a high Pearson correlation with ones estimated by transcriptomic data concluded from these published researches, indicating the potential of proteome in xCell analysis to reveal the tumor microenvironment infiltration[47,48]. We then evaluated the immune microenvironment among three subtypes by xCell analysis (Supplementary Data 4). We found that G-III subtype had the highest immune score compared with other two subtypes (Fig. 3C), and CD8+ Tem was significantly enriched in G-III subtype (fold change >2, adj $P$ value < 0.05) (Fig. 3D). To explore the association of these cell types with therapy response, we compared the xCell scores between S group and NS group. We found the xCell score of CD8+Tem was significantly dominant in S group (Fig. 3E). To validate this finding, we performed xCell analysis on the tissue proteomic data of the independent tissue cohort composed of 12 S patients and 19 NS patients. Consistently, the xCell score of CD8+Tem was significantly elevated in S group compared with NS group based on the tissue proteome (Fig. 3E). Furthermore, we performed immunohistochemistry (IHC) of the representative signatures (CD44 and GZMK) of the CD8+ Tem[49] to evaluate tumor infiltration in tissue samples from therapy-naïve CRC patients. As a result, the expression of CD44 and GZMK was significantly increased in S group compared with NS group. Moreover, S group had higher percentage of CD44 positive cells (53.9%) and GZMK positive cells (47.2%), compared with NS group (9.7% and 15.7%, respectively) ($P$ < 1E-4) (Fig. 3F). Overall, these results demonstrated that high level of CD8+Tem associated with cetuximab sensitivity, both in plasma and tissue proteomic data.

To further explore the clinical implication of CD8+Tem, we further associated the CD8+Tem score with tumor size evaluated by CT/MRI (Supplementary Fig. 3F). The results demonstrated patients with higher CD8+Tem xCell score were prone to have a smaller tumor size in the baseline evaluation (Fig. 3G). Further correlation analysis showed there was a significantly negative correlation (Pearson $r$ = −0.39, $P$ = 0.033) between tumor size and CD8+Tem xCell score (Fig. 3H), implying the CD8+Tem could be the potential marker for the cetuximab sensitivity. To further validate this finding, we included the single-cell transcriptome data from CRC patients, which provided the reference of microenvironment in CRC and could be obtained from Gene Expression Omnibus (accession number GSE108989)[49]. Then, we combined the single-cell transcriptome data and the clinical characteristics of each patient for the further analysis. We performed the correlation analysis of the CD8+Tem cell percentage and tumor size in the single-cell transcriptome data. Consistent with our findings uncovered by the proteome data, the result demonstrated the inferred proportion of CD8+Tem cell showed a significantly negative correlation (Spearman $r$ = −0.72, $P$ = 0.019) with tumor size, which further confirmed the association of CD8+Tem cell with tumor size

**Table 2 | The association of proteomic subtypes and other clinical variables**

| | G-I (N = 24) | G-II (N = 34) | G-III (N = 31) | P value |
|---|---|---|---|---|
| Age (years), median (range) | 53.5 (24–69) | 55 (21–74) | 61 (25–76) | 0.287[a] |
| Gender | | | | 0.571[b] |
| Female | 7 (29.2%) | 8 (23.5%) | 11 (35.5%) | |
| Male | 17 (70.8%) | 26 (76.5%) | 20 (64.5%) | |
| Degree of tumor differentiation | | | | 0.232[b] |
| Poorly differentiated | 7 (29.2%) | 12 (35.3%) | 5 (16.1%) | |
| Moderately differentiated | 11 (45.8%) | 16 (47.1%) | 14 (45.2%) | |
| Well differentiated | 0 (0.00%) | 1 (2.94%) | 0 (0.00%) | |
| NA | 6 (25.0%) | 5 (14.7%) | 12 (38.7%) | |
| ECOG performance status | | | | 0.187[b] |
| 1 | 22 (91.7%) | 34 (100%) | 29 (93.5%) | |
| 0 | 2 (8.33%) | 0 (0.00%) | 2 (6.45%) | |
| Lactate dehydrogenase level (U/L), median (range) | 232.5 (126–1121) | 330 (135–3000) | 326 (119–3000) | 0.190[a] |
| White Blood Cell count ($10^9$/L), median (range) | 5.5 (2.5–14.8) | 5.95 (1.7–14.7) | 5.8 (2.8–10.5) | 0.227[a] |
| Lymphocyte number ($10^9$/L), median (range) | 1.35 (0.4–2.2) | 1.4 (0.6–2.2) | 1.4 (0.3–3.1) | 0.502[a] |
| Hemoglobin (g/L), median (range) | 131 (88–155) | 120.5 (75–163) | 124 (39–163) | 0.149[a] |
| Platelet Count ($10^9$/L), median (range) | 171.5 (48–366) | 210.5 (107–452) | 216 (102–313) | 0.069[a] |

[a]one-way ANOVA analysis was applied for continuous variables.
[b]two-sided Fisher's exact test was used for categorical variables.

(Supplementary Fig. 3G). These results demonstrated that CD8+Tem could be regarded as a potential marker for the cetuximab sensitivity.

Then, we investigated the potential biological interaction associated with the therapy response. As reported, autophagy has been incorporated into multiple innate and adaptive immune pathways[50,51]. Accumulating evidence indicates that autophagy has important roles in regulating neutrophil functions, including degranulation, metabolism, and NET formation[52]. Autophagy deficiency disrupts neutrophil degranulation[53]. We presented the potential comprehensive network diagram, in which the related proteins involved in autophagy, neutrophil degranulation and T cell activation, were predominant in S group (Fig. 3I). Among these proteins, RPTOR and IMPDH2 were significantly positive correlated with CD8+Tem both in plasma and tissue samples ($r \geq 0.23$, $P < 0.05$) (Fig. 3J, and Supplementary Fig. 3H, I), and the high expression of RPTOR and IMPDH2 were significantly associated with better prognosis validated by CPTAC cohort (two-sided log rank test, HR < 1, $P < 0.05$) (Fig. 3K). In addition, the key sensitive regulators (RPTOR and IMPDH2) identified by the data-independent acquisition (DIA) strategy were further validated by parallel reaction monitoring (PRM) assay in the plasma samples from the plasma discovery cohort and another independent plasma validation cohort composed of 16 sensitive (S) patients and 15 non-sensitive (NS) patients (Fig. 3L and Supplementary Data 4). Overall, the high activation of autophagy and the aggregation of CD8+ Tem, might result in an improved response to cetuximab therapy.

As a non-sensitive subtype, the G-I subtype had the highest proportion of NS patients (Fig. 4A), which was characterized by PPAR-alpha pathway, integrin-linked kinase signaling, metabolism of vitamins and cofactors, extracellular matrix organization (ECM), RHO

GTPase cycle and MAPK signaling pathway (adj $P$ value < 0.05) (Fig. 4B). To further determine the association of pathways enriched in G-I subtype with treatment response, we performed ssGSEA analysis between S group and NS group. As a result, the ssGSEA pathway scores of MAPK signaling, RHO GTPase cycle, and ECM pathway were significantly increased in NS group ($P < 0.05$), which were validated by GSEA with significant enrichment of these pathways in the NS group (FDR < 0.05) (Fig. 4C, D), showing these signaling pathways might as the indicator of non-response to cetuximab therapy. Clinically, the *KRAS* mutation status is predictive of response to cetuximab therapy in CRC. Cetuximab has been proven to be effective for patients with *KRAS* wild type CRC in randomized clinical trials[53–56]. However, only a small percentage of CRC patients are sensitive to anti-EGFR therapy, and even those who initially respond to the therapy eventually develop resistance to it[57–59]. The other RAS-related subfamily members (RRAS and RRAS2) approximately 55–60% identical to their classic counterparts and share activating signals and cascades with KRAS proteins (such as regulation of the MAPK signaling pathway)[60–62]. The association of these members with response to cetuximab therapy remains unclear. In this study, we found that CRC patients with high expression of RRAS and RRAS2 were prone to be resistant to the cetuximab therapy (FC > 2, $P < 0.05$) (Fig. 4E). Then, we wondered and explored which downstream pathway was activated accompanied with high expression of RRAS/RRAS2. According to the average expression of RRAS/RRAS2, we divided the cohort into RRAS high expression group and RRAS low expression group. Interestingly, we found the ssGSEA scores of the MAPK signaling, RHO, and ECM pathways were significantly upregulated in the RRAS high expression group ($P < 0.05$) (Fig. 4F), which were validated by GSEA (FDR < 0.05) (Fig. 4G). As shown in Fig. 4H, we proposed the potential regulation axis that RRAS/RRAS2 proteins positively regulate the activation of the three downstream pathways, which showed association with resistance to cetuximab therapy. Among proteins involved in these pathways, FBLN1, MMP8, and ITGA5 showed significantly positive correlation with RRAS/RRAS2, and the high expression of FBLN1, MMP8, and ITGA5 were significantly associated with worse prognosis validated in CPTAC cohort (HR > 1, two-sided log rank test, $P < 0.05$) (Fig. 4I, J, K). These results demonstrated that RRAS/RAS2 positively regulate the ECM pathway, which associated with resistance to cetuximab therapy as well as poor prognosis of CRC patients. Finally, we evaluated the ability of the combination of these proteins (RRAS2, MMP8, FBLN1, RPTOR, and IMPDH2) to predict initial response to the first cetuximab treatment. The ROC analysis revealed the combination of these proteins yielded high accuracy with an AUC of 0.849 (range: 0.765–0.932). To further validate the robustness of this protein combination, we used the machine learning algorithms in the independent tissue and plasma validation cohorts. The results of ROC analyses exhibited well performance in distinguishing S patients from NS patients with AUC values of 0.816 (95%CI: 0.645–0.986) and 0.890 (95%CI: 0.757–1) in the tissue and plasma validation cohorts, respectively (Fig. 4L).

Taken together, the proteomic subtypes were identified with distinct biological functions and associated with therapy response to cetuximab. Specifically, patients of G-III subtype tended to benefit from cetuximab therapy, accompanied by anticancer immune response such as activation of T cell receptor (TCR) signaling and increase of CD8+ Tem; while patients of G-I subtype were unlikely to benefit from it, featured by RRAS and RRAS2-mediated ECM pathway activation. Importantly, we identified key proteins involved in these overrepresented pathways, including RRAS2, MMP8, FBLN1, RPTOR, and IMPDH2, which yielded a high prediction in distinguishing the sensitive patients from the non-sensitive patients receiving cetuximab therapy, which were validated in the independent plasma and tissue validation cohorts, demonstrating the stability and robustness of the predictive model for predicting initial response to the first cetuximab treatment.

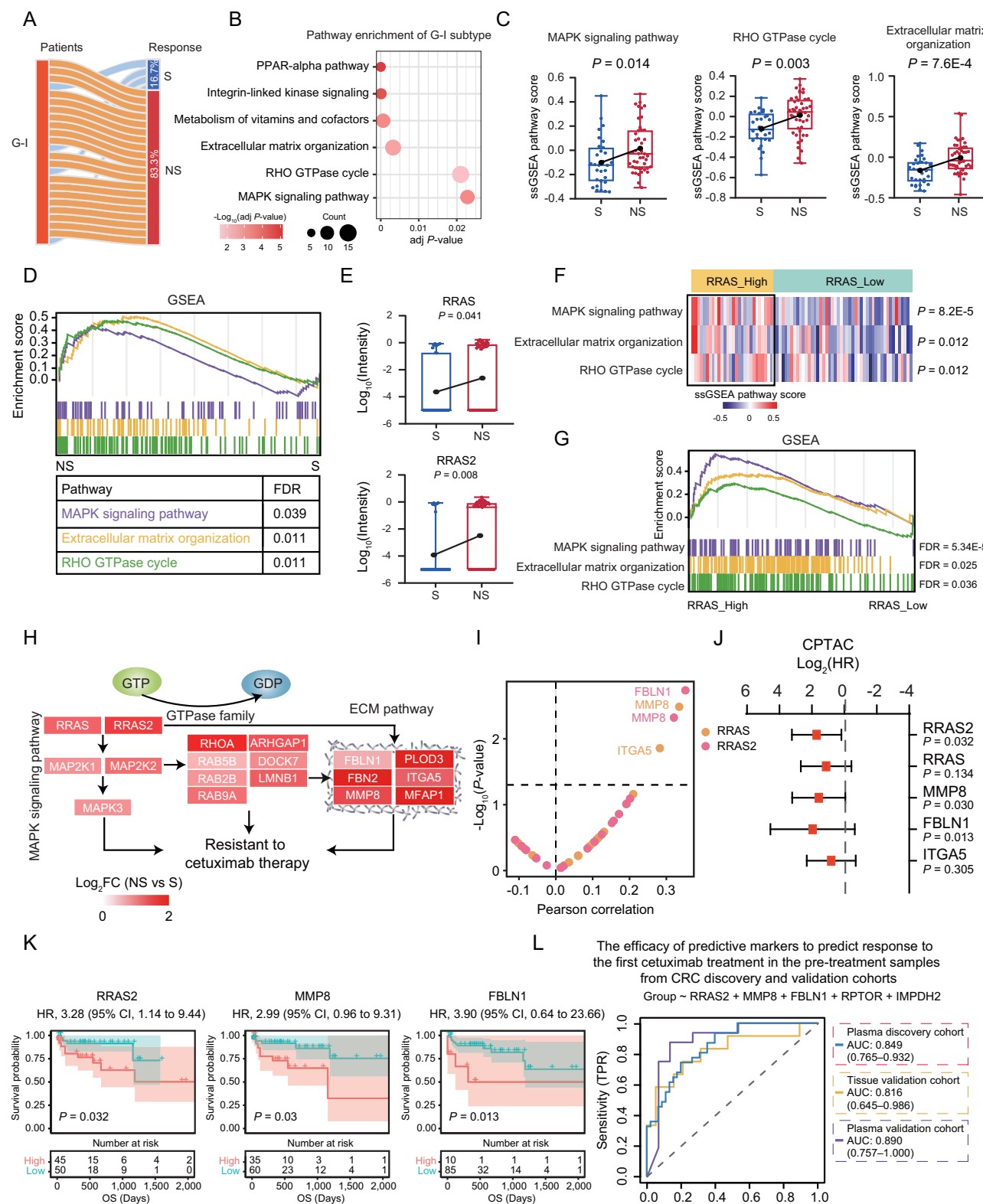

## Construction and validation of the predictive models applied for the multi-course cetuximab therapy in CRC

In this study, the plasma sampling of the plasma discovery cohort covered a cetuximab therapy period of up to 72 weeks from the therapy-naïve sampling, of which the plasma from each patient were collected when completing each therapy course (8-week therapy), resulting in a total of 474 plasma samples from 116 CRC patients covering 9 sampling time points during the cetuximab treatment period

(Methods). In the longitudinal cohort, one subset (defined as subcohort 1) was composed of 22 CRC patients sensitive to cetuximab therapy in the first treatment and gradually resistant in the following seven course cetuximab treatments, in which 105 plasma samples were collected; another subset (defined as subcohort 2) was composed of stable sensitive group (SSG: 18 patients, 38 samples) and stable non-sensitive group (SNSG: 58 patients, 153 samples), featured with consistently sensitive or non-sensitive to cetuximab therapy regardless of

**Fig. 4 | The potential mechanism and biomarkers for cetuximab resistance.**
**A** Proportion of responses to cetuximab therapy in G-I subtype. **B** The CPDB pathway enrichment (two-sided Fisher's exact test) of G-I subtype. **C** Boxplots for pathway ssGSEA score between S and NS groups (two-tailed Student's $t$ test). N (S) = 31, and N (NS) = 45. Boxplots show median (central line), upper and lower quartiles (box limits), 1.5×interquartile range (whiskers). **D** GSEA enrichment of MAPK signaling, RHO GTPase cycle, and ECM pathway in the NS group compared with S group (FDR < 0.05 is considered statistically significant). **E** Boxplots for RRAS and RRAS2 protein abundance between S and NS groups. N (S) = 31, and N (NS) = 45 (two-tailed Wilcoxon rank-sum test). Boxplots show median (central line), upper and lower quartiles (box limits), 1.5×interquartile range (whiskers). **F** Heatmap showing the relative abundance of ssGSEA pathway scores between RRAS low and

RRAS high groups (two-tailed Student's $t$ test). **G** GSEA showing the enrichment of MAPK signaling, RHO GTPase cycle, and ECM pathway in the RRAS high expression group compared with RRAS low expression group. **H** Diagram showing the potential mechanism of resistant to cetuximab therapy of CRC patients. The little heatmap under each protein depicted the log-transformed fold change in NS versus S groups. **I** Correlation of RRAS or RRAS2 and proteins involved in the three pathways (shown in (**H**)) (two-sided Pearson's correlation test). **J, K** The hazard ratio (HR) and Kaplan–Meier curves of OS in the CPTAC cohort ($N$ = 95) based on the protein abundance (two-sided log rank test). Data are presented as median values (HR) with range (95%CI). **L** The ROC curves of a panel of proteins in predicting drug sensitivity in the plasma discovery cohort and its predictive performance in the tissue and plasma validation cohorts. Source data are provided as a Source Data file.

the course of treatment in the nine courses cetuximab treatment. Taken the longitudinal characteristics of our cohort into consideration, to explore whether the biomarkers for CRC diagnosis and the initial response prediction of the first treatment could be used for the longitudinal response prediction of the multi-course treatment, we applied the diagnostic model and predictive model in the longitudinal cohort covering multi-course treatment. As a result, the diagnostic model composed of the four biomarkers (COL12A1, THBS2, S100A8, and S100A9), and the predictive model composed of the key proteins (RRAS2, MMP8, FBLN1, RPTOR, and IMPDH2) didn't well distinguish sensitive patients from non-sensitive patients during the cetuximab treatment (AUC < 0.7) (Supplementary Fig. 4A, B).

Therefore, to identify the protein dynamic change associated with the treatment response to cetuximab during multiple treatment courses, we focused on the subcohort 1 of the longitudinal cohort. According to the cetuximab treatment response trajectory, we performed statistical analysis of the distribution of CRC patients with sensitive (S)/non-sensitive (NS) to cetuximab treatment during treatment period. We found that the proportion of NS was gradually increased with the sampling time during cetuximab treatment (Fig. 5A and Supplementary Fig. 4C). Then, we explored the regulations of plasma protein levels during multiple therapy courses, which resulted in 139 significant positively (Positive correlation-sig) and 374 negatively correlated proteins (Negative correlation-sig) ($P$ < 0.05) (Fig. 5B and Supplementary Data 5). Taken together, the course-resolved analyses implicate that a large proportion of the quantified plasma proteins are significantly altered in the course of cetuximab treatment.

To further confirm the association of these proteins with response to cetuximab treatment, we performed a comparative proteomic analysis of the subcohort 2 of the longitudinal cohort. The differential analysis between SSG and SNSG groups identified the DEPs with 1.5-fold changes of each group (adj $P$ value < 0.05) (Supplementary Fig. 4D and Supplementary Data 5). We then performed pathway enrichment on the DEPs of each group according the CPDB molecular interaction annotations shown in Supplementary Fig. 4E. The SNSG group was dominant for glycolysis/gluconeogenesis, VEGFA-VEGFR2 signaling pathway, signaling by Rho GTPases, ECM proteoglycans, etc.; while SSG group was featured by downregulation of ERBB4 signaling, mRNA Splicing, autophagy, iron uptake and transport, cellular response to heat stress, ferroptosis, Neutrophil degranulation, etc. (Supplementary Fig. 4E). Ultimately, we screened for a panel of proteins identified in the overlap of both the negative correlation-sig (with a range of $r$: −0.98 to −0.75) and the SSG-sig, and defined them as sensitive biomarkers of cetuximab therapy in CRC patients ($N$ = 29). Similarly, we also found the non-sensitive biomarkers of cetuximab therapy in CRC patients (with a range of r: 0.75–0.94) ($N$ = 10) (Fig. 5C). The K-means plot demonstrated sensitive biomarkers presented a gradual downward trend with the increase of sampling time during cetuximab treatment, and non-sensitive biomarkers showed a gradual upward trend with the increasing courses during cetuximab treatment (Fig. 5C, D). We observed an obvious differential expression of these biomarkers in SSG and SNSG, and the dynamic changes across seven

courses cetuximab treatment (Fig. 5D). This highlights the potential application of the plasma biomarkers for efficient monitoring during the continuous multiple courses to cetuximab treatment.

Having identified biomarkers with fluctuations at protein level associated with response to cetuximab therapy, we next set out to determine whether these biomarkers could be used for predicting response to cetuximab treatment in CRC patients during the continuous courses. We employed stepwise logistic regression, which is robust to noise and overfitting, to identify a subset of signatures that accurately discriminates SSG and SNSG (named as S-sig and NS-sig). The S-sig include IDH3G, MDN1, and KLC4, while the NS-sig include MYL9, SBF1, and HTRA3. To train and subsequently test the model, samples were partitioned based on sample type; among them, 60% of samples were used as a training set, and the remaining 40% represented the independent testing set. Based on S-sig and NS-sig, we applied 10-fold cross-validation to the training set yielded predictive model with high AUC (0.756); when applied to the testing set, the predictive model also achieved an AUC of 0.797 (Fig. 5E). This suggested that the combination of S-sig and NS-sig proteins (IDH3G, MDN1, KLC4, MYL9, SBF1, and HTRA3) could predict the response to cetuximab treatment in CRC patients. We further determined the ability of the predictive model for the response of cetuximab treatment across different courses. After applying the predictive model to different treatment courses, we observed the predictive model had a good performance in the overall therapy courses with an accuracy of 0.724. The ability of this plasma-protein panel to identify sensitive group and non-sensitive group in the subset of different treatment courses is demonstrated in the bar graph (Fig. 5F). Importantly, this model also achieved high accuracy in predicting treatment efficacy in the second course (0.818), the third course (0.737), and the fourth course (0.800) (Fig. 5F and Supplementary Fig. 4F). To improve the interpretation of the predictive power of these signatures, we also presented the dynamic of these signatures over the multiple courses of cetuximab treatment in individual patient. The results demonstrated that, the S-sig proteins (including MDN1, KLC4, and IDH3G) exhibited a significant drop when resistance emerged, and maintained a gradual downward trend in the subsequent courses of cetuximab treatment; while the NS-sig proteins (including SBF1, HTRA3, and MYL9) exhibited a significant rise when resistance emerged, and maintained a gradual upward trend in the subsequent courses of cetuximab treatment (Supplementary Fig. 5). Overall, we observed a consistent expression pattern of these signature proteins across the multiple courses of cetuximab treatment, indicating the effectiveness of these biomarkers for predicting cetuximab response of individual patient. Subsequently, the accuracy of the predictive model was further validated in the independent cohort composed of 31 CRC patients, 77 plasma samples, covering two course cetuximab treatment. The results of confusion matrix analyses exhibited an accuracy of 0.929 (95%CI: 0.765–0.991) in the first course treatment, and 1 (95%CI: 0.815–1) in the second course treatment in the plasma validation cohort (Fig. 5G). Taken together, the predictive model had high accuracy prediction performance in different treatment course, exhibiting good performance in either

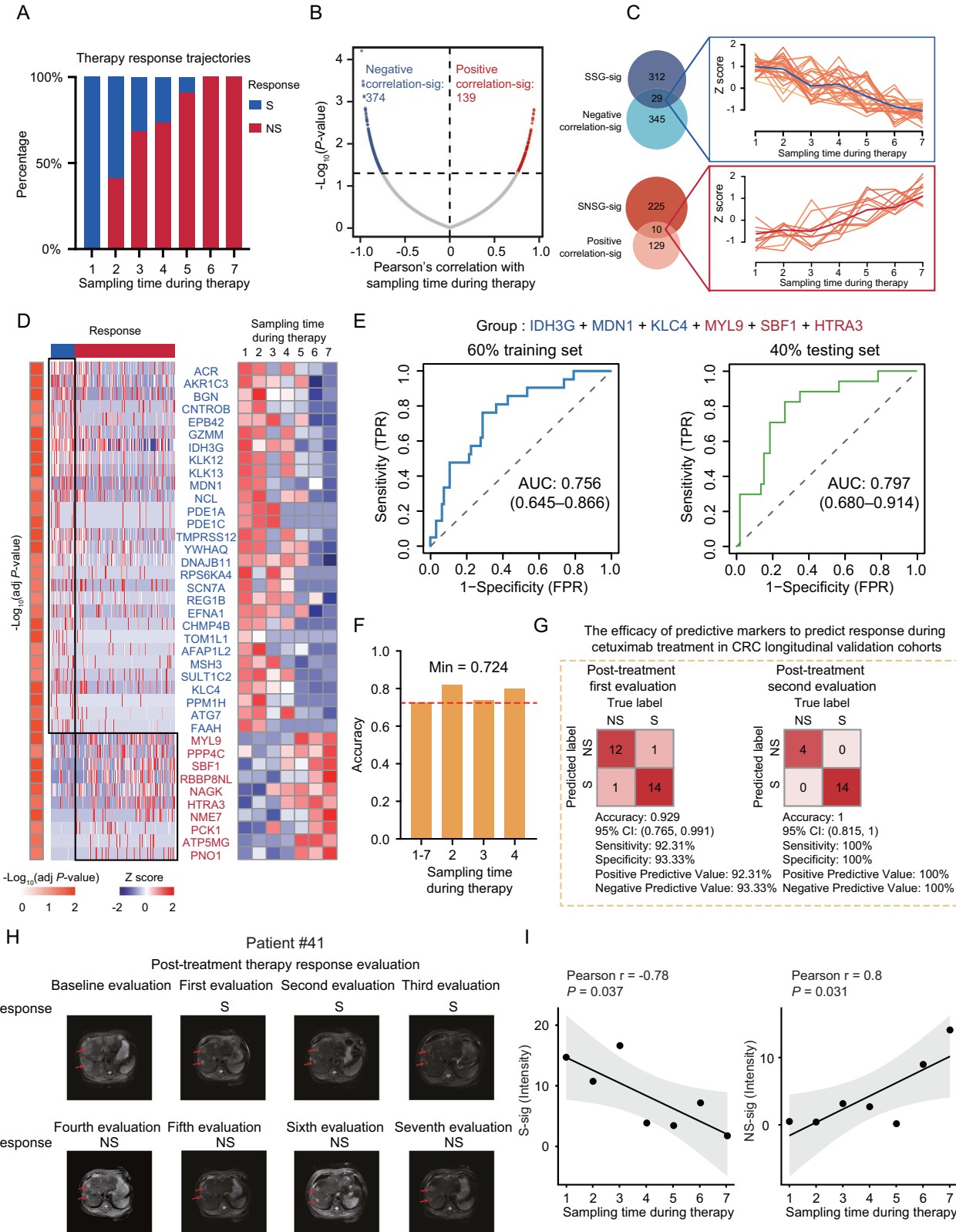

overall treatment period or the first four times of therapy evaluation, which were further validated in the independent cohort.

We further evaluate the performance of the predictive model applied in one case over seven courses treatment. In the typical case, we found the highly significantly positive correlation between protein expression of NS-sig in model and sampling time during treatment ($r = 0.8$, $P < 0.05$); while the highly significantly negative correlation

between protein expression of S-sig in model and sampling time during treatment ($r = -0.78$, $P < 0.05$), which were consistent to the MRI image related to therapy response evaluation (Fig. 5H, I). These results demonstrated the consistency between the dynamic changes of proteins included in this predictive model and clinical therapy response, which further validated the stability of the predictive model panel. Taken together, we built the predictive model, which could be applied

**Fig. 5 | The construction of predictive model for response to multiple consecutive courses of cetuximab therapy in CRC. A** The distribution of CRC patients with sensitive (S)/non-sensitive (NS) during the cetuximab treatment period. **B** Correlation of protein dynamic change abundance with sampling time during cetuximab treatment (two-sided Pearson's correlation test). **C** Upper: Venn diagram showing the overlap of negative correlation-sig and SSG-sig proteins. The K-means plot showing the dynamic trajectory of the longitudinal distribution of sensitive biomarkers. Bottom: Venn diagram showing the overlap of positive correlation-sig and SNSG-sig proteins. The K-means plot showing the dynamic trajectory of the longitudinal distribution of non-sensitive biomarkers. **D** Heatmap showing differential expression of these signature proteins in SSG and SNSG, and the dynamic changes across seven courses cetuximab treatment (two-tailed Wilcoxon rank-sum test). **E** The ROC curves of a panel of signature proteins in predicting cetuximab therapy response in 60% train set and 40% test set in stable

response cohort. **F** The accuracy of the predictive model to predict the response of cetuximab treatment at different sampling times. "1–7" represented that all sampling covered the overall treatment course were included; "2" meant that the second sampling after receiving two course treatments; "3" was defined as the third sampling after receiving three course treatments; "4" was defined as the fourth sampling after receiving four course treatments. The minimum accuracy of the predictive model was 0.724. **G** Classification error matrix using logistic regression classifier in distinguishing S and NS in the plasma longitudinal validation cohort. The number of samples identified is noted in each box. **H** The MRI image assessment at every sampling point among overall treatment process in one typical case. **I** Correlation analysis of protein expression of the panel of signature proteins in predictive model with treatment course (two-sided Pearson's correlation test). Data are presented as Pearson r with 95%CI. Source data are provided as a Source Data file.

for the continuous multiple courses of cetuximab therapy in CRC, and achieved the good performance in monitoring therapy process.

## Discussion

CRC is the main leading cause of cancer-related death. Although impressive advances, such as better screening methods and improved effective therapies, in CRC therapy have been achieved over the last years, CRC is still with undesirably high mortality, mainly due to the metastatic setting. Cetuximab, a human-mouse chimeric monoclonal antibody (IgG1 subtype), was proven to improve progression-free survival, overall survival, and response rate in several phase II and III clinical trials in combination with chemotherapy or as single agent[63,64]. Unfortunately, only a small percentage of CRC patients are sensitive to anti-EGFR therapy due to primary or innate resistance; and even those that initially respond, eventually acquire resistance and relapse under this therapy (secondary resistance). The high complexity of mechanisms of resistance to cetuximab limits the therapy efficient in CRC patients. Accordingly, a major challenge need to be addressed to optimize the efficacy of cetuximab therapy is the elucidation of the molecular basis for primary or acquired resistance to cetuximab[16], to find the biomarkers for the therapy response in the first treatment and multi-course treatment. Overall, the pressing need is to screen patients with CRC for noninvasive accurate diagnostic strategies, and screen sensitive patients to cetuximab therapy for noninvasive predictive strategies.

To achieve a comprehensive perspective of CRC patients' response to cetuximab therapy, in this study, we applied a scalable and highly reproducible MS-based proteomics workflow to 540 plasma samples from the well-characteristic CRC cohort composed of 116 CRC patients, especially with continuous multiple treatment courses of cetuximab therapy, as well as 66 healthy individuals. We firstly performed comparative proteomic analysis on plasma samples from therapy-naïve CRC patients and healthy individuals. We identified 745 DEPs up-regulated in the plasma samples from therapy-naïve CRC patients, which were mainly enriched in glycolysis/gluconeogenesis, signaling by ERBB2, MAPK activation, and extracellular matrix organization pathways. Interestingly, the alteration of proteins and pathways identified in plasma could also be reflected on the tumor tissue, suggesting a well consistency of biological change between plasma and tumor tissue, which promoted us to find biomarkers applied both in plasma and tissue for CRC diagnosis. Furthermore, after applying strict criteria in our plasma and tissue cohorts as well as the public CPTAC CRC cohort, we screened the potential candidates (including COL12A1, THBS2, S100A8, and S100A9), highly expressed in both plasma and tissue samples. The combined predictive effect of the four proteins was evaluated by ROC analysis, exhibiting a well performance in distinguishing CRC patients from healthy individuals in both plasma and tissue cohorts. The four proteins were enriched in extracellular matrix organization and interacted with other ECM proteins, showing the potential biological association. We also validated the predictive effect of the four proteins on a wider population scale of CRC patients,

but not limited to the *RAS* wild-type CRC patients or metastatic CRC patients both in plasma and tissue validation cohorts. Although the potential proteins for CRC diagnosis, were elevated both in plasma and tumor tissues of CRC patients, we couldn't completely exclude the contribution of other organs or tissues to the level of these proteins in plasma in this study. Some of these potential proteins (COL12A1, THBS2, S100A8, and S100A9) were reported to associate with other cancers and non-malignant diseases[42,65–67], which limited the clinical validity of the potential proteins in CRC. Furthermore, we validated that the four proteins could not distinguish the patients with other cancer types and the non-malignant diseases, showing the relative specificity of the biomarkers for the diagnosis of CRC. For the clinical validity of the diagnostic biomarkers for CRC, we recommended to perform a colonoscopy in a consecutive series of patients in the future.

Besides tumor diagnosis, the resistant or sensitive mechanisms for cetuximab therapy remain a major problem to be unsolved. We found these diagnostic biomarkers could not be regarded as indicators for the therapy response prediction. In this study, to search for the biomarkers for predicting the therapy response, we determined the molecular features associated with therapy response by consensus clustering analysis and identified the potential predictive biomarkers by logistic regression strategy. We identified three proteomic subtypes featured with different bioprocesses and associated with various therapeutic responses. As main findings of this study, among the three proteomic subtypes, G-III subtype was featured with autophagy mediated aggregation of CD8+ Tem, and prone to sensitive to cetuximab therapy. Consistently, as reported, tumor immune cell infiltration was associated with the cetuximab sensitivity[68]. Furthermore, the immune microenvironment estimated by xCell analysis on the plasma and tissue proteomic data revealed the association of CD8+ Tem with cetuximab sensitivity, which was further validated by IHC measurement of the representative signatures of CD8+ Tem in the tissue samples. In addition, we explored the clinical implication of CD8+ Tem, and proposed that CD8+Tem could be regarded as the potential marker for the cetuximab sensitivity, which was validated by the single-cell transcriptome data from CRC patients (GSE108989). In this study, we identified two proteins RPTOR and IMPDH2 positively associated with both expression of CD8+ Tem and better prognosis. We proposed that the potential function of RPTOR and IMPDH2 in regulation of tumor immune cell infiltration, and their association with the cetuximab sensitivity is deserved to be further explored in the future. While G-I subtype was featured with RRAS and RRAS2-mediated an axis of resistant mechanism including MAPK, RHO, and ECM pathways, and prone to be resistant to cetuximab therapy. As reported, known mechanisms of resistance to cetuximab therapy mainly included (i) upstream mutations in the extracellular domain of EGFR that directly confer resistance to antibody blockade[69], (ii) downstream pathways activated by EGFR, such as RAS-RAF-MAPK-ERK, PI3K-PTEN-AKT, and JAK/STAT pathways mainly through mutations of *KRAS, NRAS, BRAF,* and *MAP2K1*[70–72], and amplifications of *MET* and *ERBB2*[73,74]. Importantly,

in this study, we identified MAPK signaling activation mediated by RRAS and RRAS2 associated with cetuximab resistance. In addition, besides MAPK signaling, we identified the downstream ECM pathway showed a significant association with cetuximab resistance. We proposed that the potential function of RRAS and RRAS2 in regulation of ECM pathway, and the potential drugs to improve the cetuximab sensitivity were worth being developed in the future. Interestingly, the key proteins (RRAS2, MMP8, FBLN1, RPTOR, and IMPDH2) involved in these represented resistant/sensitive pathways and associated with clinical prognosis, had a good combined prediction of the initial response to first cetuximab treatment, further validating the association with cetuximab therapeutic response. These findings provided a solid reference for further investigating the sensitive or resistant mechanism, which also suggested that the non-tumor, immune related events could contribute towards responses to anti-EGFR therapy. This revealed the difference of the approach (systemic circulating markers from patient blood) versus experimental systems or tumor-only profiling where it is direct anti-tumor effect of drug without interaction with potential effector cells. In addition, although we observed the consistent alteration of the proteins and pathways uncovered both in plasma samples and tumor tissues of CRC patients, we couldn't completely exclude the contribution of other tissues or cells in plasma. It is deserved to explore the components secreted by circulating immune cells associated with the therapy response in the future.

Furthermore, the biomarkers and predictive model for the initial response to the first cetuximab therapy could not well predict the longitudinal response in the multi-course cetuximab therapy, showing the heterogeneity of different biomarkers for the initial response prediction and longitudinal response prediction respectively in the first course and multi-course cetuximab treatment. The similar phenomenon was also observed in the reported researches. For example, as Tang et al. reported in the research related to serum immune proteomics in predicting response to preoperative chemotherapy of gastric cancer, the pre-treatment serum biomarker level should have greater clinical significance than the post-treatment samples[75]. One potential reason for the attenuation of the prediction efficiency of biomarkers for initial response prediction in longitudinal response prediction was that, the biomarkers for the initial response prediction were identified and differentially expressed between the sensitive and non-sensitive groups in the therapy-naïve (pre-treatment) samples, but this difference was not significant between the sensitive and non-sensitive groups in the samples after cetuximab treatment (post-treatment). Another possibility was that the difference of the proteins or pathways related to therapy response in the pre-treatment samples possibly changed during the treatment, for instance, some sensitive features could be attenuated while some resistant features could be reserved or even enhanced during the multi-course treatment. Therefore, we focused on the longitudinal cohort to explore the longitudinal trajectories of a cohort covering seven courses cetuximab treatment, and identified proteins dynamic change associated with the treatment response to cetuximab. For more accurate to predict therapeutic response, we screened two panels of proteins significantly-regulated in SSG/SNSG and highly correlated with therapy courses. Based on these candidates, we further employed stepwise logistic regression and identified a subset of signatures that accurately discriminates SSG and SNSG, including IDH3G, MDN1, KLC4, MYL9, SBF1, and HTRA3. These signature proteins exhibited a consistent expression pattern across the multiple courses of cetuximab treatment, indicating the effectiveness of these biomarkers for predicting cetuximab response of individual patient. In addition, we also observed the variability of these proteins with longitudinal response among intra-patients, suggesting the potential heterogeneity of the biomarkers for response prediction among different individual patients. To examine the prediction effect, we further applied this predictive model in different sampling time during treatment; excitingly, we found this predictive model had stable high accuracy for predicting response in sampling time during treatment, indicating the robustness and applicable potential of the predictive model. In addition, we validated the performance of biomarkers for the longitudinal response prediction of the multi-course treatment in the independent plasma cohort. Interestingly, we also applied this predictive model in a typical case.

In conclusion, we integrated our major findings, clinical cohorts, as well as clinical characteristics, summarized the primary endpoints and second endpoints, and provided a comprehensive diagram showing the connection of each result of our study (Supplementary Fig. 6). The primary endpoints included the different biomarkers for tumor diagnosis (COL12A1, THBS2, S100A8, and S100A9), the initial response prediction of the first treatment (RRAS2, MMP8, FBLN1, RPTOR, and IMPDH2), as well as the longitudinal response prediction of the multi-course treatment (IDH3G, MDN1, KLC4, MYL9, SBF1, and HTRA3). As for the secondary endpoints, for example, (i) the over-represented pathways significantly enriched in the CRC and HC groups, and the potential biological association among the diagnostic biomarkers; (ii) the molecular features associated with different therapy response; (iii) the two protein panels positively or negatively correlated with therapy response trajectories respectively, and the molecular features associated with the stable sensitive patients and stable non-sensitive patients, we defined these findings as the secondary endpoints of the study. Overall, our study revealed the heterogeneity of different biomarkers for tumor diagnosis, the initial response prediction of the first treatment, as well as the longitudinal response prediction of the multi-course treatment. Our work emphasizes the value of longitudinal study design for biomarker discovery, which allow us to explore proteome alterations in disease progression and identify biomarkers used for dynamic monitoring in the continuous treatments. Compared to studies with single time points between CRC patients and controls that provided potential insights into potentially regulated proteins, our comparison of plasma proteomes over the cetuximab treatment course of resistance progression provided a clear set of potential biomarkers which might be used for the early intervening in the therapy process.

## Limitations of the study

In this study, we established the longitudinal plasma proteome profiling of CRC to identify the effective diagnostic markers and predictive markers for cetuximab therapy, thus contributing to the monitoring and intervening in the treatment. Among the limitations to this study, first, this study is a single-center research likely does not represent all the heterogeneous mechanisms underlying clinical resistance to cetuximab therapy. The multi-center cohorts need to be included for such a study in the future. Second, there are around 1/6 of CRC patients but not all CRC patients receiving seven course cetuximab treatment (which reflects the real clinical world), the biomarkers for the response prediction of the multi-course treatment identified in this study should be validated in the more complete multi-course longitudinal cohort. Third, although the potential biomarkers for CRC diagnosis, were elevated both in plasma and tumor tissues of CRC patients, we couldn't completely exclude the contribution of other organs or tissues to the level of these proteins in plasma, which might need to be explored through comparison among more tissues. A more rigorous multi-tissue comparison research is worth studying in the future. Fourth, it is deserved to explore the components secreted by circulating immune cells associated with the therapy response in the future.

## Methods

### Clinical sample acquisition

The studies involving human participants were reviewed and approved by the Ethics Committee of Fudan University Shanghai Cancer Center (1506147). The patients/participants provided their written informed consent to participate in this study.

The plasma samples used in this study were obtained from patients with CRC or healthy controls, from April, 2015 to February, 2021, were reviewed in the Shanghai Cancer Center, Fudan University (Shanghai, China). The study included a total of 756 plasma samples from the discovery cohort composed of 116 CRC patients undergoing anti-EGFR therapy with continuous multiple treatment courses and 66 healthy controls (HCs), the validation cohort composed of 31 CRC patients and 24 HCs, as well as the multi-cancer cohort composed of 95 patients with cancers and 20 HCs. In the plasma discovery cohort, we collected 89 pre-treatment plasma samples and 385 post-treatment plasma samples during continuous multiple treatment courses of anti-EGFR therapy from CRC patients. In the plasma validation cohort, 31 pre-treatment plasma samples and 46 post-treatment plasma samples from CRC patients were included in the plasma validation cohort. In addition, 31 tumor tissues and 27 paired NATs of CRC patients matched with the plasma samples were also included in this study. In the multi-cancer plasma independent cohort, 115 plasma samples were collected from 95 pre-treatment patients with cancers and 20 healthy controls.

For the CRC patients, there are 80 males and 36 females with a median age of 55.5 years (ranging from 21 to 76 years) in the discovery cohort, and 20 males and 11 females with a median age of 56 years (ranging from 29 to 77 years) in the independent plasma validation cohort, as well as 17 males and 14 females with a median age of 57 years (ranging from 25 to 76 years) in the independent tissue validation cohort. The anti-EGFR therapy regimen was given at standard dosing as described in previous studies, of which patients were given 500 mg/m$^2$ cetuximab once-every-2-weeks combined with FOLFOX/FOLFIRI/irinotecan[30-32]. The inclusion criteria were as follows: (i) diagnosis of CRC reviewed by three expert pathologists; (ii) presence of at least one measurable or unmeasurable but evaluable lesion (described according to Response Evaluation Criteria in Solid Tumors [RECIST] 1.1 by CT/MRI scanning and grouped into complete response (CR), partial response (PR), stable disease (SD), or progressive disease (PD)); (iii) presence of PCR-confirmed wild-type *KRAS* (exon 2/3/4), *NRAS* (exon 2/3/4), and *BRAF* (exon 15) genotypes in tumor tissue before the receipt of anti-EGFR therapy; (iv) no history of severe heart or liver disease, psychiatric disorders, hemorrhage, or perforation of the digestive tract; (v) and an ECOG performance status of 0/1 at 3 days before treatment[33]. According to NCCN guidelines, all the CRC patients receiving cetuximab therapy included in this study were left-sided *RAS* wild-type metastatic CRC[76-79]. Here, the ORR, defined as PR plus CR, was selected for the efficacy evaluation; patients with CR and PR were defined as sensitive (S) and those with SD and PD were defined as non-sensitive (NS)[46].

For the healthy controls, there are 22 males and 44 females with a median age of 62 years (ranging from 57 to 63 years) in the discovery cohort, and 12 males and 12 females with a median age of 55 years (ranging from 25 to 68 years) in the independent plasma validation cohort. The enrollment criteria for HC subjects were as follows: (i) the absence of benign or malignant tumors; (ii) a qualified physical examination finding no dysfunction of vital organs and (iii) normal renal function and without albuminuria.

For the multi-cancer plasma independent cohort, there were 95 patients with cancers and 20 healthy controls. Among them, for the patients with cancers, there were 49 males and 46 females with a median age of 63 years (ranging from 25 to 88 years); for the healthy controls, there were 12 males and 8 females with a median age of 53.5 years (ranging from 38 to 71 years). In the multi-cancer plasma independent cohort, we collected 115 plasma samples, including 95 plasma samples from treatment-naive patients with various cancer types (including colorectal cancer (CRC, $N = 20$), lung cancer (LC, $N = 15$), malignant lymphoma (ML, $N = 10$), bladder cancer (BLCA, $N = 10$), breast carcinoma (BRCA, $N = 15$), gastric cancer (GC, $N = 10$), esophageal cancer (EC, $N = 15$)), and 20 plasma samples from healthy

controls (HCs, $N = 20$). After collection, plasma and tissue samples were stored at −80 °C.

## Plasma proteome sample preparation

Blood samples were collected and centrifuged in Streck tubes. Plasma was separated by centrifugation at 1600 g at 4 °C for 10 min to remove insoluble solids and stored at −80 °C until proteomic analysis. Lysis buffer [98 μL of 50 mM NH$_4$HCO$_3$ added with 1 mM PMSF (Amresco, M145)] was mixed with the equal amounts of separated plasma sample (2 μl), and subsequently kept at 95 °C for 3 min, which was used for biomarker discovery of various body fluids in our previous study[80]. Plasma samples underwent trypsin digestion (enzyme-to-substrate ratio of 1:50 at 37 °C for 18−20 h), and the peptides were then extracted and dried (SpeedVac, Eppendorf). Peptide concentrations were measured optically at 280 nm (Nanodrop 2000, Thermo Scientific). A total of 500 ng peptide was subjected to LC-MS/MS analysis.

## Tissue proteome sample preparation

The biopsy tumor FFPE samples derived from therapy-naïve CRC patients were collected, and the tumor regions were determined by pathological examination. For clinical sample preparation, sections (10 μm thick) from FFPE blocks were macro-dissected, deparaffinized with xylene, and washed with ethanol. The ethanol was removed completely and the sections were left to air-dry. The equivalent tissues were added with the lysis buffer [0.1 M Tris-HCl (pH 8.0), 0.1 M DTT (Sigma, 43815), 1 mM PMSF (Amresco, M145)], and subsequently sonicated for 1 min (3 s on and 3 s off, amplitude 25%) on ice. The supernatants were collected, and the protein concentration was determined using the Bradford assay. The extracted tissues were then lysed with 4% sodium dodecyl sulfate and kept for 2−2.5 h at 99 °C with shaking at 1800 rpm. The solution was collected by centrifugation at 12,000 × g for 5 min. A 4-fold volume of acetone was added to the supernatant and kept in −20 °C for a minimum of 4 h. Subsequently, the acetone-precipitated proteins were washed three times with cooled acetone. Filter-aided sample preparation procedure was used for protein digestion[81]. The proteins were resuspended in 200 μL 8 M urea (pH 8.0) and loaded in 30 kD Microcon filter tubes (Sartorius) and centrifuged at 12,800 g for 20 min. The precipitate in the filter was washed three times by adding 200 μL 50 mM NH$_4$HCO$_3$. The precipitate was resuspended in 50 μL 50 mM NH$_4$HCO$_3$. Protein samples underwent trypsin digestion (enzyme-to-substrate ratio of 1:50 at 37 °C for 18−20 h) in the filter, and then were collected by centrifugation at 12,800 g for 15 min. Additional washing, twice with 200 μL of water, was essential to obtain greater yields. Finally, the centrifugate was pumped out using the AQ model Vacuum concentrator (Eppendorf, Germany).

## LC-MS/MS analysis

The acquisition of samples was randomized to avoid bias. Peptide samples were analyzed on a Q Exactive HF-X Hybrid Quadrupole-Orbitrap Mass Spectrometer (Thermo Fisher Scientific, Rockford, IL, USA) coupled with a high-performance liquid chromatography system (EASY nLC 1200, Thermo Fisher Scientific). In single-shot plasma proteome analysis, peptides, re-dissolved in Solvent A (0.1% formic acid in water), were loaded onto a 2-cm self-packed trap column (100-μm inner diameter, 3-μm ReproSil-Pur C18-AQ beads, Dr. Maisch GmbH) using Solvent A, and separated on a 150-μm-inner-diameter column with a length of 8 cm (1.9-μm ReproSil-Pur C18-AQ beads, Dr. Maisch GmbH) with 6−95% Mobile Phase B (80% ACN and 0.1% formic acid) at 600 nL/min for 8.2 min, then held constant at 95% solvent B at 800 nL/min for 4.1 min and then back to 3% B for an additional 2.7 min to equilibrate the column. The eluted peptides were ionized under 2 kV and introduced into the mass spectrometer. The MS analysis was performed in a DIA mode[82]. The DIA method consisted of MS1 Spectra full scan with m/z ranging from 300 to 1400 at a high resolution of

30,000 with an automatic gain control (AGC) target value of 3E + 06. The maximal ion injection time was 20 ms. Then, 30 DIA segments were acquired at 15,000 resolution with an AGC target 1E + 06 for maximal injection time. The setting "inject ions for all available parallelizable time" was enabled. HCD fragmentation was set to normalized collision energy of 27%. The spectra were recorded in profile mode. The default charge state for the DIA was set to 2. All data were acquired using Xcalibur software v2.2 (Thermo Fisher Scientific).

In single-shot tissue proteome analysis, peptide, re-dissolved in Solvent A (0.1% formic acid in water), were loaded onto a 2-cm self-packed trap column (100-μm inner diameter, 3-μm ReproSil-Pur C18-AQ beads, Dr. Maisch GmbH) using Solvent A, and separated on a 150-μm-inner-diameter column with a length of 15 cm (1.9-μm ReproSil-Pur C18-AQ beads, Dr. Maisch GmbH) over a 75 min gradient (Solvent A: 0.1 % formic acid in water; Solvent B: 0.1 % formic acid in 80 % ACN) at a constant flow rate of 600 nL/min (0–75 min, 0 min, 4% B; 0–10 min, 4–15% B; 10–60 min, 15–30% B; 60–69 min, 30–50% B; 69–70 min, 50–100% B; 70–75 min, 100% B). The eluted peptides were ionized under 2 kV and introduced into the mass spectrometer. MS was operated under a data-dependent acquisition (DDA) mode. For the MS1 Spectra full scan, ions with m/z ranging from 300 to 1400 were acquired by Orbitrap mass analyzer at a high resolution of 120,000 with an AGC target value of 3E + 06. The maximal ion injection time was 80 ms. MS2 Spectra acquisition was performed in top-speed mode. Precursor ions were selected and fragmented with higher energy collision dissociation with a normalized collision energy of 27%. Fragment ions were analyzed using an ion trap mass analyzer with an AGC target value of 5E + 04, with a maximal ion injection time of 20 ms. Peptides that triggered MS/MS scans were dynamically excluded from further MS/MS scans for 12 s. All data were acquired using Xcalibur software v2.2 (Thermo Fisher Scientific).

## Peptide identification and protein quantification

All data were processed using the Firmiana proteomics workstation[83], which were used in our previous studies[46,84,85]. The DIA were searched using FragPipe (v12.1) with MSFragger (2.2)[86], and the DDA files were searched using the Mascot search engine (version 2.4, Matrix Science Inc), against the NCBI human Refseq protein database. The mass tolerances were: 20 ppm for precursor and 50 mmu for product ions collected by Q Exactive HF-X. Up to two missed cleavages were allowed. The database searching considered cysteine carbamidomethylation as a fixed modification, and N-acetylation, and oxidation of methionine as variable modifications. Precursor ion score charges were limited to +2, +3, and +4. The data were also searched against a decoy database so that protein identifications were accepted at a false discovery rate (FDR) of 1%. The results of DIA data were combined into spectra libraries using SpectraST software. A total of 327 libraries were used as reference spectra libraries, which were used in our previous research related to identifying blood molecular markers for the pathophysiology and clinical progress of COVID-19[87]. The raw data were processed using DIA-NN (v1.7.10) in the "robust LC (high precision)" mode with RT-dependent median-based cross-run normalization enabled in the default settings[88]. Protein quantification was performed using the MaxLFQ algorithm[89] as implemented in the DIA-NN R-package (https://github.com/vdemichev/diann-rpackage, version 1.0, commit "eb4607a") according to the common data procession[90].

As for the DDA, the percolator was used to obtain the quality value (q value), validating the FDR (measured by the decoy hits) of every peptide-spectrum match (PSM), which was lower than 1%. All the peptides shorter than seven amino acids were removed. The cutoff ion score for peptide identification was 20. All the PSMs in all fractions were combined to comply with a stringent protein quality control strategy. We employed the parsimony principle and dynamically increased the q values of both target and decoy peptide sequences until the corresponding protein FDR was less than 1%. Finally, to reduce the false positive rate, the proteins with at least one unique peptide were selected for further investigation. The one-stop proteomic cloud platform "Firmiana" was further employed for protein quantification. Identification results and the raw data from the mzXML file were loaded. Then for each identified peptide, the extracted-ion chromatogram (XIC) was extracted by searching against the MS1 based on its identification information, and the abundance was estimated by calculating the area under the extracted XIC curve. For protein abundance calculation, the nonredundant peptide list was used to assemble proteins following the parsimony principle. The protein abundance was estimated using a traditional label-free, intensity-based absolute quantification (iBAQ) algorithm[91], which divided the protein abundance (derived from identified peptides' intensities) by the number of theoretically observable peptides. A match between runs[92] was enabled to transfer the identification between separate LC-MS/MS runs based on their accurate mass and retention time after retention time alignment. We built a dynamic regression function based on the commonly identified peptides in tumor samples. According to correlation value $R^2$, Firmiana chose linear or quadratic functions for regression to calculate the retention time (RT) of corresponding hidden peptides, and to check the existence of the XIC based on the m/z and calculated RT. Subsequently, the fraction of total (FOT), a relative quantification value was defined as a protein's iBAQ divided by the total iBAQ of all identified proteins in one experiment, and was calculated as the normalized abundance of a particular protein among experiments (Supplementary Data 2). Finally, the FOT values were further multiplied by $10^5$ for ease of presentation, and missing values were assigned $10^{-5}$ according to the previous study[36].

## Quality control of the mass spectrometry data

To quality control the MS performance, the mixture of all plasma samples was measured every twenty samples as the quality control standard. The quality control standard was digested and analyzed using the same method and conditions as the CRC samples. Pearson's correlation coefficient was calculated for all quality control runs using the R statistical analysis software v.3.5.1 (Supplementary Fig. 1A). The average correlation coefficient among the standards was 0.978, and the maximum and minimum values were 1 and 0.92, respectively. The dynamic range of protein identification of each sample was shown according to the descending sort of protein abundance with a range of 1587–2502 proteins identified in each sample (Supplementary Fig. 1C). The protein with highest intensity has the minimum rank number, representing the highest rank; the protein with lowest intensity has the maximum rank number, representing the maximum identification number in one sample.

## Differential protein and pathway analysis

Two-sided Wilcoxon rank-sum test was used to examine whether proteins were differentially expressed between CRC and HC in plasma cohort, or CRC tumors and NATs in tissue cohort. Upregulated or downregulated proteins in tumors were defined as proteins differentially expressed in CRC plasma samples compared with HC plasma samples CRC/HC > 2 or <0.5 or CRC tissue samples compared with NATs (T/NAT > 2 or <0.5) (two-sided Wilcoxon rank-sum test, Benjamini-Hochberg (BH) adjusted P-value (adj P value) < 0.05) (Supplementary Data 3). The significantly DEPs of stable sensitive group (SSG) and stable non-sensitive group (SNSG) in plasma cohort were defined as differential proteins with at least 1.5-fold change (two-sided Wilcoxon rank-sum test, adj P value < 0.05) (Supplementary Data 5); Pathway enrichment analysis was performed by CPDB based on the DEPs in each group. Pathways with an adjusted P value less than 0.05 were regarded to be significant enrichment (Supplementary Data 3 and 5).

### Identification of potential biomarkers for CRC diagnosis

The plasma and tissue samples from therapy-naïve CRC patients and healthy controls (or NATs) were included for screening potential biomarkers for CRC diagnosis. To search for potential signatures applied for plasma and tissue samples that could be used as potential biomarkers for CRC patients, we used the following criteria in this study: (1) The candidate proteins were expressed in at least 50% of the samples; (2) The candidates were significantly increased in tumor samples than normal samples or NATs (two-sided Wilcoxon rank-sum test, adj $P$ value < 0.01); (3) The candidates were identified with at least 2-fold increase in CRC samples than normal samples. The same screening criteria were adopted in our tissue validation cohort and the CPTAC CRC cohort[38]. Ultimately, we obtained four proteins (including COL12A1, THBS2, S100A8, and S100A9) for distinguishing CRC patients from healthy controls. The predictive effect of the four proteins combination was verified by ROC analysis using (pROC R package version 1.16.2 and Caret R package version 6.0–86), in which sensitivity, specificity, accuracy, and AUC were used to determine predictive values. The 10-fold cross validation was used, and samples were partitioned based on sample types; among them, 60% of samples were used as a training set, and the remaining 40% represented the independent testing set.

### Protein–protein interaction (PPI) network

Protein–protein interaction (PPI) network was constructed to reveal the correlations of proteins using the Search Tool for the Retrieval of Interacting Genes (STRING; http://string-db.org) (version 11.5) online database[93,94], which is a database of known and predicted protein-protein interactions. The PPI network identified by String and Cytoscape visualization software.

### The potential molecular features and biomarkers for the initial therapy response

The protein expression matrix of the plasma samples from therapy-naïve CRC patients was used to identify the proteomic subtypes using the consensus clustering method implemented in the R package ConsensusClusterPlus v.3.8[95,96]. The top 1500 proteins with the highest median absolute deviation were subjected to ConsensusClusterPlus in R v.3.5.1 for unsupervised consensus clustering. The cluster analysis was performed with the following setting: maxK = 10, reps = 1000, pItem = 0.8, pFeature = 1, clusterAlg = "hc", distance = "pearson" for the clustering runs. A preferred cluster result was selected by considering the profiles of the consensus cumulative distribution function (CDF) and delta area under the CDF curve for clustering solutions between 2 and 10 clusters. As shown in Supplementary Fig. 3, the rank survey profiles of the consensus CDF and the delta area under the CDF curve, along with the consensus membership heat maps, indicated a three-subtype solution for 89 samples of pre-treatment CRC using the proteomic data. This showed clear separation and the significant therapy response among 3 subtypes. To generate the abundance heatmap, the CRC samples in each subtype were rearranged from G-I to G-III, using the signature protein abundance matrix enriched in the signature pathways for each subtype. The signature proteins of each subtype were defined with significantly differential expression (fold change >1.5, adj $P$ value < 0.05, two-sided Wilcoxon rank-sum test) when compared with other subtypes (Supplementary Data 4). ConsensusPathDB (CPDB) molecular interaction data obtained from 31 different public repositories[97], and determined the dominant bioprocesses of each subtype. The adj $P$ value < 0.05 is considered statistically significant of the pathway enrichment. The key regulators involved in the overrepresented pathways associated with therapy response were identified. Among these proteins, a group of proteins with significant (positively or negatively) correlation with clinical prognosis were regarded as the potential biomarkers (including RRAS2, MMP8, FBLN1, RPTOR, and IMPDH2). The predictive effect of the five biomarkers combination was verified by

ROC analysis using (pROC R package version 1.16.2 and Caret R package version 6.0–86), in which sensitivity, specificity, accuracy, and AUC were used to determine predictive values.

### Correlation between proteomic subtype and clinical characteristics

To explore the association of clinical baseline characteristics (including age, gender, degree of tumor differentiation, and ECOG performance-status score) and other biochemical indicators (such as lactate dehydrogenase (LDH), WBC, lymphocyte number (LYMPHN), hemoglobin (HB) and platelet count) with proteomic subtypes, two-sided Fisher's exact test was performed on categorical variables and one way-ANOVA analysis was applied for comparison for the continuous variables (GraphPad Prism 8 software). $P$ values less than 0.05 were considered as significantly different (Table 2).

### The gene set enrichment analysis (GSEA)

GSEA is a computational method to determine whether a priori defined set of genes has statistical significance and concordant differences in two biological states[39], which was applied for pathway analysis in many researches[98–100]. Sample grouped according to different groups (the proteomic subtypes, or S/NS groups or RRAS high/low expression groups) were subjected to GSEA based on the proteomic data using the clusterProfiler R package (v3.18.1)[101]. Molecular Signatures Database (MSigDB) of hallmark gene sets (H), curated gene sets (H2) and GO gene sets (C5) were used for enrichment analysis. An FDR value of 0.05 was used as a cutoff. The enrichment score (ES) in GSEA was calculated by first ranking the proteins from the most to least significant, the entire ranked list was then used to assess how the proteins of each gene set were distributed across the ranked list.

### The single sample gene set enrichment analysis (ssGSEA)

All scores were inferred by single sample gene set enrichment analysis (ssGSEA) method from the GSVA R package based on the protein expression matrix. The gene set (c2.all.v7.4.symbols) of Molecular Signature Databse(MSigDB) was used to ssGSEA[40]. The parameters: min.sz = 10, max.sz = 300 were set and other parameters were used default. The ssGSEA scores of pathways in each sample were obtained for the further correlation analysis among pathways and comparison between two groups.

### Immune cell type composition

The cell type enrichment score assessed with statistical significance was calculated and used to infer the relative abundance of different cell types in the tumor microenvironment in the online enrichment streamline at https://xcell.ucsf.edu/, which was also applied in many transcriptomic and proteomic researches[47,102,103]. The abundance of 64 different cell types were computed via xCell based on proteomic profiles[104]. The Supplementary Data 4 contains the final score computed by xCell of different cell types. To identified significantly differential cell types in G-III subtype, two-sided Wilcoxon rank-sum test with Benjamini-Hochberg (BH) adjust was used to compare the difference between G-III subtype and other subtypes (G-I and G-II), cell types with fold change (G-III/others) >2 and adj $P$ < 0.05 as significantly different (Supplementary Data 4).The tumor size evaluated by CT/MRI was used for the correlation analysis with CD8+Tem xCell score. The single-cell transcriptome data (GSE108989)[49] was included for the association of CD8+Tem with tumor size.

### Identifying biomarkers for the longitudinal response prediction of the multi-course cetuximab treatment

To explore the plasma biomarkers for efficient monitoring during the continuous multiple courses to cetuximab treatment, we used the following these criteria in this study: (1) The plasma proteins were significantly altered during multiple cetuximab treatment courses

screened by significant positive or negative correlation, defined as positive correlation-sig or negative correlation-sig; (2) The candidates were significantly differential expression in the stable sensitive and non-sensitive group (fold changes> 1.5 or <0.67; two-sided Wilcoxon rank-sum test, adj *P* value < 0.05), defined as SSG-sig and SNSG-sig; (3) The overlapped proteins of SSG-sig and negative correlation-sig, as well as SNSG-sig and positive correlation-sig were regarded as the candidate biomarkers for the further stepwise logistic regression. Samples was randomly divided into 60% of individuals (the training set) and the remaining 40% (the testing set)[105]. The backward stepwise method was utilized to feature selection on the training set. The 10-fold cross validation was used. Moreover, the value of this model was verified using ROC analysis. Sensitivity, specificity, accuracy, and AUC were used to determine predictive values.

### Targeted PRM analysis

Using the library search results, a set of target peptides that unique to RPTOR and IMPDH2 proteins were selected and PRM method was designed. Besides, house-keeping proteins, such as VCP, RPLP0, PSMB4, were also included for the reference. Equal amount of plasma from each sample was digested as described in the part of profiling preparation. Peptide samples were injected into the Q Exactive HF-X Hybrid Quadrupole-Orbitrap Mass Spectrometer (Thermo Scientific) operating in PRM mode with quadrupole isolation and HCD fragmentation. The full MS mode was measured at resolution 45,000 with AGC target value of 3E6 and maximum IT of 20 ms, with scanning range of 150 to 2000 m/z. Target ions were submitted to MS/MS in the HCD cell (1.6 m/z isolation width, 27% normalized collision energy). All the PRM events were performed after MS1 scanning, at resolution 15,000 with AGC target value of 1E6 and maximum IT of 25 ms. Separation was achieved on a 150-μm-inner-diameter column with a length of 15 cm (1.9-μm ReproSil-Pur C18-AQ beads, Dr. Maisch GmbH) in an Easy 1200 nLC HPLC system (Thermo Scientific). Solvent A was 0.1 formic acid in water and solvent B was 0.1% formic acid, 80% ACN in water. Peptides were separated at 600 or 800 nL/min across a gradient as following over 15 min: 0–15 min, 0 min, 6% B, 600 nL/min; 0–6 min, 30% B, 600 nL/min; 6–8.2 min, 50% B, 600 nL/min; 8.2–9.2 min, 95% B, 800 nL/min; 9.2–12.3 min, 95% B, 800 nL/min; 12.3–13.3 min, 3% B, 800 nL/min; 13.3–15 min, 3% B, 800 nL/min.

Raw data was searched by Skyline-daily (4.2.1.19004, University of Washington, USA), as described in our previous study[46]. The proteins were quantified with the fragment total area reported by Skyline-daily. We selected peptides and tested their stability of signal and shape of peaks in the pool sample for final quantification, and referred to the ranking offered by skyline. The list of targeted peptides and the expression matrix were included in the Supplementary Data 4.

### Survival analysis

Overall survival (OS) was used as primary endpoint. Hazard ratio (HR) and *P* values were calculated by two-sided log rank test, and *P* values < 0.05 were considered as significantly different. Survminer (version 0.2.4, R package) with maximally selected rank statistics was used to determine the optimal cut-off point of a given protein for the following calculation including Kaplan–Meier analysis, log rank test, according to the previous studies[35,106]. OS curves were based on the optimal cut-off point.

### Immunohistochemistry staining and evaluation

A standard IHC protocol was followed to stain the tumor tissue samples of the CD8+ Tem markers by using the rabbit monoclonal antibody against CD44 (1:200, Signalway Antibody, catalog No: 48911-1), and the rabbit polyclonal antibody against GZMK (1:300, Signalway Antibody, catalog No: 40985-1). IHC evaluation was analyzed using an IHC profiler compatible plugin with integrated options for the quantitative analysis of digital IHC images stained for cytoplasmic or nuclear proteins. Moreover, the intensity of the cytoplasmic staining and the percentage of positively stained tumor cells were also scored numerically.

### Statistics and reproducibility

Standard statistical tests were used to analyze the clinical data, including but not limited to Student's *t* test, Wilcoxon rank-sum test, Fisher's exact test, Pearson's correlation test, Spearman correlation test, log-rank test, one-way ANOVA. Unless otherwise specified, all statistical tests were two-sided. The two-sided Fisher's exact test was performed on categorical variables (including age, gender, degree of tumor differentiation, and ECOG performance-status score), and one way-ANOVA analysis was applied for comparison for the continuous variables (including lactate dehydrogenase (LDH), WBC, lymphocyte number (LYMPHN), hemoglobin (HB) and platelet count) (GraphPad Prism 8 software). *P* values less than 0.05 were considered as significantly different. The Wilcoxon rank-sum test was used to examine whether proteins were differentially expressed between Pre-treatment CRC ($N = 89$) and HC ($N = 66$), Tumor tissues (T, $N = 31$) and normal-adjacent tissues (NAT, $N = 27$), each proteomic subtype and other subtypes (G-I ($N = 24$), G-II ($N = 34$), G-II ($N = 31$)),SSG ($N = 38$) and SNSG ($N = 153$). The Student's t test was used to examined whether the cell types were differentially expressed each proteomic subtype and other subtypes (G-I ($N = 24$), G-II ($N = 34$), G-II ($N = 31$)). All statistical tests were two-sided, and statistical significance was considered when *P* value < 0.05. To account for multiple testing, the *P* values were adjusted using the Benjamini-Hochberg FDR correction. Kaplan–Meier plots with log-rank test were used to describe survival analysis. All the analyses of clinical data were performed in R (v3.5.1) and GraphPad Prism 8 software. For functional experiments, each was repeated at least three times independently, and results were expressed as mean ± SD. Statistical analysis was performed using GraphPad Prism 8 software. The *P* values less than 0.05, 0.01, 0.001, 0.0001 were marked with *, **, ***, ****, respectively. All the statistical analysis had been checked by two statisticians.

### Reporting summary

Further information on research design is available in the Nature Portfolio Reporting Summary linked to this article.

## Data availability

The raw mass spectrometry (MS) proteomics data and parallel reaction monitoring (PRM)-MS proteomics data generated in this study have been deposited in the deposited in the ProteomeXchange Consortium (dataset identifier: PXD047207) via the iProX partner repository (https://www.iprox.cn/)[107] under the project ID IPX0005221000. The Human Protein Atlas (HPA) IHC Staining Data and the list of protein classes could be accessed at https://www.proteinatlas.org/. Molecular Signatures Database (MSigDB) could be accessed at https://www.gsea-msigdb.org/gsea/msigdb. TIMER2.0 database could be accessed at http://timer.cistrome.org/. The ConsensusPathDB (CPDB) molecular interaction data could be accessed at http://www.consensuspathdb.org/. The STRING database could be accessed at https://cn.string-db.org/. The public datasets related to ulcerative colitis (an inflammatory bowel disease) and infection disease (such as SARS-CoV-2 infection) could be obtained from Gene Expression Omnibus (GEO, https://www.ncbi.nlm.nih.gov/geo/) with the accession number GSE11223 and GSE207015, respectively. The remaining data are available within the Article and Supplementary Information. NCBI human Refseq protein database could be accessed at https://www.ncbi.nlm.nih.gov/refseq/. Source data are provided with this paper.

## Code availability

This study did not generate custom computer code. No special code was used in this study. The related R scripts used for statistical analyses

in this study have been publicly available on GitHub repository: https://github.com/buranoyanlee/Prediction[108] and Zenodo repository: https://zenodo.org/records/10200747[109].

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

## Acknowledgements

This work is supported by the National Key Research and Development Program of China (2022YFA1303200 [C.D.] and 2022YFA1303201 [C.D.]), National Natural Science Foundation of China (32330062 [C.D.] and 31972933 [C.D.]), sponsored by Program of Shanghai Academic/Technology Research Leader (22XD1420100 [C.D.]), the Major Project of Special Development Funds of Zhangjiang National Independent Innovation Demonstration Zone (ZJ2019-ZD-004 [C.D.]), the Science and Technology Commission of Shanghai Municipality (2017SHZDZX01 [C.D.]), Natural Science Foundation of Shanghai (22ZR1413500, [Z.C.]), the Young Scientists Fund of the National Natural Science Foundation of China 32301236 [Y.L.], 32201215 [J.F.] and 32201212 [Y.W.], Shanghai Sailing Program (23YF1402800 [Y.L.] and 22YF1403100 [J.F.]), and the Fudan original research personalized support project.

## Author contributions

Y.L., B.W., Z.C. and C.D. conceived the work and designed the experiments; W.Y. and J.Z. collected the plasma samples; Y.L., B.W. and F.M. performed the experiments and acquired the MS data; J.F., K.L. S.T., Y.W and Z.Q. provided expertize and technical support; Y.L., B.W. and F.M. analyzed the data; C.D. and Z.C. supervised the project and interpreted results; Y.L., B.W. and C.D. wrote the original manuscript.

## Competing interests

The authors declare no competing interests.
