## [Peer Review File · Nature Communications]

REVIEWER COMMENTS

Reviewer #1 (Remarks to the Author): expertise in proteomics bioinformatics analysis

Authors have a track record of large proteomic studies. Here, they performed longitudinal and deep proteomic profiling of a large number of plasma samples originating from CRC patients undergoing cetuximab therapy and many healthy controls. The work is important and there are interesting findings, but the reviewer has some issue with how the results are being described and interpreted. The reviewer requests some changes and clarifications before this work is suitable for publication.

- Line 182, what is the justification for comparing plasma-based protein biomarkers to proteins that were differentially expressed between normal and tumor tissue? These are not so easily related. The resulting proteins (COL12A1, THBS2, S100A8, and S100A9) could be coming from any number of sources. For example, S100 family members seem to be almost universally expressed in tissues. The authors seem to be suggesting, without evidence, that proteins secreted by the tumor are making it into plasma. What is a potential mechanism for these proteins making it from tumor to blood? Are the tumors inducing systemic changes that cause these proteins to appear in the results? Are there independent mechanisms that could be giving rise to the expression of these proteins in tumor and their appearance/secretion in plasma? This seems like a random assortment of proteins. Are these proteins related in any way (besides the two S100 family members)? Is there a plausible biological reason for the appearance of these proteins together?

- Line 205, 'To further determine whether these four proteins could effectively diagnosis patients with CRC...'? If the requirement for the proteins to be differentially expressed in tumor were dropped, would the resulting protein list (or maybe the top 10 proteins from this list) perform better than these 4 proteins?

- Line 222, the results in this section 'Proteomic subtyping of the CRC cohort...' and also in other parts the manuscript describe differences in the context of signaling, immune infiltration etc, but a majority of these findings are from plasma samples. How much of these findings are a reflection of what is going on in the tumor? How does this compare to what is going on systemically in the patient? How does this compare to what is secreted by circulating immune cells? These are very important points that are essentially ignored. Unless authors can prove the source (e.g. cell, tissue) of the changes in the proteins and pathways they describe, then the results need to be reworked, and it should be clearly stated that the origins/causes of these changes is systemic/unclear.

- Line 247, why is xCell being applied to proteomics data? Please justify the use of xCell applied to proteomics data. Deconvolution methods are generally developed with the high coverage of transcriptomics in mind. Although the authors are describing good results with a reasonable number of gene products for a proteomics study, the authors have a relatively small number of gene products compared to microarray/RNA-seq. Further, this seems to be plasma. Are the authors assuming there are enough secreted proteins found in plasma to do a good job at approximating circulating immune subpopulations? This appears to be a significant limitation of these analyses. Applying xCell to proteins found in plasma seems to be a very big overreach and the subsequent results are likely over-interpreted. The authors appear to provide scores for xCell in the supplemental data but not p-values. Were any of the FDR-adjusted p-values for these score estimates from xCell still significant? If the xCell results are to be included, then authors need to show xCell does a good job at estimating immune subpopulations from plasma. Ideally, some independent experiments to validate these immune subpopulations would be provided.

- Line 285, please clarify the sentence '...high expression of RRAS and RRAS2 were prone to resistant significantly increased (FC > 2, 286 P < 0.05) (Figure 4D).

- ssGSEA is used throughout the manuscript, and it seems to be used for comparing two groups. Why is traditional GSEA not used for two group comparisons? Were all ssGSEA scores used, or was there some cutoff based on the p-values for the score? How is differential expression done between ssGSEA scores? Please clarify these points in the methods.

- P-values are provided throughout the manuscript, but the methods suggest these are adjusted p-values. Pathway p-values appear to not be adjusted. Please distinguish between adjusted p-values (such as by indicating "padj" or similar in the text) and raw p-values.

- The biomarker analysis that identified COL12A1, THBS2, S100A8, and S100A9 seems fairly independent from the consensus clustering/subtyping analysis, and both of these analyses seem fairly independent from the longitudinal analysis in figure 5. Authors should do a better job at relating these findings to each other. Was there any common biology between these results? Did these results inform each other? Could the results be described in a more holistic way?

Reviewer #2 (Remarks to the Author): expertise in mass-spectrometry based biomarker analysis

In this manuscript, Li et al. performed an impressive plasma proteomics dataset of 580 plasma samples from the discovery cohort composed of 116 CRC patients undergoing anti-EGFR therapy with continuous multiple treatment courses and 66 healthy controls (HCs), as well as the validation cohort composed of 16 CRC patients and 24 HCs. The tumour-specific biomarkers were inferred and validated by other public cohorts, which also exhibited a well performance in the independent cohort. They identified three proteomic-based subtypes characterized with different clinical therapy responses and molecular features. In the longitudinal trajectories of a cohort covering multi-course cetuximab treatment, the authors explored proteome alterations in disease progression and identify biomarkers used for dynamic monitoring in the continuous treatments. The biomarkers were then validated by PRM.

The authors have performed a large amount of informatics analyses, exploring novel insights into the biology and identifying novel biomarkers for predicting cetuximab treatment response of CRC patients. Moreover, the dataset itself, which is deposited in public repositories, will serve as a rich resource for the CRC research community. Overall, it is a good study. However, there are some questions that require clarification before the manuscript would be acceptable for publication.

Major comments:

1. The authors screened four proteins in Figure 2A for CRC diagnosis in Fudan cohort and other cohorts, and validated them in an independent cohort. Then, whether these proteins also associated with CRC cetuximab therapy or only for CRC diagnosis?
2. This study identified proteomic subtypes featured with distinct biological features and therapy response. This significant association of proteomic subtypes and therapy response were shown in Figure 3B. What about other clinical variables? The authors should also make a statistic analysis of other clinical variables, and included these results in the supplementary figure.
3. Proteomic subtypes identified G-I as non-sensitive subtype and G-III as sensitive subtype, exploring the potential biological alteration. As for the G-II subtype, the pathway enrichment result and association with response should be further investigated.
4. The process for biomarker identification and predictive modeling should be provided detailed in the Method section.
5. The authors mentioned "The overrepresented proteins of S and NS of cohort were defined as ...($NS/S > 2$ or < 0.5); the significantly differentially expressed proteins was defined as ...2-fold change and Wilcoxon rank-sum test with a Benjamini-Hochberg (BH) adjusted p value cutoff (BH p value < 0.05) in the Method section. However, the BH adjusted p values were not provided in the Table S5. In addition, in the differential analysis between SSG and SNSG groups, the authors defined the overrepresented proteins (ORPs) with 2-fold changes with $P < 0.05$ (Line: 336-338); however, Table S5 included the DEPs. As for the differential analysis between CRC patients and healthy controls, the authors also mentioned the ORPs were proteins with 2-fold changes with $P < 0.05$ (Line: 172-176). Overall, the authors should clarify "ORPs" and "DEPs" definition, and use uniform description in the manuscript.
6. This study included a total of 580 plasma samples from the discovery cohort (composed of the well-characteristic CRC cohort composed of 116 CRC patients and 66 healthy controls) and validation cohort. The key biomarkers for diagnosis of CRC and response prediction of cetuximab therapy have been validated by various proteomic approaches, including data-independent acquisition (DIA) and parallel reaction monitoring (PRM), in the public datasets and independent cohort. Then, if any limitations in this study, the authors should have a discussion on study limitations. This is needed.
7. Hazard ratios should be included for all KM plots.
8. The targeted peptides of signature proteins were selected for targeted PRM analysis. However, the authors didn't provide the list of targeted peptides included in this study.

Minor comments:

9. The therapy response information (S and NS) is very important, and associated with many results in this study. The S/NS information corresponding to each sampling time during the continuous treatment were provided in Table S5. However, the authors didn't provide the initial therapy response information for individual patients in Table S1. The authors should provide the information, which could be enable effective reuse of the dataset in future researches.
10. In the figures (e.g. Figure 2B, Figure 3B/E/G, Figure 4C/D), the statistical analysis marked the significance with * or p values, please provide the exact p values uniformly.

11. Minor grammar problems should be corrected. For example, Line: 265: "..., might resulting in an improved response to cetuximab therapy".
12. Line 548-550: the authors described "The dynamic range of protein identification of each sample was shown according to the descending sort of protein abundance with a range of 1,587–2,502 proteins identified in each sample.", the corresponding figure should be cited.
13. The heatmap of Figure 2B shows z-score transformation of protein expression? The relative abundance should be annotated in figure legend.
14. In the legend of Figure 3A, the numbers of each subtypes should be listed.
15. The description of Figure 5G is rough. The authors should provide more details corresponding to Figure 5G.
16. All box-plot elements (center line, limits, whiskers, points) should be defined in the legends accompanied by precise n numbers, e.g. Figure 1D, Figure 3E/3G/3K, Figure 4C/4D.

Reviewer #3 (Remarks to the Author): clinical expertise in colorectal cancer and cetuximab treatment

The topic of blood-based methods, including proteomics, to screen for colorectal cancer is very interesting and the authors are commended for looking at this topic.

However, the article is confusing and requires better focus. A considerable part of the paper addresses how the molecules identified can be used to distinguish colon cancer from normal healthy issues with screening potential but this is not reflected in the title, aims and objectives – the title instead leading the reader to believe that the biomarkers under question (and assuming prior validation) are primarily for cetuximab resistance prediction. Clearly presenting the primary and/or secondary endpoints of the study would guide the reader. A better description of the selection of the cohorts (both patient and normal) would help discern the presence of selection bias. Furthermore, it is not very clear which samples, at which time points have been used at every step of the analysis.

In the discussion, no limitations have been presented. This makes it very difficult to properly judge the significance of the findings which could be overstated. The validation cohort is rather small, this would make the results weaker and the conclusions less robust. Overall, better structuring of the paper, focusing the work, having a study title that reflects the research, clearly presenting endpoints, better description of the methodology which should be robust in identifying an accepted predictive biomarker development pathway and discussing potential limitations would benefit this paper.

RESPONSE TO REVIEWERS' COMMENTS

Reviewer #1 (Remarks to the Author): expertise in proteomics bioinformatics analysis

Authors have a track record of large proteomic studies. Here, they performed longitudinal and deep proteomic profiling of a large number of plasma samples originating from CRC patients undergoing cetuximab therapy and many healthy controls. The work is important and there are interesting findings, but the reviewer has some issue with how the results are being described and interpreted. The reviewer requests some changes and clarifications before this work is suitable for publication.

Response:

We appreciate the reviewer for the constructive and insightful comments, which help to improve the quality of this manuscript. Here, we summarized the reviewer's comments as following: (1) about the diagnostic biomarkers for CRC; (2) about the justification for xCell used in proteomics data and the validation of related findings; (3) about GSEA analysis and ssGSEA analysis; (4) about distinguish between the raw p-values and adjusted p-values; (5) about the integrative association of findings obtained from different sections. The point-to-point responses are as follows.

Q1. Line 182, what is the justification for comparing plasma-based protein biomarkers to proteins that were differentially expressed between normal and tumor tissue? These are not so easily related. The resulting proteins (COL12A1, THBS2, S100A8, and S100A9) could be coming from any number of sources. For example, S100 family members seem to be almost universally expressed in tissues. The authors seem to be suggesting, without evidence, that proteins secreted by the tumor are making it into plasma. What is a potential mechanism for these proteins making it from tumor to blood? Are the tumors inducing systemic changes that cause these proteins to appear in the results? Are there independent mechanisms that could be giving rise to the expression of these proteins in tumor and their appearance/secretion in plasma? This seems like a random

assortment of proteins. Are these proteins related in any way (besides the two S100 family members)? Is there a plausible biological reason for the appearance of these proteins together?

Response to Q1:

We sincerely thank the reviewer for the constructive comments. In this study, we included 116 CRC patients and 66 healthy controls in the discovery cohort for plasma proteome profiling, established the differential plasma proteomic patterns compared with healthy controls, and identified a group of diagnostic candidates for CRC patients. In this part, we apologize for not clearly illustrating (1) the justification for screening protein biomarkers for CRC diagnosis identified both in plasma and tumor tissues, and (2) the association of the plasma biomarkers with tumor tissues, as well as (3) the potential biological function and association of the four biomarkers. To answer the reviewer's questions clearly, we divided the response into three parts as follows.

(1) The justification for screening protein biomarkers for CRC diagnosis identified both in plasma and tumor tissues

We firstly apologize for not explaining it clearly in the previous version. This study applied mass spectrometry (MS)-based plasma proteomics, which has been successfully applied to disease diagnosis (*Cell Syst.*, 2018, PMID: 30528273; *EMBO J.*, 2020, PMID: 33140861; *Cell Rep Med.* 2022, PMID: 35732154), for the biomarker discovery of CRC diagnosis in the plasma samples. To search for plasma proteins that could be used as diagnostic biomarkers for CRC patients, we used the following three criteria in this study: 1) The candidate proteins were expressed in at least 50% of the samples (1,359 proteins); 2) The candidates were significantly increased in tumor samples than normal samples (two-sided Wilcoxon rank-sum test, adj *P*-value < 0.01; 234 proteins); 3) The candidates were identified with at least 2-fold increase in CRC samples than normal samples. As a result, 148 proteins were screened, and significantly and stably overexpressed in the plasma of CRC patients.

In the previous version, to further refine a group of proteins for CRC diagnosis, we searched for the public proteomic dataset related to CRC (*Cell.*, 2019, PMID: 31031003), which was also

included for the further biomarker screening. This public study included the 96 tumor and NAT pairs of CRC patients, and identified a total of 8,067 proteins; finally defined 31 proteins elevated by more than 2-fold in tumors as colon cancer-associated proteins after the same screening criteria. Based on this, together with our plasma screening results, we ultimately obtained four proteins, COL12A1, THBS2, S100A8, and S100A9, could be used as diagnostic markers for CRC both applied in plasma samples and tumor tissues. In addition, we also surveyed similar studies of which diverse clinical sample types, especially plasma and tumor tissues, were both included for biomarker discovery (*Nat Med.*, 2022, PMID: 35654907; *Cell.*, 2020, PMID: 32795414; *Respir Res.*, 2021, PMID: 34689792); these studies demonstrated the comprehensive signature identified both in tissue and plasma could be utilized for better performance in clinical assessment of disease (for example diagnostic and prognostic value in the clinic) compared with either sample type.

Therefore, compared with the initially screened 148 proteins highly expressed in plasma of CRC patients, we proposed that the refined four proteins validated both in plasma and tumor tissues could reach high performance using fewer biomarkers as predictive combination. Further analysis validated that the four proteins combined as a predictive model exactly showed high accuracy in the discovery cohort, which also achieved good performance in an independent validation cohort. In the revision, to further validate the prediction effect of the four proteins composed model, we included more CRC patients in the independent plasma validation cohort composed of 31 CRC patients and 24 healthy controls for the proteomic measurement. Consistently, the differential proteomic analysis revealed that COL12A1 (FC = 4.25, adj *P*-value = 0.017), THBS2 (FC = 2.30, adj *P*-value = 0.034), S100A8 (FC = 2.34, adj *P*-value = 0.013), and S100A9 (FC = 28.34, adj *P*-value = 1.2E-7) were significantly up-regulated in the plasma samples from CRC patients in the plasma validation cohort (**Figure RL1A, see also Figure S2F in the revision**). We further performed the receiver operating characteristic (ROC) analysis to evaluate the predictive effect. The ROC analysis revealed the four proteins combined prediction achieved good performance with an AUC of 0.952 in the plasma validation cohort (**Figure RL1B, see also Figures 2E and 2G in the revision**). Overall,

the refined four proteins validated both in plasma and tumor tissues could well distinguish the CRC patients from healthy controls in discovery cohort and validation cohort.

Figure RL1. The validation of diagnostic model in the plasma validation cohort. (A) The boxplot showing the differential expression of the four diagnostic biomarkers in the CRC and HC groups in the plasma validation cohort (two-sided Wilcoxon rank-sum test, adj P -value < 0.05). Boxplots show median (central line), upper and lower quartiles (box limits), 1.5×interquartile range (whiskers). (B) The receiver operating characteristic (ROC) curves of diagnostic biomarkers to distinguish CRC patients from healthy controls in the plasma discovery cohort and validation cohort.

(2) The association of the plasma biomarkers with tumor tissues

As the reviewer mentioned, the plasma biomarkers could be coming from any number of sources. We agree with the reviewer that plasma proteins might be the proteins from tumor tissue or any other tissues that might be released into extracellular fluids through secretion or cell and tissue breakdown. In our study, we refined a group of biomarkers (COL12A1, THBS2, S100A8, and S100A9) for CRC diagnosis after strict screening criteria.

Firstly, to further reveal the association of the expression of these biomarkers in plasma and tumor tissues, in the revision, we reviewed all the archival formalin-fixed paraffin-embedded (FFPE) tissues from the therapy-naïve CRC patients included in this study. Finally, we collected 31 tumor tissues and 27 paired normal-adjacent tissues (NATs) of CRC patients matched with the plasma samples for mass spectrometry (MS)-based proteomic profiling. Proteomic analysis of tumor tissues identified a total of 9,258 proteins, of which 6,927 proteins were detected in

plasma (**Figures RL2A and 2B, see also Figures S1D and S1E in the revision**). The comparative proteomic analysis demonstrated that 2,573 proteins were significantly elevated in tumor tissues; while 251 proteins were down-regulated in tumor tissues (fold change (Tumor vs NAT) >2, adj *P*-value < 0.05) (**Figure RL2C, see also Figure S1F in the revision**). In the previous version, based on the plasma proteome profiling, the comparative analysis identified 745 proteins up-regulated in the CRC patients; among of these proteins, 235 (31.54%) proteins were up-regulated in tumor tissues from CRC patients (**Figure RL2D, see also Figure S1G in the revision**). Interestingly, this result showed a relatively high overlapped proportion of the differential proteins up-regulated in both the plasma samples and tumor tissue samples of the CRC patients. Evidently, COL12A1 (FC = 4.29, adj *P*-value = 8.8E-4), THBS2 (FC = 3.43, adj *P*-value = 1.6E-4), S100A8 (FC = 3.40, adj *P*-value = 8.7E-5), and S100A9 (FC = 5.10, adj *P*-value = 1.2E-4) showed a significant increase in tumor tissues (**Figure RL2E, see also Figure S2E in the revision**). Furthermore, we validated the diagnostic efficiency of the four proteins in the tissue cohort of CRC patients. The receiver operating characteristic (ROC) analysis revealed the four proteins combined prediction achieved good performance with an AUC of 0.945 in the tumor tissues (**Figure RL2F, see also Figure 2G in the revision**). Therefore, these results demonstrated the predictive model composed of four biomarkers, including COL12A1, THBS2, S100A8 and S100A9, could achieve a high prediction in both plasma and tumor tissues of CRC patients. Finally, we appreciate the reviewer for the constructive suggestions. We never know the relative consistency of the CRC plasma samples with tumor tissue samples, if we didn't include the tumor tissue samples matched with plasma from CRC patients for a proteomic comparison which suggested by the reviewer. The matched comparison of plasma samples and tumor tissue samples allow us for a better understanding of the association of plasma biomarkers with tumor tissue.

Secondly, we also reviewed the literatures of the last decade to understand the association of the four plasma biomarkers with tumor tissues. As a review published in Nature Reviews Cancer, it focused on advances regarding the role of S100 proteins in cancer diagnosis and treatment, the contribution of S100 signaling to cancer cell biology and the development of new S100 protein inhibitors for treating cancer (*Nat Rev Cancer.*, 2015, PMID: 25614008). The

expression of S100A8 and S100A9 in diverse cancers were summarized as following (**Table RL1**). S100A8 and S100A9 universally expressed in breast, lung, prostate, colorectal, brain, thyroid, thymus cancers, lymphoma and melanoma, and had significant associations with cancer's phenotype, including growth, metastasis, angiogenesis, and immune evasion. These researches demonstrated S100A8 and S100A9 could be regarded as the potential biomarkers for these cancers' diagnosis and/or prognosis. Therefore, we agree with the reviewer that the S100 family members (S100A8 and S100A9) universally expressed in many tissues with different sources; additionally, in this study, we validated the high expression of S100A8 and S100A9 in the tumor tissues of CRC patients, indicating the elevated S100A8 and S100A9 identified in plasma are most likely originated from CRC tumor. As for THBS2, a pan-cancer biomarker discovery study including tissues, plasma, and other bodily fluids, revealed a panel of proteins, including THBS2, could distinguish tumors from normal tissues with high accuracy, suggesting that these proteins, including THBS2, could be used as pan-cancer markers, involving both adult cancers (pancreatic, lung, breast, and colorectal carcinomas and melanoma) and pediatric cancers (neuroblastoma and osteosarcoma) (*Cell.*, 2020, PMID: 32795414). In addition, Kozumi K *et al.* showed THBS2 may be a useful biomarker for nonalcoholic steatohepatitis (NASH), the progressive form of nonalcoholic fatty liver disease (NAFLD), and advanced fibrosis diagnosis in patients with NAFLD (*Hepatology.*, 2021, PMID: 34105780). Carpino G *et al.* also showed that THBS2 released in the tumor microenvironment could inhibit vascular growth while promoting cancer-associated lymphangiogenesis, and proposed that targeting THBS2 may be a promising strategy to reduce cancer-associated lymphangiogenesis and counteract the invasiveness of intrahepatic cholangiocarcinoma (*J Hepatol.*, 2021, PMID: 34329660). For COL12A1, fewer previous studies but a recent study published by Papanicolaou M *et al.* demonstrated that COL12A1 may represent an indicator of breast cancer patients at high risk of metastatic relapse in patient cohorts (*Nat Commun.*, 2022, PMID: 35933466). Overall, S100A8, S100A9, and THBS2 were identified as biomarkers for different cancers (including colorectal cancer), and COL12A1 might associated with breast metastasis in the previous researches. As a review published in Nature reported, as cancer is increasingly defined by dysregulated pathways, relevant biomarkers may cut across tumor types without showing tissue specificity (*Nature.*, 2008, PMID: 18385731). In our study, the four

biomarkers (S100A8, S100A9, THBS2 and COL12A1) were screened in more strict criteria based on a large clinical CRC patient cohort, resulting in a predictive model with high accuracy for distinguishing CRC patients from healthy controls, estimated by receiver operating characteristic (ROC) analysis. Moreover, to further verify the association of these biomarkers with tumor tissues, we included tumor tissue samples matched with plasma from CRC patients for a proteomic comparison, which contributed to refine the biomarkers for distinguishing the CRC patients from healthy controls, and validated the predictive effect in the independent cohort. Therefore, the four biomarkers identified in this study demonstrated the relative robustness in distinguishing the CRC from healthy controls both in plasma samples and tumor tissue samples. The clinical measurement and implication of these potential biomarkers deserved for us to further verify in the future studies.

Figure RL2. The proteomic differential analysis between tumor (T) and normal-adjacent tissues (NATs). (A) The dynamic range of protein identification of Tumors and NATs was shown according to the descending sort of protein abundance of each sample grouped into T and NAT groups. (B) The overlap of the proteins identified in tissues and plasma samples. (C) Volcano showing the differential expressed proteins between T and NAT groups (two-sided Wilcoxon rank-sum test). Red, upregulated proteins in T group; blue, upregulated proteins in NAT group. (D) The overlap of the significantly up-regulated proteins in FFPE tumor tissues and plasma samples from CRC patients. (E) Boxplot showing the differential expression of the four diagnostic biomarkers (COL12A1, THBS2, S100A8 and S100A9) between T and NAT groups (two-sided Wilcoxon rank-sum test, adj P -value < 0.05). (F) The ROC curve showing

the predictive effect of the diagnostic model in the tissue validation cohort.

Table RL1 The expression of the four plasma biomarkers in cancers.

Protein	Cancer type	PMID
S100A8	Breast cancer	24177755
	Prostate cancer	24122301
	Colorectal cancer	21228116
	Thyroid cancer	25423568
	Thymus cancer	24859530
	Lung cancer	24778162
		21795396
S100A9	Prostate cancer	22470535
		24122301
	Colorectal cancer	21228116
	Brain cancer	23836528
	Lymphoma	18809714
	Melanoma	22470535
	Thymus cancer	23135911
THBS2	Thymus cancer	24859530
	Pan-cancer	32795414
	Liver cancer	34105780
COL12A1	Intrahepatic cholangiocarcinoma	34329660
	Breast cancer	35933466

(3) The potential biological function and association of the four biomarkers

To explore the potential biological association of the four biomarkers (COL12A1, THBS2, S100A8 and S100A9), we firstly performed gene set enrichment analysis (GSEA) (*Proc Natl Acad Sci U S A.*, 2005, PMID: 16199517) for pathway enrichment analysis, which could identify the biological association of these proteins. GSEA is a computational method to determine whether a priori defined set of genes has statistical significance and concordant differences in two biological states. Sample grouped according to tumor patients and healthy controls were subjected to GSEA based on the proteomic data. An FDR value of 0.05 was used as a cutoff. The GSEA results demonstrated that the four proteins COL12A1, THBS2, S100A8, and S100A9 were enriched in collagen-containing extracellular matrix (GO:0062023) based on the cellular component database according to the Gene Ontology annotation (**Figure RL3A, see also Figure 2C in the revision**). To validate this result, we performed single sample gene set enrichment analysis (ssGSEA) (*BMC Bioinformatics.*, 2013, PMID: 23323831), which defined an enrichment score that represents the degree of absolute of a gene set in each sample, were

applied to protein expression matrix through GSVA R package in proteomic studies (*Cancer Cell.*, 2022, PMID: 35245447; *Cell.*, 2020, PMID: 33242424). Further correlation analysis revealed that COL12A1, THBS2, S100A8, and S100A9 showed significantly positive correlation with extracellular matrix organization (**Figure RL3B, see also Figure S2D in the revision**). The protein-protein interaction network identified by String and Cytoscape visualization software revealed a close connection among these proteins, especially for the four proteins including COL12A1, THBS2, S100A8, and S100A9 (**Figure RL3C, see also Figure 2C in the revision**). As reported, S100A8/A9 are often co-expressed and function mainly as heterodimer; when these intra-cellular proteins are released into the extracellular medium, they could activate specific downstream proteins associated with tumorigenesis and in promoting tumor growth and metastasis (*Mol Cancer Res.*, 2011, PMID: 21228116; *Nat Rev Cancer.*, 2015, PMID: 25614008). COL12A1 and THBS2 were also reported involved in extracellular matrix and downstream signaling pathways, and could be served as prognostic and therapeutic biomarkers in cancers (*Cancer Med.*, 2020, PMID: 32356618; *J Cell Biochem.*, 2018, PMID: 29143985). These results also supported our finding revealed in this study. Overall, the four proteins had the potential biological association, which showed significant enrichment to the collagen-containing extracellular matrix.

Although the four proteins COL12A1, THBS2, S100A8, and S100A9, as the potential biomarkers for CRC diagnosis with high accuracy, were elevated both in plasma and tumor tissues of CRC patients, we couldn't completely exclude the contribution of other organs or tissues to the level of these proteins in plasma, which might need to be explored through comparison among more tissues. A more rigorous multi-tissue comparison research is worth studying in the future. In the revision, we also added the information in the "Limitations of the study" in the manuscript.

Finally, in the revision, we supplemented the section related to screening process of protein biomarkers for CRC diagnosis in the **Method "Screening biomarkers for CRC diagnosis"** (line 880–895 in Page 30), and the section related to the potential biological association in the **Method "The gene set enrichment analysis (GSEA)"** and **"Protein–protein interaction**

(PPI) network”, updated the result related to biological association of four biomarkers and their identification in the matched tissues validation cohort composed of 31 tumor tissues and 27 paired normal adjacent tissues (NATs) in **Figures 2 and S2**, as well as **Result “The plasma proteomic biomarkers to distinguish CRC patients from healthy controls”** in the revision (line 162–288 in Page 6–10).

Figure RL3. The potential biological function and association of the four biomarkers. (A) GSEA shows collagen containing extracellular matrix are enriched in CRC patients, and the four proteins (COL12A1, THBS2, S100A8, and S100A9) are listed in this item. FDR < 0.05 is considered statistically significant. (B) Correlation analysis of ssGSEA pathway scores of extracellular matrix organization with the four diagnostic biomarkers (two-sided Spearman’s correlation test). $P < 0.05$ is considered statistically significant. (C) The protein-protein interaction network of the extracellular matrix organization including the four diagnostic biomarkers identified by String and Cytoscape visualization software.

Q2. Line 205, 'To further determine whether these four proteins could effectively diagnosis patients with CRC...'? If the requirement for the proteins to be differentially expressed in tumor were dropped, would the resulting protein list (or maybe the top 10 proteins from this list) perform better than these 4 proteins?

Response to Q2:

Thanks for the comment of the diagnostic biomarkers. According to reviewer’s suggestions, we dropped the differential expression from 2-fold to 1.5-fold change in plasma, resulting in 190 proteins with false discovery rate (FDR) adjusted p-value (adj P -value) less than 0.01 identified in at least 50% of the samples. Compared with the previous screening criteria of 2-fold, we

identified additional 42 proteins as the potential biomarker candidates. After including the comparison of tissue proteomic data of CRC patients from the Office of Cancer Clinical Proteomics Research (CPTAC) cohort (*Cell.*, 2019, PMID: 31031003) as the screening criteria, we obtained a group of 6 proteins including COL12A1, THBS2, S100A8, S100A9, MXRA5, and S100A12 both in plasma and tissue (**Figure RL4A**). Here, we defined the four proteins COL12A1, THBS2, S100A8, and S100A9, involved in the previous version, as “Model 1”. Based on proteins included in “Model 1”, we performed the prediction of Model 1 adding MXRA5 or/and S100A12. As a result, compared with “Model 1” in the previous version, we found a gradually improvement in the predictive effect with an increasing AUC from 0.91 (Model 1), to 0.919 (“Model 2” composed of COL12A1, THBS2, S100A8, S100A9, and S100A12), 0.924 (“Model 3” composed of COL12A1, THBS2, S100A8, S100A9, and MXRA5), and 0.931 (“Model 4” composed of COL12A1, THBS2, S100A8, S100A9, S100A12, and MXRA5) (**Figures RL4B and 4C**). These results demonstrated that the four proteins had high accuracy for CRC diagnosis, and more biomarkers included in the predictive model did exactly performed better than the four proteins, as reviewer’s expected.

To validate the predictive effect of the “Model 1” composed of 4 proteins and the “Model 4” composed of 6 proteins, we included more CRC patients in the plasma validation cohort composed of 31 CRC patients and 24 healthy controls for the proteomic measurement. Differential proteomic analysis identified these proteins of Model 1 and Model 4 (including COL12A1, THBS2, S100A8, S100A9, S100A12, and MXRA5) showed a significant up-regulation with at least 2-fold change (adj *P*-value < 0.05). The receiver operating characteristic (ROC) analysis demonstrated the Model 1 composed of 4 proteins and Model 4 composed of 6 proteins could well distinguish the CRC patients from healthy controls with the AUC values of 0.952 and 0.968, respectively, in the plasma validation cohort (**Figures RL4D and 4E**). In addition, to further validate the association of these biomarkers with tumor tissues, we also reviewed all FFPE tissues from the therapy-naïve CRC patients included in this study, and collected 31 tumor tissues and 27 paired normal-adjacent tissues (NATs) of CRC patients matched with the plasma samples for the proteomic profiling. Obviously, COL12A1 (FC = 4.29, adj *P*-value = 8.8E-4), THBS2 (FC = 3.43, adj *P*-value = 1.6E-4), S100A8 (FC = 6.73, adj *P*-

value = 8.7E-5), S100A9 (FC = 7.64, adj *P*-value = 1.2E-4), S100A12 (FC = 2.04, adj *P*-value = 0.041), and MXRA5 (FC = 2.04 adj *P*-value = 0.048) showed a significant increase in the tumor tissues. Further ROC analysis revealed the Model 1 and the Model 4 achieved a good performance with AUC values of 0.945 and 0.958, respectively, in the tumor tissue validation cohort (**Figures RL4D and 4E**). Overall, these results indicated the robustness of these predictive models both in the independent plasma validation cohort and tissue validation cohort.

In the revision, considering the plasma-matched tissue samples included in this study, to search for the potential proteins applied for plasma and tissue samples that could be used as diagnostic biomarkers for CRC patients, we added our tissue validation cohort composed of 31 tumor tissues and 27 paired normal-adjacent tissues (NATs) of CRC patients matched with our plasma samples for the biomarker discovery. We adopted the same strict screening strategy in the tissue samples as follows: 1) The candidate proteins were expressed in at least 50% of the samples; 2) The candidates were significantly increased in tumor samples than normal samples (two-sided Wilcoxon rank-sum test, adj *P*-value < 0.01); 3) The candidates were identified with at least 2-fold increase in CRC samples than NATs. As a result, we screened 797 proteins significantly overexpressed in the tumor tissues of CRC patients. According to the reviewer's suggestions, we dropped the differential expression from 2-fold to 1.5-fold change in the tissue samples, resulting in 888 proteins with false discovery rate (FDR) adjusted *p*-value (adj *P*-value) less than 0.01 identified in at least 50% tumor samples than NATs. Combined our independent plasma discovery cohort, tissue validation cohort, and the CPTAC cohort, finally, we obtained a group of four proteins, COL12A1, THBS2, S100A8, and S100A9, significantly increased both in plasma and tissues of CRC patients in our study and the CPTAC study (**Figure RL4F**). We found two proteins, S100A12 and MXRA5, in the predictive model composed of six proteins (COL12A1, THBS2, S100A8, S100A9, S100A12, and MXRA5) didn't meet adj *P*-value less than 0.01 ($0.01 < \text{adj } P\text{-value} < 0.05$) in the tissue validation cohort, and were excluded. Therefore, when dropping 2-fold to 1.5-fold change, there were still only the four proteins (COL12A1, THBS2, S100A8, and S100A9) were screened among the three independent cohorts, indicating the four proteins could be used as the potential diagnostic biomarkers.

Finally, we mainly focused on the four diagnostic biomarkers, and updated the relative results in the revision (line 221–247 in Page 8–9, line 270–288 in Page 9–10). The Figures RL4D and 4F were updated in the Figures 2G and S2A in the revision.

Figure RL4. The diagnostic models with different protein combinations and their diagnostic efficacy. (A) The identification of differentially expressed proteins with 1.5-fold / 2-fold and adj P -value < 0.01 (two-sided Wilcoxon rank-sum test). (B) The predictive effects of the different models with different protein panels (Model 1, Model 2, Model 3, and Model 4). (C)

The ROC curves and AUC values of the four models (Model 1, Model 2, Model 3, and Model 4). (D and E) The ROC curves of the Model 1 composed of four proteins (COL12A1, THBS2, S100A8, and S100A9) (D) and Model 4 composed of six proteins (COL12A1, THBS2, S100A8, S100A9, S100A12, and MXRA5) (E) in the plasma discovery cohort, as well as plasma and tissue validation cohorts. (F) The screening flow of diagnostic biomarkers with 2-fold and 1.5-fold differential expression.

Q3. Line 222, the results in this section 'Proteomic subtyping of the CRC cohort...' and also in other parts the manuscript describe differences in the context of signaling, immune infiltration etc, but a majority of these findings are from plasma samples. How much of these findings are a reflection of what is going on in the tumor? How does this compare to what is going on systemically in the patient? How does this compare to what is secreted by circulating immune cells? These are very important points that are essentially ignored. Unless authors can prove the source (e.g. cell, tissue) of the changes in the proteins and pathways they describe, then the results need to be reworked, and it should be clearly stated that the origins/causes of these changes is systemic/unclear.

Response to Q3:

We sincerely thank the reviewer for the careful read and constructive comments. We apologize for ignoring the important points, and we also understand the reviewer's concerns. Here, according to the reviewer's suggestion, to explore how much of these findings in plasma are a reflection of what is going on in the tumor, we reviewed all FFPE tissues from the therapy-naïve CRC patients included in this study, and collected 31 tumor tissues and 27 paired normal-adjacent tissues (NATs) from the therapy-naïve CRC patients matched with the plasma samples for the proteomic profiling in the revision. **Firstly**, the comparative proteomic analysis demonstrated that 2,573 proteins were significantly elevated in tumor tissues; while 251 proteins were down-regulated in tumor tissues (fold change (Tumor vs NAT) >2, adj *P*-value < 0.05). Further comparison of these differential expressed proteins (DEPs) identified in plasma samples and tissue samples demonstrated that 31.54% of the proteins up-regulated in the plasma from CRC patients were identified with consistent alteration in the tissue samples. For

example, the four proteins COL12A1, THBS2, S100A8 and S100A9, as the potential biomarkers for CRC diagnosis, were significantly up-regulated both in plasma samples and tumor tissue samples from the CRC patients compared with the plasma samples from healthy controls or the NATs (at least 2-fold increase with adj *P*-value < 0.01), exhibiting good performance with AUC values of 0.952 and 0.945 respectively in plasma and tissue validation cohorts in distinguishing the CRC patients from healthy controls. To further explore the alteration of the pathways identified in the CRC plasma, we performed the pathway enrichment analysis based on these DEPs according to the ConsensusPathDB (CPDB) database. Consistent with the enrichment results in plasma proteome, we found that glycolysis and gluconeogenesis, adherens junction, TP53 transcriptional regulation, cellular response to chemical stress, and oncogenic MAPK signaling pathways were enriched in tumor tissues; while immune regulation such as complement system were enriched in NATs (**Figure RL5A, see also Figure S1H in the revision**). These results revealed the alteration of proteins and pathways identified in the comparison of between CRC patients and healthy controls based on plasma proteome have a relatively well reflection in the tissue samples.

Secondly, to validate the association of immune signaling pathways with sensitive response concluded from the plasma proteome data, we grouped the 31 CRC patients into 12 sensitive patients (S group, including 12 partial response (PR)); 19 non-sensitive patients (NS group, including 18 stable disease (SD) and 1 progress disease (PD)) according to the RECIST 1.1 guideline. Consistent with the pathway characteristics overrepresented in the sensitive subtype (G-III subtype), we found the immune pathways, such as antigen processing, TCR signaling, and autophagy were enriched in the sensitive group of the tissues of the CRC cohort (**Figure RL5B, see also Figure S3E in the revision**). In addition, to validate the immune subpopulations identified in plasma proteomic data, we performed xCell analysis on the tissue proteomic data, which could provide the reference of immune microenvironment. Consistently, the xCell score of CD8+Tem was significantly elevated in S group compared with NS group based on the tissue proteome (**Figure RL5C, see also Figure 3E in the revision**). Furthermore, we explored the representative signatures (CD44 and GZMK) related to the CD8+ Tem function (*Nature.*, 2018, PMID: 30479382). Interestingly, CD44 and GZMK had an obvious

increase in the S group both in plasma samples and tissue samples (**Figure RL5D**). To directly address the result, we also performed immunohistochemistry (IHC) of CD44 and GZMK to evaluate tumor infiltration in FFPE tumor tissue from therapy-naïve CRC patients. As a result, the expression of CD44 and GZMK was significantly increased in S group compared with NS group. Moreover, S group had significantly increased percentage of CD44 positive cells (53.9%) and GZMK positive cells (47.2%), compared with NS group (9.7% and 15.7%, respectively) ($P < 1E - 4$) (**Figure RL5E, see also Figure 3F in the revision**). These results verified that CRC patients featured with immune activation are likely to benefit from cetuximab therapy, validating our findings concluded from plasma proteomic data. Therefore, our findings in plasma had a relative consistency in the matched tissues.

Thirdly, to further associate the immune related findings with tumor response to cetuximab therapy of CRC patients, we reviewed the computed tomography (CT) or magnetic resonance imaging (MRI) scanning results, which were used to reflect tumor systemically change. Here, we provided the representative CT/MRI image evaluation of CRC patients (**Figure RL5F, see also Figure S3F in the revision**). Obviously, we observed significant tumor regression in sensitive patients and tumor progression in non-sensitive patients in the post-therapy of cetuximab when compared with the pre-therapy. Taken together with the tumor size and CD8+Tem xCell score, we found patients with higher CD8+Tem xCell score were prone to have a smaller tumor size in the baseline evaluation (**Figure RL5G, see also Figure 3G in the revision**). Further correlation analysis showed there was a significantly negative correlation (Pearson $r = -0.39$, $P = 0.033$) between tumor size and CD8+Tem xCell score (**Figure RL5H, see also Figure 3H in the revision**). These results demonstrated that CD8+Tem could be regarded as a potential marker for the cetuximab sensitivity. Moreover, based on the tissue proteome, the correlation analysis revealed a significantly positive correlation of CD8+Tem with RPTOR and IMPDH2 (Pearson $r = 0.43$, $P = 0.015$; Pearson $r = 0.43$, $P = 0.017$), respectively (**Figure RL5I, see also Figures 3J and S3H in the revision**), consistent with the finding in the plasma proteome.

We thank the reviewer's suggestion. The reviewer is correct. Although we observed the

consistent alteration of the proteins and pathways uncovered both in plasma samples and tumor tissues of CRC patients, we couldn't completely exclude the contribution of other tissues or cells in plasma. It is deserved to explore the components secreted by circulating immune cells associated with the therapy response in the future. In the revision, we have addressed this limitation of our findings and data in the **“Discussion”** and **“Limitations of the study”** sections.

Overall, the dysregulated proteins and pathways identified in plasma could have a well reflection in the proteome profiling of tumor tissues, and correlated with tumor systemically change. Finally, in the revision, we updated the results in the **Figures 3, S1 and S3**, and added the results related to the tissues in **“The plasma proteomic biomarkers to distinguish CRC patients from healthy controls”** (line 196–219 in Page 7–8), and **“The potential molecular features and biomarkers for the initial response to cetuximab therapy”** (line 343–409 in Page 12–14) in the **“Result”** section in the revision. In addition, we also made a clear statement of these proteins and pathways alteration in the **“Manuscript”**, and addressed the limitation of our findings in in the **“Discussion”** (line 622–627 in Page 21) and **“Limitations of the study”** (line 669–684 in Page 23) sections in the revision.

Figure RL5. The validation of plasma findings in tissue samples. (A) Bubble plots showing the CPDB pathway enrichment (two-sided Fisher's exact test) of tumor tissues and NATs. (B) Bubble plots showing the CPDB pathway enrichment (two-sided Fisher's exact test) of S group. (C) Boxplot for the CD8+ Tem xCell score between S group and NS group in tissue samples ($N(S) = 12$, and $N(NS) = 19$) (two-tailed Student's t test, $P < 0.05$). Boxplots show median

(central line), upper and lower quartiles (box limits), 1.5×interquartile range (whiskers). $P < 0.05$ is considered statistically significant. (D) Barplot showing the differential expression of CD44 and GZMK between S and NS group in the plasma samples and tissue samples. (E) The qualification of CD44 and GZMK stained by immunohistochemistry (IHC) in representative examples. $P < 0.05$ is considered statistically significant. Data are analyzed by two-sided Student's t test and shown as mean \pm SD ($n = 3$ independent experiments). (F) The computed tomography (CT) or magnetic resonance imaging (MRI) scanning features of the representative patients in S and NS group. (G) Sankey plot showing the association between CD8+Tem xCell score with tumor size, and S/NS group in the tissue validation cohort. The little heatmap indicates the relative abundance of CD8+Tem (Z score) and tumor size (cm). (H) The scatter plot showing correlation analysis of CD8+Tem xCell and score tumor size in our tissue cohort (two-sided Pearson's correlation test). $P < 0.05$ is considered statistically significant. (I) The scatter plot showing correlation analysis of CD8+Tem xCell score and RPTOR/IMPDH in the tissue samples (two-sided Pearson's correlation test). $P < 0.05$ is considered statistically significant.

Q4. Line 247, why is xCell being applied to proteomics data? Please justify the use of xCell applied to proteomics data. Deconvolution methods are generally developed with the high coverage of transcriptomics in mind. Although the authors are describing good results with a reasonable number of gene products for a proteomics study, the authors have a relatively small number of gene products compared to microarray/RNA-seq. Further, this seems to be plasma. Are the authors assuming there are enough secreted proteins found in plasma to do a good job at approximating circulating immune subpopulations? This appears to be a significant limitation of these analyses. Applying xCell to proteins found in plasma seems to be a very big overreach and the subsequent results are likely over-interpreted. The authors appear to provide scores for xCell in the supplemental data but not p-values. Were any of the FDR-adjusted p-values for these score estimates from xCell still significant? If the xCell results are to be included, then authors need to show xCell does a good job at estimating immune subpopulations from plasma. Ideally, some independent experiments to validate these immune

subpopulations would be provided.

Response to Q4:

Thanks for the comment. We apologize for not clarifying the xCell used in the proteomics data, as well as statistical interpretation and the validation of the xCell results. According to reviewer's suggestions, we divided the response into three parts to answer: (1) about the justification for the use of xCell applied to proteomics data; (2) about the statistical significance in the xCell analysis; (3) about the validation of the findings uncovered by xCell analysis.

(1) About the justification for the use of xCell applied to proteomics data

As this section mentioned by reviewer, we determined the dominant bioprocesses of each subtype. As a sensitive subtype, the G-III subtype was featured by cellular response to chemical stress, autophagy, neutrophil degranulation, innate immune system, TP53 regulates metabolic genes, and T cell receptor signaling pathway. Further analysis based on ssGSEA score indicated the interaction of autophagy with neutrophil degranulation and innate immune system. These results demonstrated that the overrepresented pathways dominated in the G-III subtype (sensitive subtype) were immune related pathways. As reported, xCell (*Genome Biol.*, 2017, PMID: 29141660, <https://xcell.ucsf.edu/>) is a webtool that performs cell type enrichment analysis from gene expression data for 64 immune and stroma cell types, which has been applied in many researches related to tumor proteomics and transcriptomics, and uncovered novel insight which also be validated by other independent experiments (*Cell.*, 2019, PMID: 31675502; *Cell.*, 2020, PMID: 33242424; *J Hematol Oncol.*, 2022, PMID: 35659036; *Hepatology.*, 2023, PMID: 35716043). As reported, the tumor microenvironment infiltration estimated by proteomic data had a high Pearson correlation with ones estimated by transcriptomic data (*Cell.*, 2019, PMID: 31675502; *J Hematol Oncol.*, 2022, PMID: 35659036) concluded from these published researches, indicating the potential of proteome in xCell analysis to reveal the tumor microenvironment infiltration. For example, in the multilevel proteomic research of diffuse-type and intestinal-type gastric cancer (*Nat Commun.*, 2023, PMID: 36788224), the immune clustering of xCell-deconvoluted tumor microenvironment components based on proteomic data revealed Th1/Th2 ratio could serve as an indicator for immunotherapeutic effectiveness, which was validated in an independent GC anti-PD1

therapeutic patient group. In addition, a proteogenomic search of cholangiocarcinoma (*Hepatology.*, 2023, PMID: 35716043) revealed that a higher level of xCell-derived CD4+ T cells based on proteomic data was associated with the favorable prognosis, which was further confirmed in a combined cohort. These researches showed the novel findings uncovered by xCell based on proteomic data could be further validated by other independent experiments, indicating the importance of proteomic data in the tumor microenvironment. Therefore, in this study, to further explore the role of immune modulation in the therapy response, we attempted to evaluate the immune microenvironment through xCell analysis applied in proteome data.

(2) About the statistical significance in the xCell analysis

In this study, the cell type enrichment score assessed with statistical significance was calculated and used to infer the relative abundance of different cell types in the tumor microenvironment in the online enrichment streamline at <https://xcell.ucsf.edu/> (*Genome Biol.*, 2017, PMID: 29141660), which was also applied in many transcriptomic and proteomic researches (*Cell.*, 2019, PMID: 31675502; *Cell.*, 2020, PMID: 33242424; *Cell.*, 2020, PMID: 33212010). In this study, we found that G-III subtype had the highest immune score compared with other two subtypes, and CD8+ Tem were significantly enriched in G-III subtype (fold change > 2, adj *P*-value < 0.05). Further comparative analysis showed xCell score of CD8+Tem was significantly dominant in S group compared with NS group. According to the reviewer's suggestions, we performed the false discovery rate (FDR) estimation on these xCell scores by the Benjamini-Hochberg (BH), which were defined as adj *P*-value in the revision. After BH adjust, we found CD8+Tem xCell score still significant with an adj *P*-value of 0.027 in the comparison between G-III subtype and other two subtype (G-I and G-II) (**Figure RL6A, see also Figure 3D in the revision**), which showed consistency with our major results in the last version. We also updated these results, especially the adj *P*-value, in the figures and supplementary tables in the revised version.

(3) About the validation of findings uncovered by xCell analysis

To further validate the immune subpopulations uncovered in plasma proteomic data, we adopted two strategies: (I) xCell data analysis based on the tissue proteomic data; (II)

immunohistochemistry (IHC) staining and evaluation using antibodies.

(I) Firstly, we reviewed all FFPE tissues from the therapy-naïve CRC patients included in this study, and collected 31 tumor tissues from the therapy-naïve CRC patients matched with the plasma samples for the proteomic profiling in the revision. According to the RECIST 1.1 guideline, the 31 CRC patients were grouped into 12 sensitive patients (S group, including 12 partial response (PR)); 19 non-sensitive patients (NS group, including 18 stable disease (SD) and 1 progress disease (PD)). To validate the immune subpopulations identified in plasma proteomic data, we performed xCell analysis on the tissue proteomic data, which could provide the reference of immune microenvironment. Consistently, the xCell score of CD8+Tem was significantly elevated in S group compared with NS group based on the tissue proteome (**Figure RL6B, see also Figure 3E in the revision**). The results showed a consistency of the immune subpopulations identification in plasma samples and tissue samples.

(II) Secondly, we explored the representative signatures (CD44 and GZMK) of the CD8+ Tem (*Nature.*, 2018, PMID: 30479382). Interestingly, CD44 and GZMK had an obvious increase in the S group both in plasma samples and tissue samples (**Figure RL6C**). To directly address the result, we also performed immunohistochemistry (IHC) of CD44 and GZMK to evaluate tumor infiltration in FFPE tumor tissue from therapy-naïve CRC patients. As a result, the expression of CD44 and GZMK was significantly increased in S group compared with NS group. Moreover, S group had significantly increased percentage of CD44 positive cells (53.9%) and GZMK positive cells (47.2%), compared with NS group (9.7% and 15.7%, respectively) ($P < 1E - 4$) (**Figure RL6D, see also Figure 3F in the revision**). These results verified that CRC patients featured with higher CD8+Tem level are likely to benefit from cetuximab therapy, validating our findings concluded from plasma proteomic data and demonstrating a relative consistency in the matched tissues. Therefore, our findings validated that the immune subpopulations revealed by xCell analysis in plasma proteomic data could be further verified by the proteome measurement and IHC evaluation on tissue samples.

To further explore the clinical implication of CD8+Tem, we further associated the CD8+Tem

score with tumor size evaluated by CT/MRI. The results demonstrated patients with higher CD8+Tem xCell score were prone to have a smaller tumor size in the baseline evaluation (**Figure RL6E, see also Figure 3G in the revision**). Further correlation analysis showed there was a significantly negative correlation (Pearson $r = -0.39$, $P = 0.033$) between tumor size and CD8+Tem xCell score (**Figure RL6F, see also Figure 3H in the revision**), implying the CD8+Tem could be the potential marker for the cetuximab sensitivity. To further validate this finding, we included the single-cell transcriptome data from CRC patients (*Nature.*, 2018, PMID: 30479382), which provided the reference of microenvironment in colorectal cancer and could be obtained from Gene Expression Omnibus (GEO) (accession number GSE108989). Then, we combined the single-cell transcriptome data and the clinical characteristics of each patient for the further analysis. We performed the correlation analysis of the CD8+Tem cell percentage and tumor size in the single-cell transcriptome data. Consistent with our findings uncovered by the proteome data, the result demonstrated the inferred proportion of CD8+Tem cell showed a significantly negative correlation (Spearman $r = -0.72$, $P = 0.019$) with tumor size, which further confirmed the association of CD8+Tem cell with tumor size (**Figure RL6G, see also Figure S3G in the revision**). These results demonstrated that CD8+Tem could be regarded as a potential marker for the cetuximab sensitivity.

Overall, xCell analysis applied in plasma proteomic data assist us finding the important indicator CD8+Tem for cetuximab sensitivity, which were further validated by the proteome measurement, IHC evaluation on tissue samples, and the independent single-cell transcriptome data. Finally, we toned down the description of the relative results, updated the FDR adjusted P -values (named as “adj P -value”) in the **Figure 3 and Table S4 in the revision**; supplemented the validation results of proteome measurement and IHC evaluation on tissue samples in the **“The potential molecular features and biomarkers for the initial response to cetuximab therapy”** (line 362–391 in Page 13–14) in the **“Result”** section in the revision, **“Immune cell type composition”** (line 961–973 in Page 33), **“Immunohistochemistry staining and evaluation”** (line 1026–1033 in Page 35) in the **“Method”** section, as well as **Figure 3 and Figure S3 in the revision**.

Figure RL6. Immune subpopulations uncovered in plasma samples were validated by the proteome measurement and IHC evaluation on tissue samples. (A) Volcano showing the differential cell types of G-III subtype compared with other two subtypes (two-sided Wilcoxon rank-sum test, adj P -value < 0.05 ; fold change > 2). (B) Boxplot for the CD8+Tem xCell score between S group and NS group in the independent tissue validation cohort (N (S) = 12, and N (NS) = 19) (two-tailed Student's t test, $P < 0.05$). Boxplots show median (central line), upper and lower quartiles (box limits), $1.5 \times$ interquartile range (whiskers). $P < 0.05$ is considered statistically significant. (C) Barplot showing the differential expression of CD44 and GZMK between S and NS group in the plasma samples and tissue samples. (D) The qualification of CD44 and GZMK stained by immunohistochemistry (IHC) in representative examples. $P < 0.05$ is considered statistically significant. Data are analyzed by two-sided

Student's t test and shown as mean \pm SD (n = 3 independent experiments). (E) Sankey plot showing the association between CD8+Tem xCell score with tumor size, and S/NS group in the tissue validation cohort. The little heatmap indicates the relative abundance of CD8+Tem (Z score) and tumor size (cm). (F) The scatter plot showing correlation analysis of CD8+Tem xCell and score tumor size in our tissue cohort (two-sided Pearson's correlation test). $P < 0.05$ is considered statistically significant. (G) The scatter plot showing correlation analysis of CD8+Tem xCell and score tumor size in the single-cell transcriptome data (two-sided Pearson's correlation test). $P < 0.05$ is considered statistically significant.

Q5. Line 285, please clarify the sentence '...high expression of RRAS and RRAS2 were prone to resistant significantly increased (FC > 2, 286 P < 0.05) (Figure 4D).

Response to Q5:

Thank the reviewer for pointing out the unclear description. Here, in this study, we found that RRAS and RRAS2 were significantly increased in the non-sensitive (NS) group, indicating CRC patients featured with high expression of RRAS and RRAS2 were prone to be resistant to the cetuximab therapy. Therefore, we revised the sentence as "In this study, we found CRC patients with high expression of RRAS and RRAS2 were prone to be resistant to the cetuximab therapy (FC > 2, $P < 0.05$) (Figure 4E)" in the revision (**line 428–430 in Page 15**).

Q6. ssGSEA is used throughout the manuscript, and it seems to be used for comparing two groups. Why is traditional GSEA not used for two group comparisons? Were all ssGSEA scores used, or was there some cutoff based on the p-values for the score? How is differential expression done between ssGSEA scores? Please clarify these points in the methods.

Response to Q6:

Thanks for the reviewer constructive comments. We apologize for not clarifying the details in the ssGSEA analysis. As the reviewer mentioned, the ssGSEA was involved in two sections in our manuscript: the one section is the correlation analysis of the featured pathways up-

regulated in G-III subtype; the other section is the differential analysis of the featured pathways up-regulated in G-I subtype between S/NS groups, as well as RRAS high/low expression groups. Here, we separately present related description of the ssGSEA, especially for the statistical method; in addition, we performed the traditional GSEA according to the reviewer's suggestions, and concluded the consistent results. The point-to-point response was summarized as follows:

(1) As for the correlation analysis of the featured pathways up-regulated in G-III subtype:

In our study, we identified three proteomic subtypes (G-I, G-II, and G-III) associated with CRC cetuximab therapy response based on the plasma proteome, of which G-III subtype had higher proportion of sensitive patients while G-I subtype had higher proportion of non-sensitive patients. To determine the dominant biological pathways of each subtype, we first performed ConsensusPathDB (CPDB) enrichment analysis (<http://cpdb.molgen.mpg.de/CPDB>, a meta-database combining molecular interaction data obtained from 31 different public repositories for humans) using the overrepresentation method based on the differential expression proteins (*Nucleic Acids Res.*, 2022, PMID: 34850110; *Cell.*, 2020, PMID: 32649877). Pathway enrichment analysis revealed G-III subtype was featured with cellular response to chemical stress, autophagy, neutrophil degranulation, innate immune system, TP53 regulates metabolic genes, and T cell receptor signaling pathway. To explore the association of these pathways, we further applied the single sample gene set enrichment analysis (ssGSEA) method in GSVA R package (*BMC Bioinformatics.*, 2013, PMID: 23323831) based on the protein expression matrix, which could define an enrichment score representing the degree of absolute of a gene set in each sample based on the gene set (c2.all.v7.4.symbols) from Molecular Signature Database (MSigDB), and also be used in many proteomic studies (*Cancer Cell.*, 2022, PMID: 35245447; *Cell.*, 2020, PMID: 33242424). In the ssGSEA, the GSVA algorithm itself does not evaluate statistical significance for the enrichment of gene sets (*BMC Bioinformatics.*, 2013, PMID: 23323831); in the following analysis, the significance with respect to a phenotype can be evaluated using conventional statistical models. Then, we obtained the ssGSEA score of pathways in each sample, and further performed the correlation analysis based on ssGSEA scores. The results showed that among these pathways enriched in G-III subtype, autophagy

showed significant association with neutrophil degranulation or innate immune system, indicating the interaction of autophagy with neutrophil degranulation and innate immune system. The ssGSEA analysis provided us a more in-depth understanding of the association among these pathways.

In addition, according to the reviewer's suggestions, we also performed the gene set enrichment analysis (GSEA) for pathway enrichment analysis, to validate the pathways mentioned above enriched in the G-III subtype compared with the other two subtypes. GSEA is a computational method to determine whether a priori defined set of genes has statistical significance and concordant differences in two biological states (*Proc Natl Acad Sci U S A.*, 2005, PMID: 16199517). Sample grouped according to the proteomic subtypes were subjected to GSEA based on the proteomic data. Molecular Signatures Database (MSigDB) of hallmark gene sets (H), curated gene sets (H2) and GO gene sets (C5) were used for enrichment analysis. An FDR value of 0.05 was used as a cutoff. Consistently, the result showed that these pathways, including cellular response to chemical stress (FDR= 0.0013), autophagy (FDR = 0.028), neutrophil degranulation (FDR = 0.039), innate immune system (FDR =0.0013), TP53 regulates metabolic genes (FDR = 0.0195), and T cell receptor signaling pathway (FDR = 0.0009), were also enriched in G-I subtype (**Figure RL7A, see also Figure S3D in the revision**). Therefore, the results the featured pathways up-regulated in G-III subtype showed a well consistency through two methods (both ssGSEA and GSEA).

(2) As for the differential analysis of the featured pathways up-regulated in G-I subtype between S/NS groups, as well as RRAS high/low expression groups

Among the proteomic subtypes, pathway enrichment analysis revealed G-I subtype was featured with PPAR-alpha pathway, integrin-linked kinase signaling, metabolism of vitamins and cofactors, extracellular matrix organization (ECM), RHO GTPase cycle and MAPK signaling pathway. In this section, to further determine the association of these signature pathways enriched in G-I subtype with treatment response, we performed ssGSEA analysis to obtain the gene set score corresponding to each sample. Further comparison of the ssGSEA pathway scores between S group and NS group revealed MAPK signaling, RHO GTPase cycle,

and ECM pathway were significantly increased in NS group, showing these signaling pathways activation might as the indicator of non-response to cetuximab therapy. Moreover, we also found the ssGSEA scores of the three pathways (MAPK signaling, RHO GTPase cycle, and ECM pathway) showed a significantly increase in the RRAS high expression group than in the RRAS low expression group. The differential analysis of ssGSEA score between S/NS groups, as well as RRAS high/low expression groups was according to the average score. *P*-value was calculated by two-sided Wilcoxon rank-sum test, *P* < 0.05 was considered statistically significant.

Furthermore, we also utilized traditional gene set enrichment analysis (GSEA) for pathway enrichment analysis. Sample grouped according to S/NS groups, as well as RRAS high/low expression groups were subjected to GSEA based on the proteomic data. An FDR value of 0.05 was used as a cutoff. The result showed that MAPK signaling, RHO GTPase cycle, and ECM pathway were significantly enriched in NS group, as well as the RRAS high expression group, with an FDR value less than 0.05 (**Figures RL7B and 7C, see also Figures 4D and 4G in the revision**). These findings revealed by GSEA demonstrated a consistency to the results uncovered in the comparison of ssGSEA pathway scores between S/NS groups, as well as RRAS high/low expression groups.

Finally, in the revision, we clarified the statistical details related to ssGSEA in the part of “The single sample gene set enrichment analysis (ssGSEA)”, and also supplemented the “The Gene set enrichment analysis (GSEA)” in the **Method (line 920–938 in Page 31–32)**. The GSEA results were also added in the “**The potential molecular features and biomarkers for the initial response to cetuximab therapy**” in the “**Results**” of revised **Manuscript (line 340–344 in Page 12, line 414–417 in Page 14, line 431–434 in Page 15), Figure S3 and Figure 4 in the revision.**

Figure RL7. The GSEA enrichment analysis for the validation of pathways identified by ssGSEA. (A) GSEA showing the enrichment of cellular response to chemical stress, autophagy, neutrophil degranulation, innate immune system, TP53 regulates metabolic genes, and T cell receptor signaling pathways in G-III subtype compared with the other two subtypes. (B and C) GSEA showing the enrichment of MAPK signaling, RHO GTPase cycle, and ECM pathway in the NS group compared with S group, as well as RRAS high expression group compared with RRAS low expression group. FDR < 0.05 is considered statistically significant.

Q7. P-values are provided throughout the manuscript, but the methods suggest these are adjusted p-values. Pathway p-values appear to not be adjusted. Please distinguish between adjusted p-values (such as by indicating "padj" or similar in the text) and raw p-values.

Response to Q7:

Thanks for the comment. We apologize for the confusion caused by the use and description of *P*-values in our manuscript. We understand the importance of distinguishing between adjusted and raw *P*-values. In this study, *P*-values or adjusted *P*-values were included in the differential analysis of protein expressions and xCell scores, as well as the pathway enrichment analysis, to indicate the statistical significance. According to the reviewer's suggestions, we corrected the *P*-values for multiple testing using the Benjamini-Hochberg (BH) method to control the false positive rate by using the *qvalue* package in R. The adjusted *P*-values were used and annotated with "adj *P*-value". The adj *P*-value < 0.05 is considered statistically significant. As shown in **Figure RL8**, in the revision, the pathway enrichment results were updated from "*P*-value" to "adj *P*-value" in **Figure 1G**, **Figure 1H**, **Figure 3C**, **Figure 4B**, and **Figure S3C** (corresponding to **Figure 1F**, **Figure 1G**, **Figure S3C**, **Figure 4B**, and **Figure S4E** in the revision); the differential analysis of protein expressions and xCell scores were updated from "*P*-value" to "adj *P*-value" in **Figure 2B**, **Figure 3F**, **Figure 5E**, and **Figure S3B** (corresponding to **Figure 2B**, **Figure 3D**, **Figure 5D**, and **Figure S4D** in the revision). The BH adjusted *P*-values of these pathways/proteins/xCell scores still had statistical significance. In addition, in the **Figure 1F** related to the differential analysis of CRC patients and healthy controls, we actually regarded the adj *P*-values < 0.05 as the statistically significant; however, we mistakenly labeled the " $-\text{Log}_{10}(P\text{-value})$ " in the Y axis of this figure. In the revision, we corrected the annotation of " $-\text{Log}_{10}(P\text{-value})$ " as " $-\text{Log}_{10}(\text{adj } P\text{-value})$ " in **Figure 1F** (corresponding to **Figure 1E** in the revision). Taken together, according to the reviewer's comments, the pathways/proteins/xCell scores in the differential analysis were corrected by Benjamini-Hochberg method. Finally, we updated the adj *P*-values in the "**Manuscript**", "**Method**" (line 866–895 in Page 29–30, line 918–923 in Page 31, line 967–989 in Page 33), **Figures 1, 2, 3, 4, and 5**, and **Figures S3 and S4** (shown as follows), and **Tables S3, S4, and S5** in the revision, and ensured the consistency in the manuscript, figures and tables, thoroughly.

Figure RL8. The P -values with multiple testing using the Benjamini-Hochberg (BH) method were annotated with "adj P -value" in these figures and supplementary figures in the revision, including Figures 1F/1G/1H (Figures 1E/1F/1G in the revision), Figure 2B, Figures 3C/3F

(Figures S3C/3D in the revision), Figures S3B/S3C (Figures S4D/S4E in the revision), Figure 4B, Figure 5E (Figure 5D in the revision).

Q8. The biomarker analysis that identified COL12A1, THBS2, S100A8, and S100A9 seems fairly independent from the consensus clustering/subtyping analysis, and both of these analyses seem fairly independent from the longitudinal analysis in figure 5. Authors should do a better job at relating these findings to each other. Was there any common biology between these results? Did these results inform each other? Could the results be described in a more holistic way?

Response to Q8:

We appreciate the reviewer for the careful read and valuable suggestions to present our manuscript more coherent and logical. According to the reviewer's comments, we supplemented more analysis results and transition statements to integrate our findings revealed in this study. To answer the reviewer's questions clearly, we divided the response into 6 parts as follows: (1) the diagnostic biomarkers to distinguish CRC patients from healthy controls; (2) the possible application of the four diagnostic biomarkers in predicting the therapy response; (3) the potential resistant/sensitive mechanism uncovered by consensus clustering analysis; (4) the possible association of the key proteins involved in the potential resistant/sensitive mechanism in the longitudinal analysis; (5) the prediction models applied for the continuous multiple courses of anti-EGFR therapy in CRC revealed in longitudinal analysis; (6) the more holistic presentation of integrating major findings and clinical characteristics.

(1) As for the diagnostic biomarkers of distinguishing CRC patients from healthy controls

In this study, to search for plasma proteins that could be used as diagnostic biomarkers for CRC patients, we used the following three criteria in this study: 1) The candidate proteins were expressed in at least 50% of the samples (1,359 proteins); 2) The candidates were significantly increased in tumor samples than normal samples (two-sided Wilcoxon rank-sum test, adj *P*-value < 0.01; 234 proteins); 3) The candidates were identified with at least 2-fold increase in CRC samples than normal samples. As a result, 148 proteins were screened, and significantly

and stably overexpressed in the plasma of CRC patients. To further validate whether the candidates identified in plasma were also commonly in tissue samples from CRC patients, we applied the same criteria in the pairwise comparison between tumor tissue and matched non-tumor adjacent tissue proteomic data of CRC patients from the Office of Cancer Clinical Proteomics Research (CPTAC) cohort, which resulted in 31 proteins that were significantly overexpressed in tumor tissues of CRC patients. Ultimately, we obtained a group of proteins, COL12A1, THBS2, S100A8, and S100A9, validated both in plasma and tissues of CRC patients. Further receiver operating characteristic (ROC) analysis determined a well performance of distinguishing CRC patients from healthy controls in our plasma discovery cohort.

To validate the prediction effect of the four proteins, in the revision, we included more CRC patients in the plasma validation cohort composed of 31 CRC patients and 24 healthy controls for the proteomic measurement. Consistently, the differential proteomic analysis revealed that COL12A1, THBS2, S100A8, and S100A9 were significantly up-regulated in the plasma samples from CRC patients in the plasma validation cohort. We further performed the receiver operating characteristic (ROC) analysis to evaluate the predictive effect. The ROC analysis revealed the combination of the four proteins achieved a good performance with an AUC at least 0.952 in the plasma validation cohort (**Figure RL1**).

To further reveal the association of the expression of these biomarkers in plasma and tumor tissues, in the revision, we reviewed all the archival formalin-fixed paraffin-embedded (FFPE) tissues from the therapy-naïve CRC patients included in this study. Finally, we collected 31 tumor tissues and 27 paired normal-adjacent tissues (NATs) of CRC patients matched with the plasma samples for mass spectrometry (MS)-based proteomic profiling. Proteomic analysis of tumor tissues identified a total of 9,258 proteins, of which 6,927 proteins were detected in plasma. The comparative proteomic analysis demonstrated that 2,573 proteins were significantly elevated in tumor tissues; while 251 proteins were down-regulated in tumor tissues (fold change (Tumor vs NAT) >2, adj *P*-value < 0.05). In the previous version, based on the plasma proteome profiling, the comparative analysis identified 745 proteins up-regulated in the CRC patients; among of these proteins, 235 (31.54%) proteins were up-regulated in tumor

tissues from CRC patients. Interestingly, this result showed a relatively high overlapped proportion of the differential proteins up-regulated in both the plasma samples and tumor tissue samples of the CRC patients. Evidently, COL12A1 (FC = 4.29, adj *P*-value = 8.8E-4), THBS2 (FC = 3.43, adj *P*-value = 1.6E-4), S100A8 (FC = 3.40, adj *P*-value = 8.7E-5), and S100A9 (FC = 5.10, adj *P*-value = 1.2E-4) showed a significant increase in tumor tissues. Furthermore, we validated the diagnostic efficiency of the four proteins in the tissues of the CRC cohort. The receiver operating characteristic (ROC) analysis revealed the four proteins combined prediction achieved good performance with AUC value of 0.945 in the tissue validation cohort (**Figure RL2**). Overall, the refined biomarkers could well distinguish the CRC patients from healthy controls both in plasma discovery cohort and plasma/tissue validation cohort.

(2) As for the possible application of the diagnostic biomarkers in predicting the therapy response

To further confirm the potential possible application of these diagnostic biomarkers in predicting the therapy response, we explored the differential expression of the four biomarkers (COL12A1, THBS2, S100A8, and S100A9) between sensitive (S) and non-sensitive (NS) groups. As a result, these proteins showed no significant difference between S and NS groups. The ROC analysis also indicated that the combination of these proteins had a poor prediction in therapy response (**Figure RL9A, see also Figure S2G in the revision**). Taken together, these diagnostic biomarkers could not be regarded as indicators for the therapy response prediction. According to the reviewer's suggestion, we added the result in the supplementary figure, and supplemented the transition description in the revision.

(3) As for the potential biological mechanisms and biomarkers for the therapy response

To further explore the potential resistant/sensitive mechanisms and biomarkers for therapy response, we firstly applied consensus clustering analysis, an unsupervised clustering method, and preliminarily determined the association between proteome pattern and therapy response. In this study, we identified three subtypes featured with distinct clinical characteristics and biological functions. For example, the G-III subtype, as a sensitive subtype, was featured by the elevation of autophagy and immune response; while G-I subtype, as a non-sensitive

subtype, was characterized by extracellular matrix organization (ECM) and MAPK signaling pathway. Then, we validated the alteration of these overrepresented pathways and the key regulators in the sensitive and non-sensitive groups. Finally, we identified a group of proteins significantly associated with clinical prognosis, including RRAS2, MMP8, FBLN1, RPTOR, and IMPDH2. The ROC analysis revealed the combination predictive model composed of RRAS2, MMP8, FBLN1, RPTOR, and IMPDH2, yielded a high prediction with AUC values of 0.849, in distinguishing the sensitive patients from the non-sensitive patients receiving cetuximab therapy in the plasma discovery cohort. To validate the predictive effect of this predictive model for the initial therapy response, in the revision, we included two independent cohorts: tissue validation cohort and plasma validation cohort in the revision. The therapy-naïve CRC patients of the two independent cohorts were grouped into sensitive (S) and non-sensitive (NS) groups according to the RECIST 1.1 guideline. We collected the plasma/tissue samples from the therapy-naïve CRC patients for the proteomic measurement. ROC analysis revealed that the combination predictive model composed of RRAS2, MMP8, FBLN1, RPTOR, and IMPDH2 had a good performance with AUC values of 0.890 and 0.816 in the independent plasma validation cohort and tissue validation cohort, demonstrating the stability and robustness of the predictive model for predicting response to the initial cetuximab treatment (**Figure RL9B, see also Figure 4L in the revision**).

(4) As for the possible association of the key proteins involved in the potential resistant/sensitive mechanism as well as the diagnostic biomarkers, in the longitudinal analysis

In this study, the plasma sampling covered a cetuximab therapy period of up to 72 weeks from the therapy-naïve sampling, of which the plasma from each patient were collected when completing each therapy course (8-week therapy), resulting in a total of 474 plasma samples from 116 CRC patients covering 9 sampling time points during the treatment period. In the revision, according to the reviewer's suggestion, we have taken the longitudinal characteristics of our cohort into consideration, and applied the diagnostic model and predictive model in the longitudinal cohort covering multi-course treatment. As a result, we found both the diagnostic model composed of the four biomarkers (COL12A1, THBS2, S100A8, and S100A9), and the

predictive model composed of the key proteins (RRAS2, MMP8, FBLN1, RPTOR, and IMPDH2) didn't achieve a good distinguish between sensitive and non-sensitive patients during the cetuximab treatment (**Figure RL9C and D, see also Figures S4A and S4B in the revision**).

(5) As for the prediction models applied for the multi-course of cetuximab therapy in longitudinal analysis

It is important to identify the protein dynamic change associated with the treatment response to cetuximab due to the drug resistance which could occur in any treatment course during the treatment. Therefore, to identify the protein dynamic change associated with the treatment response to cetuximab during multiple treatment courses, we focused on a longitudinal cohort composed of 22 CRC patients, 105 plasma samples, covering seven-course cetuximab treatment. In this section, we screened for a panel of proteins with significant negative correlation and up-regulated in the stable sensitive group, and a panel of proteins with significant positive correlation and up-regulated in the stable non-sensitive group. Having identified biomarkers with fluctuations at protein level associated with response to cetuximab therapy, we next set out to determine whether these biomarkers could be used for predicting response to cetuximab treatment in CRC patients during the continuous courses. We employed stepwise logistic regression and identified a subset of signatures (including IDH3G, MDN1, KLC4, MYL9, SBF1, and HTRA3) that accurately discriminates stable sensitive group and stable non-sensitive group. We further determined the ability of the predictive model for the response of cetuximab treatment across different courses. In the revision, we expanded our validation cohort, which composed of 31 CRC patients, 77 plasma samples, covering two-course cetuximab treatment according to the reviewer's suggestion. We validated the prediction performance of this subset of signatures in the plasma validation cohort. The results of confusion matrix analyses exhibited an accuracy of 0.929 (95%CI: 0.765–0.991) in the first course treatment, and 1 (95%CI: 0.815–1) in the second course treatment in the plasma validation cohort (**Figure RL9E, see also Figure 5G in the revision**). Taken together, the predictive model had high accuracy prediction performance in different treatment course, , which were further validated in the independent cohort.

In a summary, this study identified the different biomarkers for the tumor diagnosis, the initial response prediction of the first treatment, as well as the longitudinal response prediction of the multi-course treatment, revealing the alteration of the biological function in different management stages of the tumor, as well as the tumor heterogeneity.

(6) As for the more holistic presentation of integrating major findings, clinical cohort as well as clinical characteristics

We really appreciate the reviewer for the suggestion, which help to improve the quality of this manuscript and provide a better understanding for readers. According to the reviewer's suggestion, we (i) added the possible application of the diagnostic biomarkers in predicting the therapy response (**the second part**) followed the diagnostic biomarkers (**the first part**), (ii) added the possible application of the key proteins involved in the potential resistant/sensitive mechanism (**the third part**), as well the diagnostic biomarkers (**the first part**), in the following longitudinal analysis covering continuous multiple courses (**the fourth part**), (iii) added a summary about the heterogeneity of different management stages of the tumor, including tumor diagnosis, the initial response prediction of the first treatment, as well as the longitudinal response prediction of the multi-course treatment in the revision (**the fifth part**). In addition, we integrated major findings, clinical cohorts, as well as clinical characteristics, and provided a comprehensive diagram showing the connection of each result as follows (**Figure RL9F, see also Figure S5 in the revision**). Finally, we updated all the results in the figures (**Figures 4 and 5**), supplementary figures (**Figures S2, S4, and S5**), and manuscript (**line 292–308 in Page 10–11, line 444–463 in Page 15–16, line 467–485 in Page 16–17, line 602–607 in Page 21, and line 646–667 in Page 22–23**) in the revision.

Figure RL9. The potential application of the different biomarkers and the integration of our cohort, data analysis, and major findings of this study. (A) The ROC curves of diagnostic biomarkers to distinguish sensitive patients from non-sensitive patients in the first cetuximab treatment. (B) The validation of prediction effect of the biomarkers for predicting the

therapy response in the independent tissue and plasma validation cohorts. (C and D) The prediction effect of the biomarkers for the CRC diagnosis and biomarkers for the therapy response in the CRC longitudinal cohort. (E) Classification error matrix using logistic regression classifier in distinguishing S and NS in the plasma longitudinal validation cohort. The number of samples identified is noted in each box. (F) The diagram summarizing the connection of clinical samples, data analysis, and major findings.

Reviewer #2 (Remarks to the Author): expertise in mass-spectrometry based biomarker analysis

In this manuscript, Li et al. performed an impressive plasma proteomics dataset of 580 plasma samples from the discovery cohort composed of 116 CRC patients undergoing anti-EGFR therapy with continuous multiple treatment courses and 66 healthy controls (HCs), as well as the validation cohort composed of 16 CRC patients and 24 HCs. The tumour-specific biomarkers were inferred and validated by other public cohorts, which also exhibited a well performance in the independent cohort. They identified three proteomic-based subtypes characterized with different clinical therapy responses and molecular features. In the longitudinal trajectories of a cohort covering multi-course cetuximab treatment, the authors explored proteome alterations in disease progression and identify biomarkers used for dynamic monitoring in the continuous treatments. The biomarkers were then validated by PRM.

The authors have performed a large amount of informatics analyses, exploring novel insights into the biology and identifying novel biomarkers for predicting cetuximab treatment response of CRC patients. Moreover, the dataset itself, which is deposited in public repositories, will serve as a rich resource for the CRC research community. Overall, it is a good study. However, there are some questions that require clarification before the manuscript would be acceptable for publication.

Response:

Thank you very much for the concise summary of our work. We thank the reviewer for the careful read and thoughtful comments on our manuscript. According to reviewer's comments, we made corresponding revision to address all the points as follows.

Major comments:

Q9. The authors screened four proteins in Figure 2A for CRC diagnosis in Fudan cohort and other cohorts, and validated them in an independent cohort. Then, whether these proteins also associated with CRC cetuximab therapy or only for CRC diagnosis?

Response to Q9:

Thanks for the comment. According to reviewer's suggestions, we divided the response into four parts to answer: (1) The screening criteria of diagnostic biomarkers for CRC patients; (2) The process of screening key signatures related to the resistant/sensitive mechanism for initial response prediction; (3) The process of screening predictive biomarkers for the longitudinal response prediction of the multi-course treatment; (4) The relationship between the diagnostic biomarkers and therapy response of CRC patients with cetuximab.

(1) The screening criteria of diagnostic biomarkers for CRC patients

To search for plasma proteins that could be used as diagnostic biomarkers for CRC patients, we used the following four criteria in this study: 1) The candidate proteins were expressed in at least 50% of the samples; 2) The candidates were significantly increased in tumor samples than normal samples (two-sided Wilcoxon rank-sum test, adj P -value < 0.01); 3) The candidates were identified with at least 2-fold increase in CRC samples than normal samples; 4) The same criteria was applied in the pairwise comparison between tumor tissue and matched non-tumor adjacent tissue proteomic data of CRC patients from the Office of Cancer Clinical Proteomics Research (CPTAC) cohort (*Cell*, 2019, PMID: 31031003). Ultimately, we obtained a group of proteins, COL12A1, THBS2, S100A8, and S100A9, which showed significant elevation in the plasma samples and tumor tissues from CRC patients compared with healthy controls. ROC analysis demonstrated the predictive model composed of the four proteins could achieve high accuracy for CRC diagnosis.

In the revision, to further validate the prediction effect of the four proteins composed model, we included more CRC patients in the plasma validation cohort composed of 31 CRC patients and 24 healthy controls for the proteomic measurement. Consistently, the differential proteomic analysis revealed that COL12A1 (FC = 4.25, adj *P*-value = 0.017), THBS2 (FC = 2.30, adj *P*-value = 0.034), S100A8 (FC = 2.34, adj *P*-value = 0.013), and S100A9 (FC = 28.34, adj *P*-value = 1.2E-7) were significantly up-regulated in the plasma samples from CRC patients in the plasma validation cohort (**Figure RL10A, see also Figure S2F in the revision**). We further performed the receiver operating characteristic (ROC) analysis to evaluate the predictive effect. The ROC analysis revealed the four proteins combined prediction achieved good performance with an AUC of 0.952 in the plasma validation cohort (**Figure RL10B, see also Figure 2G in the revision**).

In addition, we also collected 31 tumor tissues and 27 paired normal-adjacent tissues (NATs) of CRC patients matched with the plasma samples for mass spectrometry (MS)-based proteomic profiling. Proteomic analysis identified 2,573 proteins were significantly elevated in tumor tissues; while 251 proteins were down-regulated in tumor tissues (fold change (Tumor vs NAT) >2, adj *P*-value < 0.05). Evidently, COL12A1 (FC = 4.29, adj *P*-value = 8.8E-4), THBS2 (FC = 3.43, adj *P*-value = 1.6E-4), S100A8 (FC = 3.40, adj *P*-value = 8.7E-5), and S100A9 (FC = 5.10, adj *P*-value = 1.2E-4) showed a significant increase in tumor tissues (**Figure RL10C, see also Figure S2E in the revision**). Furthermore, we validated the diagnostic efficiency of the four proteins in the tissues of the CRC cohort. The receiver operating characteristic (ROC) analysis revealed the four proteins combined prediction achieved good performance with an AUC of 0.945 in the tumor tissues (**Figure RL10D, see also Figure 2G in the revision**). Therefore, these results demonstrated the four diagnostic biomarkers, including COL12A1, THBS2, S100A8 and S100A9, could achieve a high prediction in distinguishing CRC patients from healthy controls, both in plasma and tissue validation cohort. Finally, we appreciate the reviewer for the constructive suggestions.

(2) The process of screening key signatures related to the resistant/sensitive mechanism for initial response prediction

To further explore the potential resistant/sensitive mechanism and biomarkers for therapy response, we firstly applied consensus clustering analysis, an unsupervised clustering method, and preliminarily determined the association between proteome pattern and therapy response. In this study, we identified three subtypes featured with distinct clinical characteristics and biological functions. For example, the G-III subtype, as a sensitive subtype, was featured by the elevation of autophagy and immune response; while G-I subtype, as a non-sensitive subtype, was characterized by extracellular matrix organization (ECM) and MAPK signaling pathway. We identified key proteins involved in these overrepresented pathways, including RRAS2, MMP8, FBLN1, RPTOR, and IMPDH2, which yielded a high prediction with AUC values of 0.849 in distinguishing the sensitive patients from the non-sensitive patients receiving cetuximab therapy in the plasma discovery cohort. To validate the potential association of these key proteins with the response, we included two independent cohorts: tissue validation cohort and plasma validation cohort in the revision. The therapy-naïve CRC patients of the two independent cohorts were grouped into sensitive (S) and non-sensitive (NS) groups according to the RECIST 1.1 guideline. There were 12 S patients and 19 NS patients included in the independent tissue validation cohort, as well as 16 S patients and 15 NS patients included in the independent plasma validation cohort. We collected the plasma/tissue samples from therapy-naïve CRC patients for the proteomic measurement. ROC analysis revealed that the combination predictive model composed of RRAS2, MMP8, FBLN1, RPTOR, and IMPDH2 had a good performance with AUC values of 0.890 and 0.816 in the independent plasma validation cohort and tissue validation cohort, demonstrating the stability and robustness of the predictive model for predicting response to first cetuximab treatment (**Figure RL10E, see also Figure 4L in the revision**).

(3) The process of screening predictive biomarkers for the longitudinal response prediction of the multi-course treatment

To explore the plasma biomarkers for efficient monitoring during the continuous multiple courses to cetuximab treatment, we used the following these criteria in this study: 1) The

plasma proteins were significantly altered during multiple cetuximab treatment courses screened by significant positive or negative correlation, defined as positive correlation-sig or negative correlation-sig; 2) The candidates were significantly differential expression in the stable sensitive and non-sensitive group (fold changes > 1.5 or < 0.67; two-sided Wilcoxon rank-sum test, adj *P*-value < 0.05), defined as SSG-sig and SNSG-sig; 3) The overlapped proteins of SSG-sig and negative correlation-sig, as well as SNSG-sig and positive correlation-sig were regarded as the candidate biomarkers for the further stepwise logistic regression. The backward stepwise method was utilized to feature selection. Ultimately, we obtained a plasma-protein panel, the S-sig includes IDH3G, MDN1, and KLC4, while the NS-sig includes MYL9, SBF1, and HTRA3. We further determined the ability of the predictive model for the response of cetuximab treatment across different courses. In the revision, we expanded our validation cohort, which composed of 31 CRC patients, 77 plasma samples, covering two-course cetuximab treatment according to the reviewer's suggestion. We validated the prediction performance of this subset of signatures in the plasma validation cohort. The results of confusion matrix analyses exhibited an accuracy of 0.929 (95%CI: 0.765–0.991) in the first course treatment, and 1 (95%CI: 0.815–1) in the second course treatment in the plasma validation cohort (**Figure RL10F, see also Figure 5G in the revision**). Taken together, the predictive model had high accuracy in the multi-course treatment period, which were further validated in the independent longitudinal cohort.

(4) The relationship between the diagnostic biomarkers and therapy response of CRC patients with cetuximab

According to reviewer's suggestion, to further confirm the potential possible application of these diagnostic biomarkers in predicting the therapy response, we explored the difference of the four biomarkers (COL12A1, THBS2, S100A8, and S100A9) between S and NS groups. The ROC analysis also indicated that the combination of these proteins had a poor prediction in the initial therapy response of the first treatment (**Figure RL10G, see also Figure S2G in the revision**). In addition, we also applied the diagnostic model in the longitudinal cohort covering multi-course treatment to examine the longitudinal response prediction. As a result, we found these diagnostic biomarkers could not well distinguish the sensitive patients from non-sensitive

patients in the multi-course treatment (**Figure RL10H, see also Figure S4A in the revision**), suggesting these diagnostic biomarkers could not be regarded as indicators for the therapy response prediction. According to the reviewer's suggestion, we added the result in the supplementary figure, and supplemented the transition description before the consensus clustering analysis.

Finally, we thank the reviewer's suggestion, and remind us thinking about the association of diagnostic marker and therapy response. In the revision, we updated these results in the figures (**Figures 2, 4, and 5 in the revision**) and supplementary figures (**Figures S2 and S4 in the revision**), as well as the corresponding parts in the "**Result**" (**line 270–308 in Page 9–11, line 444–485 in Page 15–17, line 541–545 in Page 19**).

Figure RL10. The prediction effect of different biomarkers for diagnosis, initial response prediction, and longitudinal response prediction, as well as the application of diagnostic biomarkers in the prediction of therapy response. (A) The boxplot showing the differential expression of the four diagnostic biomarkers between the CRC and HC groups in the plasma

validation cohort (two-sided Wilcoxon rank-sum test, adj P -value < 0.05). Boxplots show median (central line), upper and lower quartiles (box limits), 1.5×interquartile range (whiskers). (B) The ROC curves of diagnostic biomarkers to distinguish CRC patients from healthy controls in the plasma discovery and validation cohorts. (C) The boxplot showing the differential expression of the four diagnostic biomarkers between the T and NAT groups in the tissue validation cohort (two-sided Wilcoxon rank-sum test, adj P -value < 0.05). Boxplots show median (central line), upper and lower quartiles (box limits), 1.5×interquartile range (whiskers). (D) The ROC curve of diagnostic biomarkers to distinguish tumor from NAT in the tissue validation cohort. (E) The ROC curves of the predictive markers for initial response prediction in the discovery and validation cohorts. (F) Classification error matrix using logistic regression classifier in distinguishing S and NS in the plasma longitudinal validation cohort. The number of samples identified is noted in each box. (G and H) The prediction effect of the diagnostic biomarkers in predicting the initial therapy response (G), and the longitudinal therapy response (H).

Q10. This study identified proteomic subtypes featured with distinct biological features and therapy response. This significant association of proteomic subtypes and therapy response were shown in Figure 3B. What about other clinical variables? The authors should also make a statistic analysis of other clinical variables, and included these results in the supplementary figure.

Response to Q10:

Thanks for the comment. We apologize for not showing the association of proteomic subtypes with other clinical characteristics. In this study, we collected the baseline clinical characteristics of CRC patients, including age, gender, degree of tumor differentiation, Eastern Cooperative Oncology Group (ECOG) performance-status score, and other biochemical indicators (such as lactate dehydrogenase (LDH) level, white blood cell (WBC) count, lymphocyte number (LYMPHN), hemoglobin (HB), and platelet count). Actually, we simultaneously performed Fisher's exact test on the categorical variables and one way-ANOVA analysis on the continuous variables among three subtypes (G-I, G-II, and G-III). As a result, we found an obvious

association between proteomic subtypes and therapy response (Fisher’s exact test, $P = 0.014$), but this association was not observed with either grade, degree of tumor differentiation, ECOG, or any other biochemical indicators ($P > 0.05$) (Table RL2, see also Table 2 in the revision). These results demonstrated that the proteomic subtype associated with therapy response but not with other clinical variables.

In the revision, we have added the result in the part of “The potential molecular features and biomarkers for the initial therapy response” in the “Result” (line 315–319 in Page 11). Meanwhile, we have updated the result in Table 2 in the revision.

	G-I (N=24)	G-II (N=34)	G-III (N=31)	P-value
Age (years), median (range)	53.5 (24-69)	55 (21-74)	61 (25-76)	0.287 ^a
Gender				0.571 ^b
Female	7 (29.2%)	8 (23.5%)	11 (35.5%)	
Male	17 (70.8%)	26 (76.5%)	20 (64.5%)	
Degree of tumor differentiation				0.232 ^b
Poorly differentiated	7 (29.2%)	12 (35.3%)	5 (16.1%)	
Moderately differentiated	11 (45.8%)	16 (47.1%)	14 (45.2%)	
Well differentiated	0 (0.00%)	1 (2.94%)	0 (0.00%)	
NA	6 (25.0%)	5 (14.7%)	12 (38.7%)	
ECOG performance status				0.187 ^b
1	22 (91.7%)	34 (100%)	29 (93.5%)	
0	2 (8.33%)	0 (0.00%)	2 (6.45%)	
Lactate dehydrogenase level (U/L), median (range)	232.5 (126-1,121)	330 (135-3,000)	326 (119-3,000)	0.190 ^a
White Blood Cell count ($10^9/L$), median (range)	5.5 (2.5-14.8)	5.95 (1.7-14.7)	5.8 (2.8-10.5)	0.227 ^a
Lymphocyte number ($10^9/L$), median (range)	1.35 (0.4-2.2)	1.4 (0.6-2.2)	1.4 (0.3-3.1)	0.502 ^a
Hemoglobin(g/L), median (range)	131 (88-155)	120.5 (75-163)	124 (39-163)	0.149 ^a
Platelet Count ($10^9/L$), median (range)	171.5 (48-366)	210.5 (107-452)	216 (102-313)	0.069 ^a

a, one-way ANOVA analysis was applied for continuous variables. b, fisher’s exact test was used for categorical variables.

Q11. Proteomic subtypes identified G-I as non-sensitive subtype and G-III as sensitive subtype, exploring the potential biological alteration. As for the G-II subtype, the pathway enrichment result and association with response should be further investigated.

Response to Q11:

Thanks for the comment. We apologized for not providing the pathway enrichment results of G-II subtype. We agree with the reviewer that pathway enrichment result and association with response should be further investigated and included in this manuscript. In this section, we determined the signature proteins among the three proteomic subtypes: 313 (G-I), 403 (G-II),

and 416 (G-III) GPs (adj *P*-value < 0.05, fold change > 1.5). Based on these differential proteins, we performed a functional enrichment analysis according to the ConsensusPathDB (CPDB) molecular interaction data obtained from 31 different public repositories (*Nucleic Acids Res.*, 2022, PMID: 34850110), and further determined the dominant bioprocesses of each subtype. In the last version, we mainly focused on the G-I and G-III subtypes, which showed association with therapy response; and ignored the biological features of G-II subtype. Here, according to the reviewer's suggestion, we provided the pathway enrichment result of G-II subtype in the revision. The G-II subtype was featured by DNA Double Strand Break Response (adj *P*-value = 1.21E-6), protein ubiquitination (adj *P*-value = 2.60E-5), protein processing in endoplasmic reticulum (adj *P*-value = 2.60E-5), spliceosome (adj *P*-value = 7.82E-4), hemostasis (adj *P*-value = 7.82E-4), and ribosome (adj *P*-value = 4.71E-2) (**Figure RL11A, see also Figure S3C in the revision**). We further investigated the association of proteins involved in the dominant pathways of G-II subtype and therapy response. We calculated the fold change and adjusted *P*-values (adj *P*-value) of these proteins. Evidently, we found that there was no significant difference between sensitive group and non-sensitive group (adj *P*-value > 0.05) (**Figure RL11B**).

In the revision, we updated the enrichment result in the part of **“The potential molecular features and biomarkers for the initial response to cetuximab therapy”** in the **“Result”** (line 325–341 in Page 11–12). Meanwhile, we have added the result in **Figure S3C in the revision**.

Figure RL11. The G-II dominant pathways and association with therapy response. (A) Bubble plot showing the CPDB pathway enrichment (two-sided Fisher's exact test) of G-I to G-III subtypes. **(B)** Volcano showing no significant difference of molecules involving in the G-II dominant pathways between sensitive group and non-sensitive group (two-sided Wilcoxon rank-sum test), $\text{adj } P < 0.05$ is considered statistically significant.

Q12. The process for biomarker identification and predictive modeling should be provided detailed in the Method section.

Response to Q12:

Thanks for the comment. According to the reviewer's suggestion, we added the detailed process for biomarker identification and predictive modeling in the revision. In our study, there are two parts involving biomarker identification and predictive modeling, respectively for CRC diagnosis and therapy response prediction. Here, we divided the response into two parts to answer: (1) The screening criteria of diagnostic biomarkers and model construction for CRC diagnosis; (2) The process of identifying biomarkers for the initial response prediction of the first treatment; (3) The process of screening predictive biomarkers for the longitudinal response prediction of the multi-course treatment.

(1) The screening criteria of diagnostic biomarkers and model construction for CRC diagnosis

To search for plasma proteins that could be used as diagnostic biomarkers for CRC patients, we used the following four criteria in this study: 1) The candidate proteins were expressed in at least 50% of the samples; 2) The candidates were significantly increased in tumor samples than normal samples (two-sided Wilcoxon rank-sum test, adj P -value < 0.01); 3) The candidates were identified with at least 2-fold increase in CRC samples than normal samples; 4) The same criteria was applied in the pairwise comparison between tumor tissue and matched non-tumor adjacent tissue proteomic data of CRC patients from the Office of Cancer Clinical Proteomics Research (CPTAC) cohort (*Cell*, 2019, PMID: 31031003). Ultimately, we obtained a group of proteins, COL12A1, THBS2, S100A8, and S100A9, which showed significant elevation in the plasma samples and tumor tissues from CRC patients compared with healthy controls. ROC analysis using (pROC R package version 1.16.2 and Caret R package version 6.0–86) demonstrated the predictive model composed of the four proteins could achieve high accuracy for CRC diagnosis. Furthermore, the combination of the four proteins could also reach a good performance with AUC values of 0.952 and 0.945 in the plasma validation cohort and tissue validation cohort, respectively, in the revision (**Figure RL10B and 10D, see also Figure 2G in the revision**), indicating the robustness of the four proteins in plasma and tissues.

In the revision, considering the plasma-matched tissue samples included in this study, to search for the potential proteins applied for plasma and tissue samples that could be used as diagnostic biomarkers for CRC patients, we added our tissue validation cohort composed of 31 tumor tissues and 27 paired normal-adjacent tissues (NATs) of CRC patients matched with our plasma samples for the biomarker discovery. We adopted the same strict screening strategy in the tissue samples as follows: 1) The candidate proteins were expressed in at least 50% of the samples; 2) The candidates were significantly increased in tumor samples than normal samples (two-sided Wilcoxon rank-sum test, adj P -value < 0.01); 3) The candidates were identified with at least 2-fold increase in CRC samples than NATs. Combined our independent plasma discovery cohort, tissue validation cohort, and the CPTAC cohort, finally, we obtained a group of four proteins, COL12A1, THBS2, S100A8, and S100A9, significantly increased both in

plasma and tissues of CRC patients in our study and the CPTAC study. In the revision, we updated the section of “**Screening biomarkers for CRC diagnosis**” in the “**Method**” as follows.

“The plasma and tissue samples from therapy-naïve CRC patients and healthy controls (or NATs) were included for screening diagnostic biomarkers for CRC patients. To search for potential signatures applied for plasma and tissue samples that could be used as diagnostic biomarkers for CRC patients, we used the following criteria in this study: 1) The candidate proteins were expressed in at least 50% of the samples; 2) The candidates were significantly increased in tumor samples than normal samples or (normal-adjacent tissues) NATs (two-sided Wilcoxon rank-sum test, adj *P*-value < 0.01); 3) The candidates were identified with at least 2-fold increase in CRC samples than normal samples or NATs. The same screening criteria were adopted in our tissue validation cohort and the CPTAC CRC cohort. Ultimately, we obtained four biomarkers (including COL12A1, THBS2, S100A8, and S100A9) for CRC diagnosis. The diagnostic effect of the four biomarkers combination was verified by ROC analysis using (pROC R package version 1.16.2 and Caret R package version 6.0–86), in which sensitivity, specificity, accuracy, and AUC were used to determine predictive values. The 10-fold cross validation was used, and samples were partitioned based on sample types; among them, 60% of samples were used as a training set, and the remaining 40% represented the independent testing set.”
(line 880–895 in Page 30).

(2) The process of identifying biomarkers for the initial response prediction of the first treatment

To further explore the potential resistant/sensitive mechanism and biomarkers for therapy response, we firstly applied consensus clustering analysis, an unsupervised clustering method, and preliminarily determined the association between proteome pattern and therapy response. In this study, we identified three subtypes featured with distinct clinical characteristics and biological functions. For example, the G-III subtype, as a sensitive subtype, was featured by the elevation of autophagy and immune response; while G-I subtype, as a non-sensitive subtype, was characterized by extracellular matrix organization (ECM) and MAPK signaling

pathway. We identified key proteins involved in these overrepresented pathways, including RRAS2, MMP8, FBLN1, RPTOR, and IMPDH2, yielded a high prediction with AUC values of 0.804 in the training set and 0.955 in the testing set, in distinguishing the sensitive patients from the non-sensitive patients with receiving cetuximab therapy. To validate the potential association of these key proteins with the response, in the revision, we grouped these therapy-naïve CRC patients into sensitive (S) and non-sensitive (NS) groups according to the RECIST 1.1 guideline. There were 12 S patients and 19 NS patients included in the tissue validation cohort, as well as 16 S patients and 15 NS patients included in the plasma validation cohort. We collected the plasma/tissue samples from therapy-naïve CRC patients for the proteomic measurement. ROC analysis revealed that the combination predictive model composed of RRAS2, MMP8, FBLN1, RPTOR, and IMPDH2 had a good performance with AUC values of 0.890 and 0.816 in the independent plasma validation cohort and tissue validation cohort, demonstrating the stability and robustness of the predictive model for predicting response to cetuximab first-time treatment (**Figure RL10E, see also Figure 4L in the revision**). In the revision, we updated the section of **“The potential molecular features and biomarkers for the initial therapy response”** in the **“Method”** as follows.

“The protein expression matrix of the plasma samples from therapy-naïve CRC patients was used to identify the proteomic subtypes using the consensus clustering method implemented in the R package ConsensusClusterPlus v.3.8 (Bioinformatics., 2010, PMID: 20427518). The top 1,500 proteins with the highest median absolute deviation were subjected to ConsensusClusterPlus in R v.3.5.1 for unsupervised consensus clustering. The cluster analysis was performed with the following setting: maxK = 10, reps = 1,000, pltem = 0.8, pFeature = 1, clusterAlg = "hc", distance = "pearson" for the clustering runs. A preferred cluster result was selected by considering the profiles of the consensus cumulative distribution function (CDF) and delta area under the CDF curve for clustering solutions between 2 and 10 clusters. As shown in Figure S3, the rank survey profiles of the consensus CDF and the delta area under the CDF curve, along with the consensus membership heat maps, indicated a three-subtype solution for 89 samples of pre-treatment CRC using the proteomic data. This showed clear separation and the significant therapy response among 3 subtypes. To generate the abundance

heatmap, the CRC samples in each subtype were rearranged from G-I to G-III, using the signature protein abundance matrix enriched in the signature pathways for each subtype. The signature proteins of each subtype were defined with significantly differential expression (fold change > 1.5, adj *P*-value < 0.05, two-sided Wilcoxon rank-sum test) when compared with other subtypes. ConsensusPathDB (CPDB) molecular interaction data obtained from 31 different public repositories (Nucleic Acids Res., 2022, PMID: 34850110) and determined the dominant bioprocesses of each subtype. The adj *P*-value < 0.05 is considered statistically significant of the pathway enrichment. The key regulators involved in the overrepresented pathways associated with therapy response were identified. Among these proteins, a group of proteins with significant (positively or negatively) correlation with clinical prognosis were regarded as the potential biomarkers (including RRAS2, MMP8, FBLN1, RPTOR, and IMPDH2). The predictive effect of the five biomarkers combination was verified by ROC analysis using (pROC R package version 1.16.2 and Caret R package version 6.0–86), in which sensitivity, specificity, accuracy, and AUC were used to determine predictive values.” (line 903–930 in Page 31).

(3) The process of screening predictive biomarkers for the longitudinal response prediction of the multi-course treatment

In addition to the identification of diagnostic markers for CRC, in this study, it is more crucial to screen a group of biomarkers for the prediction for the therapy response, especially for the multi-course cetuximab treatment. To explore the plasma biomarkers for efficient monitoring during the continuous multiple courses to cetuximab treatment, we used the following these criteria in this study: 1) The plasma proteins were significantly altered during multiple cetuximab treatment courses screened by significant positive or negative correlation, defined as positive correlation-sig or negative correlation-sig; 2) The candidates were significantly differential expression in the stable sensitive and non-sensitive group (SSG and SNSG) (fold change (two-sided Wilcoxon rank-sum test, adj *P*-value < 0.05), defined as SSG-sig and SNSG-sig; 3) The overlapped proteins of SSG-sig and negative correlation-sig, as well as SNSG-sig and positive correlation-sig were regarded as the candidate biomarkers for the further stepwise logistic regression. Samples was randomly divided into 60% of individuals (the training set) and the remaining 40% (the testing set). The backward stepwise method was utilized to feature

selection on the training set. Moreover, the value of this model was verified using ROC analysis. Sensitivity, specificity, accuracy, and AUC were used to determine predictive values. Ultimately, we obtained a plasma-protein panel (IDH3G, MDN1, KLC4, MYL9, SBF1, and HTRA3), which could accurately discriminate sensitive patients and non-sensitive patients with AUC values of 0.756 in the training set and 0.797 in the testing set. The application of the predictive model to different treatment courses revealed an accuracy of 0.724 at least, indicating the good performance in the overall therapy courses. In the revision, we further determined the ability of the predictive model for the response of cetuximab treatment across different courses. We expanded our validation cohort, which composed of 31 CRC patients, 77 plasma samples, covering two course cetuximab treatment. We validated the prediction performance of this subset of signatures in the plasma validation cohort. The results of confusion matrix analyses exhibited an accuracy of 0.929 (95%CI: 0.765–0.991) in the first course treatment, and 1 (95%CI: 0.815–1) in the second course treatment in the plasma validation cohort (**Figure RL10F, see also Figure 5G in the revision**). Taken together, the predictive model had high accuracy in the multi-course treatment course, which were further validated in the independent cohort. In the revision, we updated the section of “**Identifying biomarkers for the longitudinal response prediction of the multi-course treatment**” in the “**Method**” as follows.

“To explore the plasma biomarkers for efficient monitoring during the continuous multiple courses to cetuximab treatment, we used the following these criteria in this study: 1) The plasma proteins were significantly altered during multiple cetuximab treatment courses screened by significant positive or negative correlation, defined as positive correlation-sig or negative correlation-sig; 2) The candidates were significantly differential expression in the stable sensitive and non-sensitive group (two-sided Wilcoxon rank-sum test, adj *P*-value < 0.05), defined as SSG-sig and SNSG-sig; 3) The overlapped proteins of SSG-sig and negative correlation-sig, as well as SNSG-sig and positive correlation-sig were regarded as the candidate biomarkers for the further stepwise logistic regression. Samples was randomly divided into 60% of individuals (the training set) and the remaining 40% (the testing set). The backward stepwise method was utilized to feature selection on the training set. Moreover, the value of this model was verified using ROC analysis. Sensitivity, specificity, accuracy, and AUC

were used to determine predictive values.” (line 975–989 in Page 33).

Finally, in the revision, we supplemented the descriptions in the part of “**Screening biomarkers for CRC diagnosis**” (line 880–895 in Page 30), “**The potential molecular features and biomarkers for the initial therapy response**” (line 903–930 in Page 31), and “**Identifying biomarkers for the longitudinal response prediction of the multi-course treatment**” (line 975–989 in Page 33) in the “**Method**”.

Q13. The authors mentioned “The overrepresented proteins of S and NS of cohort were defined as ...($NS/S > 2$ or < 0.5); the significantly differentially expressed proteins was defined as ...2-fold change and Wilcoxon rank-sum test with a Benjamini-Hochberg (BH) adjusted p value cutoff (BH p value < 0.05) in the Method section. However, the BH adjusted p values were not provided in the Table S5. In addition, in the differential analysis between SSG and SNSG groups, the authors defined the overrepresented proteins (ORPs) with 2-fold change with $P < 0.05$ (Line: 336-338); however, Table S5 included the DEPs. As for the differential analysis between CRC patients and healthy controls, the authors also mentioned the ORPs were proteins with 2-fold change with $P < 0.05$ (Line: 172-176). Overall, the authors should clarify “ORPs” and “DEPs” definition, and use uniform description in the manuscript.

Response to Q13:

Thanks for the comment. We apologized for not uniformly using "ORPs" and "DEPs" in the manuscript, figures and tables. As the reviewer mentioned, in the differential analysis of this study, the significantly differentially expressed proteins (DEPs) were defined with adj P -value < 0.05 (two-sided Wilcoxon rank-sum test with a Benjamini-Hochberg (BH) adjusted p value), in the comparison of plasma proteome in CRC patients and healthy controls, sensitive patients and non-sensitive patients, as well as the comparison of proteome in tumor tissues and normal-adjacent tissues, respectively. In the revision, we uniformly define and use "DEP" to represent the differential expressed proteins in the manuscript and supplementary tables. In addition, we also provided Benjamini-Hochberg (BH) adjusted p values (adj P -values) in **Tables S3, S4, and**

S5. Overall, we clarified and uniformed the definition of “DEPs” in the revised manuscript.

Q14. This study included a total of 580 plasma samples from the discovery cohort (composed of the well-characteristic CRC cohort composed of 116 CRC patients and 66 healthy controls) and validation cohort. The key biomarkers for diagnosis of CRC and response prediction of cetuximab therapy have been validated by various proteomic approaches, including data-independent acquisition (DIA) and parallel reaction monitoring (PRM), in the public datasets and independent cohort. Then, if any limitations in this study, the authors should have a discussion on study limitations. This is needed.

Response to Q14:

We appreciate the reviewer for the comment. According to the reviewer’s suggestion, we supplemented the discussion about the limitations of the study. In the revision, we added the part of “**Limitations of the study**” in the manuscript as following:

“In this study, we established the first longitudinal plasma proteome profiling of colorectal cancer to identify the effective diagnostic markers and predictive markers for cetuximab therapy, thus contributing to the monitoring and intervening in the treatment. Among the limitations to this study, first, this study is a single-center research likely does not represent all the heterogeneous mechanisms underlying clinical resistance to cetuximab therapy. The multi-center cohorts need to be included for such a study in the future. Second, there are around 1/6 of CRC patients but not all CRC patients receiving seven course cetuximab treatment (which reflects the real clinical world), the biomarkers for the response prediction of the multi-course treatment identified in this study should be validated in the more complete multi-course longitudinal cohort. Third, although the potential biomarkers for CRC diagnosis, were elevated both in plasma and tumor tissues of CRC patients, we couldn’ t completely exclude the contribution of other organs or tissues to the level of these proteins in plasma, which might need to be explored through comparison among more tissues. A more rigorous multi-tissue comparison research is worth studying in the future. Fourth, it is deserved to explore the components secreted by circulating immune cells associated with the therapy response in the

future.” (line 669–684 in Page 23).

Q15. Hazard ratios should be included for all KM plots.

Response to Q15:

Thanks for the comment. According to the reviewer’s suggestion, we added the hazard ratio (HR) result in the KM plots included in all survival analysis as shown in **Figure RL12**, and updated the description in the manuscript in the revision as following:

(1) “Among these proteins, RPTOR and IMPDH2 were significantly positive correlated with CD8+Tem both in plasma and tissue samples ($r \geq 0.23$, $P < 0.05$) (Figures 3J, S3G and S3H), and the high expression of RPTOR and IMPDH2 were significantly associated with better prognosis validated by CPTAC cohort (two-sided log rank test, $HR < 1$, $P < 0.05$) (Figure 3K).”

(line 399–403 in Page 14 in the revision)

(2) “Among proteins involved in these pathways, FBLN1, MMP8, and ITGA5 showed significantly positive correlation with RRAS/RRAS2, and the high expression of RRAS2, FBLN1 and MMP8 were significantly associated with poor prognosis in the in CPTAC cohort ($HR > 1$; two-sided log rank test, $P < 0.05$) (Figures 4I, 4J, and 4K).”

(line 438–442 in Page 15 in the revision)

In addition, we supplemented the detailed description in the “**Survival analysis**” of the “**Method**” (line 1018–1024 in Page 34–35 in the revision). Overall, we supplemented the analysis result in the part of “**The potential molecular features and biomarkers for the initial response to cetuximab therapy**” in the “**Result**”, and **Figures 3K and 4K** in the revision.

Figure RL12. The hazard ratio (HR) and Kaplan–Meier curves of OS in the CPTAC cohort based on the protein abundance (two-sided log rank test).

Q16. The targeted peptides of signature proteins were selected for targeted PRM analysis. However, the authors didn't provide the list of targeted peptides included in this study.

Response to Q16:

Thanks for the constructive comment. In this study, to validate the signature proteins identified by the data-independent acquisition (DIA) strategy, we performed parallel reaction monitoring (PRM), a powerful targeted approach to detect and quantify pre-specified proteins with a high throughput using high-resolution mass spectrometers (*Mol Cell Proteomics*, 2012, PMID: 22962056). We apologize for not providing the list of targeted peptides in the last version. According to the reviewer's suggestion, we supplemented the list of targeted peptides in **Table RL3 as follows**. In addition, in the revision, we included more CRC patients in the plasma validation cohort composed of 31 CRC patients, which were grouped into 16 sensitive (S) and 15 non-sensitive (NS) patients according to the RECIST 1.1 guideline. We collected the plasma samples from the therapy-naïve CRC patients for the PRM measurement. The PRM quantification revealed that RPTOR and IMPDH2 showed significant increase in the S group compared with NS group, validating the association of RPTOR and IMPDH2 with cetuximab sensitivity (**Figure RL13, see also Figure 3L in the revision**). Finally, in the revision, we updated the results in **Figure 3L**, and added the list of targeted peptides (**Table RL3**) in the **Table S4** in the revision.

Table RL3 Targeted peptides that unique to the signature proteins.

Protein	Sequence	m/z [Da]
IMPDH2	TSSAQVEGGVHSLHSYEKR	1036.52864
IMPDH2	KQLLCGAAIGTHEDDKYR	692.35682
RPTOR	GAGMVVDWEQETGLLMSSGDVR	785.02875
RPTOR	NPEMVTAWQGLSDMLPTTR	716.35677
RPTOR	IEGSKSLAQSWR	454.58245
RPTOR	MKDRMK	412.70374
RPTOR	VYDRRMALSECR	786.3723

Figure RL13. Boxplot showing RPTOR and IMPDH2 protein differential expression identified by PRM between S and NS groups in two cohorts. Discovery cohort: N (S) = 31, and N (NS) = 45; validation cohort, N (S) = 16, and N (NS) = 15. Boxplots show median (central line), upper and lower quartiles (box limits), 1.5×interquartile range (whiskers). $P < 0.05$ calculated by two-sided Wilcoxon rank-sum test is considered statistically significant.

Minor comments:

Q17. The therapy response information (S and NS) is very important, and associated with many results in this study. The S/NS information corresponding to each sampling time during the continuous treatment were provided in Table S5. However, the authors didn't provide the initial therapy response information for individual patients in Table S1. The authors should provide the information, which could be enable effective reuse of the dataset in future researches.

Response to Q17:

Thanks for the comment. We apologized for not providing the initial therapy response information. In the revision, we updated the initial therapy response information in “**Table S1**”, which could be effectively reused.

Q18. In the figures (e.g. Figure 2B, Figure 3B/E/G, Figure 4C/D), the statistical analysis marked the significance with * or p values, please provide the exact p values uniformly.

Response to Q18:

Thanks for the comment. We apologized for not providing the exact p values. In the revision, we replaced "*" with the exact p values in these figures, unifying the display format of statistical analysis results of all related figures (including **Figure 2B (Figure 2B in the revision)**, **Figure 3B, 3E, 3G (Figure 3A, 3C, 3E in the revision)**, **Figures 4C and 4D (Figures 4C and 4E in the revision)**). In addition, we also supplemented the exact *P*-values and adjusted *P*-values (annotated as adj *P*-values) in the supplementary tables, including **Tables S3, S4, and S5** in the revision.

Figure RL14. The statistical analysis marked the exact P values of these figures in the revision.

Q19. Minor grammar problems should be corrected. For example, Line: 265: "..., might resulting in an improved response to cetuximab therapy".

Response to Q19:

Thanks for the comment. We apologize for the grammar problem existed in the manuscript. We have carefully checked the manuscript to correct grammatical and spelling errors. The revised manuscript was also edited by professional native speakers. All changes were highlighted with red text in the revised manuscript. Specifically, as mentioned by the reviewer, we corrected this sentence in the revision as following:

"Overall, the high activation of autophagy and the aggregation of CD8+ Tem, might result in an

improved response to cetuximab therapy.” (line 408–409 in Page 14 in the revision).

Q20. Line 548-550: the authors described “The dynamic range of protein identification of each sample was shown according to the descending sort of protein abundance with a range of 1,587–2,502 proteins identified in each sample.”, the corresponding figure should be cited.

Response to Q20:

Thanks for the comment. In the revision, we accordingly rearranged subfigures, and added the figure about dynamic range of protein identification of each sample in **Figure RL15 (corresponding to Figure S1C in the revision)**. In addition, we also correctly updated the citation of this subfigure in the revised manuscript as following:

“The dynamic range of protein identification of each sample was shown according to the descending sort of protein abundance with a range of 1,587–2,502 proteins identified in each sample (Figure S1C).” (line 859–861 in Page 29 in the revision).

Figure RL15. The dynamic range of protein identification of each sample was shown according to the descending sort of protein abundance of each sample grouped into HC and CRC groups.

Q21. The heatmap of Figure 2B shows z-score transformation of protein expression?

The relative abundance should be annotated in figure legend.

Response to Q21:

Thanks for the comment. As mentioned by the reviewer, the heatmap of Figure 2B depicts the Z score transformation of protein expression of the four diagnostic biomarkers based on the plasma proteomic data in Fudan cohort and CPTAC cohort. In the revision, we supplemented the annotation of “Z score” under the bottom of the heatmap, and replaced the “*” (indicated the *P*-values) with the heatmap of the exact *P*-value (Log transformation of adj *P*-values) representing the statistical significance (**Figure RL16**). In addition, in the revision, we collected 31 tumor (T) tissues and 27 paired normal-adjacent tissues (NATs) of CRC patients matched with the plasma samples for mass spectrometry (MS)-based proteomic profiling. Differential analysis of T and NAT groups revealed a significant increase of COL12A1, THBS2, S100A8, and S100A9 in tumor tissues (**Figure RL16, see also Figure 2B in the revision**). Finally, we updated all the result in the **Figure 2B in the revision**.

Figure RL16. The heatmaps showing the relative abundance (Z score) of the diagnostic biomarkers in Fudan cohort (our plasma and tissue cohorts in this study), as well as the CPTAC CRC cohort. The little heatmaps and barplots showing the differential expression (two-sided Wilcoxon rank-sum test, adj *P*-value < 0.05).

Q22. In the legend of Figure 3A, the numbers of each subtype should be listed.

Response to Q22:

Thanks for the comment. In this study, we performed the consensus clustering analysis for the 89 therapy-naïve samples, which resulted in three subtypes: G-I (N = 24), G-II (N = 34), and G-

III (N = 31). According to the suggestion, in the revision, we added the numbers of each subtype in Figure 3A as well as its legend description (**Figure RL17, see also Figure S3B in the revision**) as following:

“(A) The heatmap depicts the relative abundance (Z score) of the signature proteins in three subtypes of CRC pre-treatment samples. N (G-I) = 24, N (G-II) = 34, and N (G-III) = 31. Biological functions related to these signature proteins are denoted on the right.”

Figure RL17. The heatmap depicts the relative abundance (Z score) of the signature proteins in three subtypes of CRC pre-treatment samples. N (G-I) = 24, N (G-II) = 34, and N (G-III) = 31. Biological functions related to these signature proteins are denoted on the right.

Q23. The description of Figure 5G is rough. The authors should provide more details corresponding to Figure 5G.

Response to Q23:

Thanks for the comment. We apologize for not providing the sufficient illustration for the Figure 5G. We updated a detailed statement of **Figure 5G (corresponding to Figure 5F in the revision)** in the part of “**Figure Legend**” in the revision (Please see **line 1444–1449 in Page 49**). The revised description of “**Figure Legends**” in the revision was shown as following:

“(G) The accuracy of the predictive model to predict the response of cetuximab treatment at different sampling times. “1-7” represented that all sampling covered the overall treatment

course were included; “2” meant that the second sampling after receiving two course treatments; “3” was defined as the third sampling after receiving three course treatments; “4” was defined as the fourth sampling after receiving four course treatments. The minimum accuracy of the predictive model was 0.724.”

Q24. All box-plot elements (center line, limits, whiskers, points) should be defined in the legends accompanied by precise n numbers, e.g. Figure 1D, Figure 3E/3G/3K, Figure 4C/4D.

Response to Q24:

Thanks for the comment. According to the reviewer’s suggestion, we added the box-plot elements (including median (central line), upper and lower quartiles (box limits), $1.5 \times$ interquartile range (whiskers)) and precise numbers in the legend description of **Figure 1D (corresponding to the Figure 1C in the revision), Figure 3E/3G/3K (corresponding to the Figure 3C/3E/3L in the revision), Figure 4C/4D (Figure 4C/4E in the revision)**. In the revision, we corrected in parts of “**Figure Legend**” as following:

Figure 1

“(D) Venn diagram showing the protein overlap of pre-treatment CRC and HC. The number of proteins is quantified in pre-treatment CRC patients (N = 89) and healthy controls (N = 66). Boxplots show median (central line), upper and lower quartiles (box limits), $1.5 \times$ interquartile range (whiskers). *P*-value is calculated by two-sided Student’s *t* test. $P < 0.05$ is considered statistically significant.” **(corresponding to the Figure 1C in the revision, line 1313–1317 in Page 44).**

Figure 3

“(E) Boxplot for Immune score evaluated by xCell among three proteomic subtypes. N (G-I) = 24, N (G-II) = 34, and N (G-III) = 31 (two-tailed Student’s *t* test, $P < 0.05$). Boxplots show median (central line), upper and lower quartiles (box limits), $1.5 \times$ interquartile range (whiskers). $P < 0.05$

is considered statistically significant. **(corresponding to the Figure 3C in the revision, line 1361–1364 in Page 46).**

(G) Boxplots for CD8+Tem score evaluated by xCell between S and NS groups between S and NS groups in plasma samples (left: N (S) = 16, and N (NS) = 15) and tissue samples (right: N (S) = 12, and N (NS) = 19). (two-tailed Student's t test, $P < 0.05$). Boxplots show median (central line), upper and lower quartiles (box limits), 1.5×interquartile range (whiskers). $P < 0.05$ is considered statistically significant. **(corresponding to the Figure 3E in the revision, line 1367–1371 in Page 46).**

(K) The boxplot showing RPTOR and IMPDH2 protein differential expression identified by PRM between S and NS groups in two cohorts. Discovery cohort: N (S) = 31, and N (NS) = 45; validation cohort, N (S) = 16, and N (NS) = 15. Boxplots show median (central line), upper and lower quartiles (box limits), 1.5×interquartile range (whiskers). $P < 0.05$ calculated by two-sided Wilcoxon rank-sum test is considered statistically significant. **(corresponding to the Figure 3L in the revision, line 1389–1393 in Page 47).**

Figure 4

(C) Boxplots for pathway ssGSEA score between S and NS groups. N (S) = 31, and N (NS) = 45. Boxplots show median (central line), upper and lower quartiles (box limits), 1.5×interquartile range (whiskers). $P < 0.05$ calculated by two-tailed Student's t test is considered statistically significant. **(corresponding to the Figure 4C in the revision, line 1400–1403 in Page 47).**

(D) Boxplots for RRAS and RRAS2 protein abundance between S and NS groups. N (S) = 31, and N (NS) = 45. (two-tailed Student's t test, $P < 0.05$). Boxplots show median (central line), upper and lower quartiles (box limits), 1.5×interquartile range (whiskers). $P < 0.05$ calculated by two-tailed Wilcoxon rank-sum test is considered statistically significant. **(corresponding to the Figure 4E in the revision, line 1407–1410 in Page 47).**

Reviewer #3 (Remarks to the Author): clinical expertise in colorectal cancer and cetuximab treatment

The topic of blood-based methods, including proteomics, to screen for colorectal cancer is very interesting and the authors are commended for looking at this topic. However, the article is confusing and requires better focus.

Q25. (1) A considerable part of the paper addresses how the molecules identified can be used to distinguish colon cancer from normal healthy issues with screening potential but this is not reflected in the title, aims and objectives – the title instead leading the reader to believe that the biomarkers under question (and assuming prior validation) are primarily for cetuximab resistance prediction. (2) Clearly presenting the primary and/or secondary endpoints of the study would guide the reader. (3) A better description of the selection of the cohorts (both patient and normal) would help discern the presence of selection bias. (4) Furthermore, it is not very clear which samples, at which time points have been used at every step of the analysis. (5) In the discussion, no limitations have been presented. This makes it very difficult to properly judge the significance of the findings which could be overstated. (6) The validation cohort is rather small, this would make the results weaker and the conclusions less robust. (7) Overall, better structuring of the paper, focusing the work, having a study title that reflects the research, clearly presenting endpoints, better description of the methodology which should be robust in identifying an accepted predictive biomarker development pathway and discussing potential limitations would benefit this paper.

Response to Q25:

We appreciate the reviewer for the constructive and insightful comments, which help to improve the quality of this manuscript. According to the suggestions, we divided the comments into seven parts, and the point-to-point responses were summarized as following:

(1) About the manuscript title correction for fully reflecting the objectives of the research:

We apologize for the inappropriate title “**Longitudinal plasma proteome profiling reveals the dynamic biomarkers for prediction of cetuximab therapy response in colorectal cancer patients**” for the research, due to only reflecting the biomarker discovery for cetuximab resistance prediction of colorectal cancer (CRC), while not including the potential biomarker screening for the diagnosis of CRC. In this study, we included the discovery cohort composed of 116 CRC patients undergoing anti-EGFR therapy with continuous multiple treatment courses (defined as “CRC cohort”) and 66 healthy controls (HCs) (“defined as normal cohort”) for the biomarkers’ discovery. A total of 540 plasma samples were included for proteomic measurement. This study aims to screen the noninvasive diagnostic biomarkers for the CRC, and identify the potential biological characteristics and biomarkers for the cetuximab therapy response. The comparison between CRC cohort and normal cohort was used for screening the noninvasive diagnostic biomarkers for the CRC; and the cohort composed of continuous multiple samplings during the overall treatment course was used for exploring the potential resistant mechanism and predictive biomarkers for cetuximab therapy in CRC. According to the reviewer’s suggestion, we revised the title as “**Longitudinal plasma proteome profiling reveals the diversity of biomarkers for diagnosis and cetuximab therapy response of colorectal cancer**” to well reflect our study objectives, in the revision.

(2) About the presentation of primary and secondary endpoints of the study:

We apologize for the misleading caused by not providing the sufficient presentation of the primary and secondary endpoints of the study in the last version. We appreciate the reviewer for the constructive and insightful comments, which help to improve the quality of this manuscript. In this study, based on the comprehensive plasma proteome profiling, we (i) determined the biological alteration and screened the diagnostic biomarkers for CRC; (ii) constructed the proteomic subtypes featured with distinct biological features which associated with the initial cetuximab therapy response, as well as the combination prediction of the key regulators for the initial cetuximab therapy response; (iii) constructed and validated the predictive models applied for the continuous multiple courses of cetuximab therapy in CRC. In this study, we identified different biomarkers for tumor diagnosis (COL12A1, THBS2, S100A8, and S100A9), the initial response prediction of the first treatment (RRAS2, MMP8, FBLN1,

RPTOR, and IMPDH2), as well as the longitudinal response prediction of the multi-course treatment (IDH3G, MDN1, KLC4, MYL9, SBF1, and HTRA3), which were the primary endpoints in this study. As for the secondary endpoints, here, we clarified the findings involved in this study. (i) Besides the diagnostic biomarkers for CRC (the primary endpoint), we identified the overrepresented pathways significantly enriched in the CRC and HC groups, explored the potential biological association among the diagnostic biomarkers (in the revision), which were defined as the secondary endpoints of the study. (ii) Besides the protein signatures for the initial response prediction of the first treatment (the primary endpoint), we identified the three proteomic subtypes featured with distinct biological features and associated with different therapy response. For example, the G-III subtype, as a sensitive subtype, was featured by the elevation of autophagy and immune response; while G-I subtype, as a non-sensitive subtype, was characterized by extracellular matrix organization (ECM) and MAPK signaling pathway, which were regarded as the secondary endpoints of the study. While the protein signatures for the initial response prediction of the first treatment were associated with the clinical prognosis and the initial therapy response of the first treatment, which were the primary endpoints. (iii) Besides the predictive biomarkers for the longitudinal response prediction of the multi-course treatment (the primary endpoint), we identified two protein panels positively or negatively correlated with therapy response trajectories respectively, determined the molecular features associated with the stable sensitive patients and stable non-sensitive patients, which were defined as the secondary endpoints of the study. Overall, this study revealed the heterogeneity of different biomarkers dominated for tumor diagnosis, the initial response prediction of the first treatment, as well as the longitudinal response prediction of the multi-course treatment, which is also a primary endpoint concluded in this study. We summarized our primary and secondary endpoints of the study in the **Table RL4**. In the revision, we also summarized the primary and secondary endpoints in the “**Discussion**” section (**line 646–667 in Page 22–23**).

To better reconstruct our paper and focus the primary endpoints, in the revision, we accordingly adjusted the length of manuscript corresponding to primary or secondary endpoints, and arrangement of all kinds of supporting materials, including figures, tables, et al., after adding necessary analyses based on the new additional plasma and tissue proteomic data, according

to our structure, to make the article more coherent and logical. In addition, to integrate our findings well, we (i) added the possible application of the diagnostic biomarkers in predicting the therapy response followed the diagnostic biomarkers (**Figure S2G in the revision**), (ii) added the possible application of the key proteins involved in the potential resistant/sensitive mechanism, as well the diagnostic biomarkers, in the following longitudinal analysis covering multi-course treatment (**Figures S4A and S4B in the revision**), (iii) added a summary about the heterogeneity of different management stages of the tumor, including tumor diagnosis, the initial response prediction of the first treatment, as well as the longitudinal response prediction of the multi-course treatment in the revision. In the revision, we updated all the relative results in the supplementary figures as the transition section among the previous relatively independent findings. After revisions, we believed that the manuscript would be more focused and coherent.

Table RL4: The summary of primary endpoints and secondary endpoints in each CRC management stage.

CRC management stage	Related figures	Primary endpoints	Secondary endpoints
The tumor diagnosis	Figure 1, Figure 2, Figure S1, Figure S2;	Determined the biological alteration and screened the diagnostic biomarkers for CRC (COL12A1, THBS2, S100A8, and S100A9)	(1) Identified the overrepresented pathways significantly enriched in the CRC and HC group (2) Explored the potential biological association among the diagnostic biomarkers
The first treatment	Figure 3, Figure 4, Figure S3;	Identified the key regulators combination for the prediction of the initial cetuximab therapy response (RRAS2, MMP8, FBLN1, RPTOR, and IMPDH2)	Identified the molecular features associated with different therapy response
The multi-course treatment	Figure 5, Figure S4;	Constructed and validated the predictive models applied for the continuous multiple courses of cetuximab therapy in CRC (IDH3G, MDN1, KLC4, MYL9, SBF1, and HTRA3)	(1) Identified two protein panels positively or negatively correlated with therapy response trajectories respectively (2) Determined the molecular features associated with the stable sensitive patients and stable non-sensitive patients
Summary	Figure S5	The heterogeneity/diversity of biomarkers for different management stages of the CRC patients	

(3) About the cohort description included patients and normal individuals:

In this study, to investigate the proteomic patterns of colorectal cancer (CRC) diagnosis and the association with response to cetuximab therapy, we collected 540 plasma samples from the discovery cohort composed of two independent cohorts including CRC cohort (CRC patients, N = 116) and normal cohort (healthy controls, N = 66), of which, CRC cohort included pre-treatment samples (N = 89) and post-treatment samples (N = 385) during continuous multiple treatment courses of cetuximab therapy. In this study, based on the comprehensive plasma proteome profiling, we (i) determined the biological alteration and screened the diagnostic biomarkers for CRC; (ii) constructed the proteomic subtypes featured with distinct biological features which associated with the initial cetuximab therapy response, as well as the

combination prediction of the key regulators for the initial cetuximab therapy response; (iii) constructed the predictive models applied for the continuous multiple courses of cetuximab therapy in CRC. Finally, we identified different biomarkers for the tumor diagnosis (COL12A1, THBS2, S100A8 and S100A9), the initial response prediction of the first treatment (RRAS2, MMP8, FBLN1, RPTOR, and IMPDH2), as well as the longitudinal response prediction of the multi-course treatment (IDH3G, MDN1, KLC4, MYL9, SBF1, and HTRA3), revealing the alteration of the biological function in different management stages of the tumor, as well as the tumor heterogeneity.

In the revision, to validate the prediction effect of the biomarkers for diagnosis, initial therapy response of the first treatment, as well as the longitudinal therapy response of the multi-course treatment, we expanded our plasma validation cohort composed of 31 CRC patients and 24 healthy controls. According to the Response Evaluation Criteria in Solid Tumors [RECIST] 1.1, the 31 CRC patients were grouped into 16 sensitive (S) patients and 15 non-sensitive (NS) patients in the plasma validation cohort. For the 31 CRC patients, the plasma sampling covered a cetuximab therapy period of up to 16 weeks from the therapy-naïve sampling, of which the plasma from each patient were collected when completing each therapy course (8-week therapy), resulting in a total of 77 plasma samples from 31 CRC patients covering 2 sampling time points during the treatment period. Consistently, the same sampling procedure was applied both in the discovery cohort and plasma validation cohort in the revision. In addition, in the revision, to further reveal the association of the expression of these biomarkers in plasma and tumor tissues, we reviewed all the archival formalin-fixed paraffin-embedded (FFPE) tissues from the therapy-naïve CRC patients included in this study. Finally, we collected 31 tumor tissues and 27 paired normal-adjacent tissues (NATs) of the therapy-naive CRC patients matched with the plasma samples for mass spectrometry (MS)-based proteomic profiling. We appreciate the reviewer for the comment. According to the reviewer's suggestion, we supplemented the detailed description about the inclusion of the CRC patients and healthy controls (HCs) in the part of (i) "Method" and (ii) "Manuscript" in the revision as follows. In the revision, all the baseline characteristics were shown in the **Table RL5**.

(i) About the update in the “Method”:

“The studies involving human participants were reviewed and approved by the Ethics Committee of Fudan University Shanghai Cancer Center (1506147). The patients/participants provided their written informed consent to participate in this study. The plasma samples used in this study were obtained from patients with CRC or healthy controls, from April, 2015 to February, 2021, were reviewed in the Shanghai Cancer Center, Fudan University (Shanghai, China). The study included a total of 641 plasma samples from discovery cohort composed of 116 CRC patients undergoing anti-EGFR therapy with continuous multiple treatment courses and 66 healthy controls (HCs), as well as the validation cohort composed of 31 CRC patients and 24 HCs. In the discovery cohort, CRC cohort included pre-treatment samples (N = 89) and post-treatment samples (N = 385) during continuous multiple treatment courses of anti-EGFR therapy. In addition, 31 pre-treatment plasma samples and 46 post-treatment plasma samples were included in the plasma validation cohort in the revision. Moreover, 31 tumor tissues and 27 paired normal-adjacent tissues (NATs) of CRC patients matched with the plasma samples were included in the revision.

For the CRC patients, there are 80 males and 36 females with a median age of 55.5 years (ranging from 21 to 76 years) in the discovery cohort, and 20 males and 11 females with a median age of 56 years (ranging from 29 to 77 years) in the independent plasma validation cohort, as well as 17 males and 14 females with a median age of 57 years (ranging from 25 to 76 years) in the independent tissue validation cohort. The anti-EGFR therapy regimen was given at standard dosing as described in previous studies, of which patients were given 500 mg/m² cetuximab once-every-2-weeks combined with FOLFOX/FOLFIRI/irinotecan (*Br J Cancer.*, 2008, PMID: 18665167; *JAMA Oncol.*, 2018, PMID: 29450468; *Ann Oncol.*, 2013, PMID: 23559149). The inclusion criteria were as follows: (i) diagnosis of CRC reviewed by three expert pathologists; (ii) presence of at least one measurable or unmeasurable but evaluable lesion (described according to Response Evaluation Criteria in Solid Tumors [RECIST] 1.1 by CT/MRI scanning and grouped into complete response (CR), partial response (PR), stable disease (SD), or progressive disease (PD)); (iii) presence of polymerase chain reaction (PCR)-confirmed wild-type *KRAS* (exon 2/3/4), *NRAS* (exon 2/3/4), and *BRAF* (exon 15) genotypes in

tumor tissue before the receipt of anti-EGFR therapy; (iv) no history of severe heart or liver disease, psychiatric disorders, hemorrhage, or perforation of the digestive tract; (v) and an Eastern Cooperative Oncology Group performance status of 0/1 at 3 days before treatment (*Front Oncol.*, 2022, PMID: 35280779).

For the healthy controls, there are 22 males and 44 females with a median age of 62 years (ranging from 57 to 63 years) in the discovery cohort, and 12 males and 12 females with a median age of 55 years (ranging from 25 to 68 years) in the independent plasma validation cohort. The enrollment criteria for HC subjects were as follows: (i) the absence of benign or malignant tumors; (ii) a qualified physical examination finding no dysfunction of vital organs and (iii) normal renal function and without albuminuria. After collection, plasma and tissue samples were stored at -80 °C.” (line 687–731 in Page 23–25).

(ii) About the update in the “Manuscript”:

“The age and sex distributions were balanced among the CRC patients and healthy controls both in plasma discovery cohort, plasma validation cohort and tissue validation cohort, showing there was no selection bias among these individuals. In addition, there was no statistically significant difference in the clinical parameters of serum lactate dehydrogenase (LDH) level, white blood cell count, lymphocyte number, hemoglobin, and platelet count among the CRC groups (Table RL5, see also Table 1 in the revision).” (line 150–159 in Page 5–6).

Table RL5 Baseline characteristics in the discovery and validation cohorts.

	Plasma discovery cohort		Plasma validation cohort		Tissue validation cohort	P-value
	Healthy control (N = 66)	CRC (N = 116)	Healthy control (N = 24)	CRC (N = 31)	CRC (N = 31)	
Age (years), median (range)	62 (57-63)	55.5 (21-76)	55 (25-68)	56 (29-77)	57 (25-76)	0.660 ^a
Gender						0.334 ^a
Female	44 (66.7%)	36 (31.0%)	12 (50.0%)	11 (35.5%)	14 (45.2%)	
Male	22 (33.3%)	80 (69.0%)	12 (50.0%)	20 (64.5%)	17 (54.8%)	
Degree of tumor differentiation						0.444 ^b
Poorly differentiated	-	29 (25.0%)	-	5 (16.1%)	11 (35.5%)	
Moderately differentiated	-	54 (46.6%)	-	17 (54.8%)	16 (51.6%)	
Well differentiated	-	2 (1.7%)	-	0 (0.0%)	0 (0.0%)	
NA	-	31 (26.7%)	-	9 (29.0%)	4 (12.9%)	
ECOG performance status						0.511 ^b
1	-	111 (95.7%)	-	31 (100%)	31 (100%)	
0	-	5 (4.3%)	-	0 (0.0%)	0 (0.0%)	
Lactate dehydrogenase level (U/L), median (range)	-	237 (97-3,000)	-	184 (118-2,231)	235 (118-2,231)	0.262 ^a
White Blood Cell count (10 ⁹ /L), median (range)	-	5.75 (1.7-15.2)	-	6.1 (2.9-11.9)	6.5 (2.9-14.8)	0.529 ^a
Lymphocyte number (10 ⁹ /L), median (range)	-	1.4 (0.3-3.1)	-	1.7 (0.3-107)	1.6 (0.7-107)	0.139 ^a
Hemoglobin (g/L), median (range)	-	126 (39-163)	-	127 (97-164)	126 (39-152)	0.365 ^a
Platelet Count (10 ⁹ /L), median (range)	-	198.5 (48-561)	-	206 (105-417)	205 (112-506)	0.692 ^a

a, one-way ANOVA analysis was applied for continuous variables. b, fisher's exact test was used for categorical variables.

(4) About the samples and sampling times included in each analysis:

We apologize for not clearly providing the information of the samples and its sampling time in the last version. In this study, we collected the plasma samples from each healthy control and patient. For patients included in the discovery cohort of this study, the plasma sampling covered a cetuximab therapy period of up to 72 weeks from the therapy-naïve sampling, of which the plasma from each patient were collected when completing each therapy course (8-week therapy), resulting in a total of 474 plasma samples from 116 CRC patients covering 9 sampling time points during the treatment period. In detail, we sampled the plasma and performed therapy response evaluation according to the RECIST, when patients completed one course treatment every time; and this procedure endured up to 9 times for 72 weeks at most in the therapy trajectories. The same sampling procedure was also applied in the plasma validation cohort composed of 31 CRC patients (in the revision). To answer the reviewer's question, we split into three parts according to our analysis as following:

(I) In the differential analysis to identify diagnostic markers for CRC, we used the plasma samples collected from 89 therapy-naïve CRC patients and 66 healthy controls, and constructed differential expressed plasma proteome, finally determined the biological alteration and screened the diagnostic biomarkers for CRC. The validation for the diagnostic markers was also based on the plasma samples in the therapy-naïve CRC patients in the plasma validation cohort. In the revision, we included 55 plasma samples from 31 CRC patients and 24 healthy controls as the independent plasma validation cohort, as well as 31 tumor tissues and 27 paired normal-adjacent tissues (NATs) of CRC patients matched with the plasma samples as the independent tissue validation cohort, for proteomic measurement to validate the prediction effect of diagnostic biomarkers in distinguishing CRC patients from healthy controls.

(II) In the consensus clustering analysis to identify proteomic subtyping, we used the plasma samples collected from 89 therapy-naïve CRC patients, and achieved the proteomic subtypes featured with distinct molecular features associated with therapy response. We identified key regulators involved the featured pathways related to the therapy response. To validate the potential association of these key proteins with the response, in the revision, we grouped these therapy-naïve CRC patients into sensitive (S) and non-sensitive (NS) groups according to the RECIST 1.1 guideline. There were 12 S patients and 19 NS patients included in the tissue

validation cohort, as well as 16 S patients and 15 NS patients included in the plasma validation cohort. We collected the plasma/tissue samples from therapy-naïve CRC patients for the proteomic measurement. ROC analysis revealed that the prediction effect of these key proteins showed a high accuracy both in the plasma validation cohort and tissue validation cohort.

(III) In the part of construction and validation of the predictive model applied for the continuous multiple courses of cetuximab therapy, we focused on 105 plasma samples from a longitudinal cohort composed of 22 CRC patients, covering seven sampling times (that is, seven courses cetuximab treatment) and response evaluation results. In the revision, we expanded our validation cohort, which composed of 31 CRC patients, 77 plasma samples, covering two-course cetuximab treatment according to the reviewer's suggestion. We validated the prediction performance of this subset of signatures in the plasma validation cohort. The results of confusion matrix analyses exhibited an accuracy of 0.929 (95%CI: 0.765–0.991) in the first course treatment, and 1 (95%CI: 0.815–1) in the second course treatment in the plasma validation cohort, validating the robustness of this predictive model in the multi-course treatment. According to the reviewer's suggestions, we added the sampling information of samples used for each analysis in the main text of the "**Result**" section and the corresponding description of the "**Method**" section in the revision.

(5) About the discussion of study limitations:

We appreciate the reviewer for the comment. According to the reviewer's suggestion, we supplemented the discussion about the limitations of the study. In the revision, we added the part of "Limitations of the study" in the manuscript as following:

"In this study, we established the first longitudinal plasma proteome profiling of colorectal cancer to identify the effective diagnostic markers and predictive markers for cetuximab therapy, thus contributing to the monitoring and intervening in the treatment. Among the limitations to this study, first, this study is a single-center research likely does not represent all the heterogeneous mechanisms underlying clinical resistance to cetuximab therapy. The multi-center cohorts need to be included for such a study in the future. Second, there are around 1/6 of CRC patients but not all CRC patients receiving seven course cetuximab treatment (which

reflects the real clinical world), the biomarkers for the response prediction of the multi-course treatment identified in this study should be validated in the more complete multi-course longitudinal cohort. Third, although the potential biomarkers for CRC diagnosis, were elevated both in plasma and tumor tissues of CRC patients, we couldn't completely exclude the contribution of other organs or tissues to the level of these proteins in plasma, which might need to be explored through comparison among more tissues. A more rigorous multi-tissue comparison research is worth studying in the future. Fourth, it is deserved to explore the components secreted by circulating immune cells associated with the therapy response in the future." (line 669–684 in Page 23).

(6) About the enlargement of the validation cohort for the robustness of predictive models:

We appreciate the reviewer for the valuable suggestions about the validation of the predictive models. According to the reviewer's suggestions, in the revision, we included 101 plasma samples from 31 CRC patients and 24 healthy controls as the independent plasma validation cohort. In addition, we reviewed all the archival formalin-fixed paraffin-embedded (FFPE) tissues from the therapy-naïve CRC patients included in this study, and finally collected 31 tumor tissues and 27 paired normal-adjacent tissues (NATs) of CRC patients matched with the plasma samples as the independent tissue validation cohort. In the revision, we added the prediction results of diagnostic and predictive models developed in this study in the independent plasma and tissue validation cohorts. Here, we divided three parts to answer the question:

(i) As for the diagnostic biomarkers of distinguishing CRC patients from healthy controls

In this study, to search for plasma proteins that could be used as diagnostic biomarkers for CRC patients, we used the following three criteria in this study: 1) The candidate proteins were expressed in at least 50% of the samples (1,359 proteins); 2) The candidates were significantly increased in tumor samples than normal samples (two-sided Wilcoxon rank-sum test, adj *P*-value < 0.01; 234 proteins); 3) The candidates were identified with at least 2-fold increase in CRC samples than normal samples. As a result, 148 proteins were screened, and significantly and stably overexpressed in the plasma of CRC patients. To further validate whether the

candidates identified in plasma were also commonly in tissue samples from CRC patients, we applied the same criteria in the pairwise comparison between tumor tissue and matched non-tumor adjacent tissue proteomic data of CRC patients from the Office of Cancer Clinical Proteomics Research (CPTAC) cohort, which resulted in 31 proteins that were significantly overexpressed in tumor tissues of CRC patients. Ultimately, we obtained a group of proteins, COL12A1, THBS2, S100A8, and S100A9, validated both in plasma and tissues of CRC patients. Further receiver operating characteristic (ROC) analysis determined a well performance of distinguishing CRC patients from healthy controls.

To validate the prediction effect of the four proteins composed model, in the revision, we included more CRC patients in the plasma validation cohort composed of 31 CRC patients and 24 healthy controls for the proteomic measurement. Consistently, the differential proteomic analysis revealed that COL12A1, THBS2, S100A8, and S100A9 were significantly up-regulated in the plasma samples from CRC patients in the plasma validation cohorts. We further performed the receiver operating characteristic (ROC) analysis to evaluate the predictive effect. The ROC analysis revealed the four proteins combined prediction achieved good performance with an AUC of 0.952 in the plasma validation cohort (**Figure RL18A**).

To further reveal the association of the expression of these biomarkers in plasma and tumor tissues, in the revision, we reviewed all the archival formalin-fixed paraffin-embedded (FFPE) tissues from the therapy-naïve CRC patients included in this study. Finally, we collected 31 tumor tissues and 27 paired normal-adjacent tissues (NATs) of CRC patients matched with the plasma samples for mass spectrometry (MS)-based proteomic profiling. Proteomic analysis of tumor tissues identified a total of 9,684 proteins, of which 6,927 proteins were detected in plasma samples. The comparative proteomic analysis demonstrated that 2,573 proteins were significantly elevated in tumor tissues; while 251 proteins were down-regulated in tumor tissues (fold change (Tumor vs NAT) >2, adj *P*-value < 0.05). In the previous version, based on the plasma proteome profiling, the comparative analysis identified 745 proteins up-regulated in the CRC patients; among of these proteins, 235 (31.54%) proteins were up-regulated in tumor tissues from CRC patients. Interestingly, this result showed a relatively high overlapped

proportion of the differential proteins up-regulated in both the plasma samples and tumor tissue samples of the CRC patients. Evidently, COL12A1 (FC = 4.29, adj *P*-value = 8.8E-4), THBS2 (FC = 3.43, adj *P*-value = 1.6E-4), S100A8 (FC = 3.40, adj *P*-value = 8.7E-5), and S100A9 (FC = 5.10, adj *P*-value = 1.2E-4), showed a significant increase in tumor tissues. Furthermore, we validated the diagnostic efficiency of the four proteins in the independent tissue validation cohort. The receiver operating characteristic (ROC) analysis revealed the four proteins combined prediction achieved good performance with an AUC of 0.945, in the independent tissue validation cohort (**Figure RL18A, see also Figure 2G in the revision**). Overall, the refined biomarkers could well distinguish the CRC patients from healthy controls both in plasma discovery cohort and plasma/tissue validation cohort.

(ii) As for the potential key signatures related to the resistant/sensitive mechanism uncovered by consensus clustering analysis

To further explore the potential resistant/sensitive mechanism and biomarkers for therapy response, we firstly applied consensus clustering analysis, an unsupervised clustering method, and preliminarily determined the association between proteome pattern and therapy response. In this study, we identified three subtypes featured with distinct clinical characteristics and biological functions. For example, the G-III subtype, as a sensitive subtype, was featured by the elevation of autophagy and immune response; while G-I subtype, as a non-sensitive subtype, was characterized by extracellular matrix organization (ECM) and MAPK signaling pathway. We identified key proteins involved in these overrepresented pathways, including RRAS2, MMP8, FBLN1, RPTOR, and IMPDH2, which yielded a high prediction with AUC values of 0.804 in the training set and 0.955 in the testing set, in distinguishing the sensitive patients from the non-sensitive patients with receiving cetuximab therapy. To validate the potential association of these key proteins with the response, in the revision, we grouped these therapy-naïve CRC patients into sensitive (S) and non-sensitive (NS) groups according to the RECIST 1.1 guideline. There were 12 S patients and 19 NS patients included in the tissue validation cohort, as well as 16 S patients and 15 NS patients included in the plasma validation cohort. We collected the plasma/tissue samples from therapy-naïve CRC patients for the proteomic measurement. ROC analysis revealed that the combination predictive model

composed of RRAS2, MMP8, FBLN1, RPTOR, and IMPDH2 had a good performance with AUC values of 0.890 and 0.816 in the independent plasma validation cohort and tissue validation cohort, demonstrating the stability and robustness of the predictive model for predicting response to cetuximab first-time treatment (**Figure RL18B, see also Figure 4L in the revision**).

(iii) As for the prediction models applied for the continuous multiple courses of anti-EGFR therapy in CRC revealed in longitudinal analysis

To identify the protein dynamic change associated with the treatment response to cetuximab during multiple treatment courses, we focused on a longitudinal cohort composed of 22 CRC patients, 105 plasma samples, covering seven-course cetuximab treatment. In this section, we screened for a panel of proteins with significant negative correlation and up-regulated in the stable sensitive group, and a panel of proteins with significant positive correlation and up-regulated in the stable non-sensitive group. Having identified biomarkers with fluctuations at protein level associated with response to cetuximab therapy, we next set out to determine whether these biomarkers could be used for predicting response to cetuximab treatment in CRC patients during the continuous courses. We employed stepwise logistic regression and identified a subset of signatures (including IDH3G, MDN1, KLC4, MYL9, SBF1, and HTRA3) that accurately discriminates stable sensitive group and stable non-sensitive group. We further determined the ability of the predictive model for the response of cetuximab treatment across different courses. In the revision, we expanded our validation cohort, which composed of 31 CRC patients, 77 plasma samples, covering two-course cetuximab treatment according to the reviewer's suggestion. We validated the prediction performance of this subset of signatures in the plasma validation cohort. The results of confusion matrix analyses exhibited an accuracy of 0.929 (95%CI: 0.765–0.991) in the first course treatment, and 1 (95%CI: 0.815–1) in the second course treatment in the plasma validation cohort (**Figure RL18C, see also Figure 5G in the revision**). Taken together, the predictive model had high accuracy in the multi-course treatment course, exhibiting better performance for the early warning in the initial treatment, which were further validated in the independent cohort.

Figure RL18. The validation of the diagnostic biomarkers, signatures related to the response to first cetuximab treatment, and the predictive markers for the multi-courses during the treatment. (A and B) The receiver operating characteristic (ROC) curves showing the prediction effect of the diagnostic markers in the discovery and validation cohorts (A), and the predictive markers for initial response prediction in the discovery and validation cohorts (B). (C) Classification error matrix using logistic regression classifier in distinguishing S and NS in the plasma longitudinal validation cohort. The number of samples identified is noted in each box.

(7) Summary

Overall, in the revision, according to reviewers' constructive comments, firstly, we revised the manuscript title as **“Longitudinal plasma proteome profiling reveals the diversity of biomarkers for diagnosis and cetuximab therapy response of colorectal cancer”** which fully reflecting the objective of this research. Then, we removed the redundant information into supplementary materials, prioritized and integrated overabundant minor points, making the

primary and secondary endpoints of the study clear and focused. We also summarized the primary and secondary endpoints in the “Discussion” section. Next, we also supplemented the full introduction of our cohorts in the method, and clearly make a statement related to the samples used for each data analysis in the revision. Moreover, we enlarged our plasma validation and additionally included the independent tissue validation cohort to validate the stability and robustness of our predictive models in the revision. According to the reviewer’s suggestion, we supplemented the discussion about the limitations of this study to provide a better understanding of our data and results.

In addition, for a better structure and holistic presentation of our story, we firstly added the possible application of the diagnostic biomarkers (COL12A1, THBS2, S100A8 and S100A9) in predicting the therapy response followed the diagnostic biomarkers in the revision, which revealed the combination of these proteins had a poor prediction in therapy response. Then, to explore the potential resistant/sensitive mechanism and biomarkers for therapy response, we applied consensus clustering analysis and determined three subtypes featured with distinct clinical characteristics and biological functions. We identified key proteins (RRAS2, MMP8, FBLN1, RPTOR, and IMPDH2), which yielded a high prediction in distinguishing the sensitive patients from the non-sensitive patients with receiving the initial cetuximab therapy. Taken the longitudinal characteristics of our cohort into consideration, in the revision, we applied the diagnostic biomarkers and predictive model for therapy response of the initial cetuximab treatment in the longitudinal cohort covering multi-course treatment. As a result, we found both the diagnostic model composed of the four biomarkers (COL12A1, THBS2, S100A8, and S100A9), and the predictive model composed of the key proteins (RRAS2, MMP8, FBLN1, RPTOR, and IMPDH2) for therapy response of the initial cetuximab treatment didn’t achieve a good distinguish between sensitive and non-sensitive patients during the multi-course cetuximab treatment. Therefore, followed by this, we focused on a longitudinal cohort composed of 22 CRC patients, 105 plasma samples, covering seven-course cetuximab treatment. In the revision, we integrated our major findings in a holistic strategy, and supplemented the transitions among the findings. As a summary, we integrated major findings, clinical cohorts, as well as clinical characteristics, and provided a comprehensive diagram

showing the connection of each result as follows (**Figure RL19, see also Figure S5 in the revision**). Overall, our study revealed the heterogeneity of different biomarkers dominated for tumor diagnosis, the initial response prediction of the first treatment, as well as the longitudinal response prediction of the multi-course treatment.

At last, after compiling all the materials, we adjusted arrangement of all kinds of supporting materials, including figures, tables, et al., according to the primary and secondary endpoints of the study to make the article more coherent and logical. After revision, we believe that the manuscript would be more concise, understandable and focused.

Figure RL19. The diagram summarizing the connection of clinical samples, data analysis, and major findings.

REVIEWER COMMENTS

Reviewer #1 (Remarks to the Author):

Authors have done an excellent job addressing reviewer concerns point by point. The straightforward explanations, updated figures, and appropriate references to text are all appreciated. This work is now suitable for acceptance/publication.

Reviewer #2 (Remarks to the Author):

The authors addressed the raised concerns, and I think this manuscript is ready for publication.

Reviewer #4 (Remarks to the Author):

This revised manuscript profiles 641 plasma samples from 147 colorectal cancer patients under cetuximab therapy, alongside 90 controls. It identifies diagnostic biomarkers (COL12A1, THBS2, S100A8, S100A9) in plasma and tissue and predicts initial response using RRAS2, MMP8, FBLN1, RPTOR, and IMPDH2. Longitudinal tracking reveals two patient clusters, and attempts to build a robust predictive model that is evaluated in an independent cohort.

In response to reviewers, the authors have done a substantial amount of work, Yet, unresolved issues remain, requiring further attention for an informative manuscript that addresses each of the applications/claims of the authors

I will address this review based on the claims/potential applications of the diagnostic markers the authors have proposed

(1) screening /diagnostic biomarkers for CRC patients;-- proposed 4 refined proteins (COL12A1, THBS2, S100A8, and S100A9),

It is not clear that the 4 markers are specific to CRC or indeed to cancer and is not ready for use as a screening nor diagnostic marker.

They proposed 4 refined proteins (COL12A1, THBS2, S100A8, and S100A9), validated in both plasma and tumor tissues, effectively distinguished CRC patients from healthy controls in both discovery and validation cohorts They collected 31 tumor tissues and 27 paired normal-adjacent tissues (NATs) of CRC patients matched with the plasma samples for mass spectrometry (MS)-based proteomic profiling. Their study suggest differences with healthy controls.

However, the use of the 4 refined proteins as a diagnostic marker is not certain, do they intend to use this in screening, non-invasive diagnostics. While the study provides evidence of their association with CRC, it's important to note that these proteins may also be found in plasma samples from patients with other conditions. The authors have noted that THBS2 is associated with NAFLD. Elevated levels of COL12A1, THBS2 (thrombospondin 2), S100A8 (calgranulin A), and S100A9 (calgranulin B) have been associated with various medical conditions and diseases beyond cancer, such as rheumatoid arthritis (RA), inflammatory bowel disease (IBD), cardiovascular disease, osteoarthritis (OA), diabetes, neurological disorders, autoimmune diseases, and wound healing processes, extending beyond their associations with cancer.

Interestingly, The four biomarkers showed potential biological roles in extracellular matrix-related processes based on GSEA, positive correlations with extracellular matrix organization from ssGSEA, and strong interactions with potential crosstalk and cooperation in cellular processes, particularly in the context of the extracellular matrix. within the protein-protein interaction network. Again, pointing to non-cancer specific processes.

Clarifying the intention of the diagnostic marker, and then showing it's clinical validity in the appropriate population is required. Rather than from a simple case-control study, consecutive

patients in a colonoscopy cohort would be more appropriate for the proposed utility of these set of diagnostic markers.

Notably, most or all of the cases in this cohort are patients due to receive anti-EGFR therapy, thus only representing (1) metastatic CRC patients (2) patients without RAS mutations [not reflective of all CRC patients, possibly] and thus not suited for this diagnostic/screening application.

I appreciate this might be beyond the scope of the manuscript and discussion of this caveat (that whilst it is shown to be different in CRC vs healthy, it is up in other cancers & also up in other non-malignant conditions) and may be reflective of general inflammatory processes and is not ready for use as a cancer nor CRC diagnostic will be required. This obviously lowers the novelty and practical use of their finding. Statements regarding it's use as a cancer screening marker must thus be removed from the manuscript or substantially toned down. If they do have aspiration as a diagnostic or screening markers, then further cohorts will be required with appropriate controls for example an screening endoscopy cohort showing that the non-invasive marker does distinguish the cancers from other healthy patients found to have normal endoscopy.

(2) Proposed (RRAS2, MMP8, FBLN1, RPTOR, and 22 IMPDH2) for the initial response prediction & suggested potential biological mechanisms and biomarkers for the therapy response

Plasma analyses of 89 therapy-naive CRC patients with initial therapy response was evaluated based on the RECIST 1.1 criteria, classifying patients into sensitive (S) and non-sensitive (NS) groups.

Unsupervised consensus clustering analysis was performed on 1,500 of the most variable proteins, resulting in the identification of three proteomic subtypes: G-I, G-II, and G-III.

Key findings:

- Patients in the G-III proteomic subtype tended to benefit from cetuximab therapy, with an increase in CD8+ Tem cells and activation of T cell receptor signaling.
- Patients in the G-I proteomic subtype were unlikely to benefit from cetuximab therapy and exhibited activation of the RRAS and RRAS2-mediated ECM pathway.
- Key proteins involved in these pathways were identified and combined to create a predictive model for initial response to cetuximab therapy, which showed robustness and stability across validation cohorts.
- CD8+ Tem cells were identified as potential markers for cetuximab sensitivity, validated across different data types and cohorts.

Comment:

R2 had asked (Q10) about other clinical parameters that may associate with response to anti-EGFR therapy, and whether these were accounted for. This is inadequately answered, in particular, given the association with Right and Left RAS-wildtype CRC with anti-EGFR therapy from phase 3 trials, this must be addressed both as a table showing how proteomic subtypes relate to left and right cancers and whether the predictive abilities remain after adjusting for left/right status.

The findings in relation to pathways related to response/non-response are by nature speculative or observational. The authors should discuss this in relation to current known mechanisms of response/non-response and also discuss why they think it did not predict longitudinal response, just initial response

(3) Longitudinal dynamics of biomarkers in relation to therapy response of CRC patients with cetuximab

Small cohort:

22 CRC patients, 105 plasma samples, covering seven-course cetuximab treatment.

subset of signatures (including IDH3G, MDN1, 912 KLC4, MYL9, SBF1, and HTRA3) markers suggested

Update: 31 CRC patients, 77 plasma samples, covering twocourse cetuximab treatment

- Initial diagnostic and predictive models did not effectively distinguish between sensitive and non-

sensitive patients during cetuximab treatment.

- Protein dynamic changes were observed in patients' responses to cetuximab treatment during multiple courses.
- Sensitive and non-sensitive biomarkers of cetuximab therapy were identified based on protein level fluctuations during treatment.
- Pathway enrichment analysis identified pathways associated with these differentially expressed proteins (DEPs), highlighting differences in glycolysis, signaling by Rho GTPases, ECM proteoglycans.

Comment: Details of the protein level dynamic fluctuations need to be provided.

Given the small number of patients, a supplementary figure showing how the levels of these proteins changed over time in relation to recist response over time in each of these patients will allow the reader/reviewer to appreciate the robustness and dynamic range of this set of protein markers.

Since this marker is being proposed to provide early readouts of response/non-response, acquired resistance amongst patients with initial response, this information will help clinical readers to interpret the results

RESPONSE TO REVIEWERS' COMMENTS

Reviewer #1 (Remarks to the Author):

Authors have done an excellent job addressing reviewer concerns point by point. The straightforward explanations, updated figures, and appropriate references to text are all appreciated. This work is now suitable for acceptance/publication.

Response:

We appreciate the reviewer's satisfaction with the last round revision. We deeply thank the reviewer for giving us the chance for publication in Nature Communications.

Reviewer #2 (Remarks to the Author):

The authors addressed the raised concerns, and I think this manuscript is ready for publication.

Response:

Thank the reviewer for supporting the publication of our manuscript. It's our pleasure to publish our manuscript in Nature Communications.

Reviewer #4 (Remarks to the Author):

This revised manuscript profiles 641 plasma samples from 147 colorectal cancer patients under cetuximab therapy, alongside 90 controls. It identifies diagnostic biomarkers (COL12A1, THBS2, S100A8, S100A9) in plasma and tissue and predicts initial response using RRAS2, MMP8, FBLN1, RPTOR, and IMPDH2. Longitudinal tracking reveals two patient clusters, and attempts to build a robust predictive model that is evaluated in an independent cohort.

In response to reviewers, the authors have done a substantial amount of work, Yet, unresolved issues remain, requiring further attention for an informative manuscript that addresses each of

the applications/claims of the authors

I will address this review based on the claims/potential applications of the diagnostic markers the authors have proposed.

Response:

Thanks for the reviewer's patient reading and concise summary of all the revision in this manuscript. We also appreciate the reviewer's constructive suggestions, which help to understand the potential of these biomarkers in clinical use and make the manuscript to be more rigorous, improving the overall quality of this manuscript. Here, according to the reviewer's comments, we provided point-to-point responses for (1) about the potential proteins in distinguishing CRC patients from healthy controls, (2) about the biomarkers for the initial response prediction, (3) about the biomarkers for the longitudinal response prediction, respectively. The point-to-point responses are as follows.

Q1. screening /diagnostic biomarkers for CRC patients;-- proposed 4 refined proteins (COL12A1, THBS2, S100A8, and S100A9),

It is not clear that the 4 markers are specific to CRC or indeed to cancer and is not ready for use as a screening nor diagnostic marker.

They proposed 4 refined proteins (COL12A1, THBS2, S100A8, and S100A9), validated in both plasma and tumor tissues, effectively distinguished CRC patients from healthy controls in both discovery and validation cohorts They collected 31 tumor tissues and 27 paired normal-adjacent tissues (NATs) of CRC patients matched with the plasma samples for mass spectrometry (MS)-based proteomic profiling. Their study suggested differences with healthy controls.

However, the use of the 4 refined proteins as a diagnostic marker is not certain, do they intend to use this in screening, non-invasive diagnostics. While the study provides

evidence of their association with CRC, it's important to note that these proteins may also be found in plasma samples from patients with other conditions. The authors have noted that THBS2 is associated with NAFLD. Elevated levels of COL12A1, THBS2 (thrombospondin 2), S100A8 (calgranulin A), and S100A9 (calgranulin B) have been associated with various medical conditions and diseases beyond cancer, such as rheumatoid arthritis (RA), inflammatory bowel disease (IBD), cardiovascular disease, osteoarthritis (OA), diabetes, neurological disorders, autoimmune diseases, and wound healing processes, extending beyond their associations with cancer.

Interestingly, The four biomarkers showed potential biological roles in extracellular matrix-related processes based on GSEA, positive correlations with extracellular matrix organization from ssGSEA, and strong interactions with potential crosstalk and cooperation in cellular processes, particularly in the context of the extracellular matrix. within the protein-protein interaction network. Again, pointing to non-cancer specific processes.

Clarifying the intention of the diagnostic marker, and then showing its clinical validity in the appropriate population is required. Rather than from a simple case-control study, consecutive patients in a colonoscopy cohort would be more appropriate for the proposed utility of these set of diagnostic markers.

Notably, most or all of the cases in this cohort are patients due to receive anti-EGFR therapy, thus only representing metastatic CRC patients without RAS mutations [not reflective of all CRC patients, possibly] and thus not suited for this diagnostic/screening application.

I appreciate this might be beyond the scope of the manuscript and discussion of this caveat (that whilst it is shown to be different in CRC vs healthy, it is up in (1) other cancers & also up in (2) other non-malignant conditions) and may be reflective of general inflammatory processes and is not ready for use as a cancer nor CRC diagnostic will be

required. This obviously lowers the novelty and practical use of their finding. Statements regarding its use as a cancer screening marker must thus be removed from the manuscript or substantially toned down. If they do have aspiration as a diagnostic or screening markers, then further cohorts will be required with appropriate controls for example a screening endoscopy cohort showing that the non-invasive marker does distinguish the cancers from other healthy patients found to have normal endoscopy.

Response to Q1:

We sincerely thank the reviewer for the constructive comments and in-depth discussion about the diagnostic biomarkers. To answer the reviewer's questions clearly, we divided the response into two parts as follows: (1) about the specificity of the four proteins (COL12A1, THBS2, S100A8, and S100A9) in distinguishing patients with CRC, but not other diseases, from healthy controls; (2) about the clinical validity of the four proteins on a wider population scale of CRC patients, but not limited to the metastatic CRC patients or *RAS* wild-type CRC patients.

(1) about the specificity of the four proteins (COL12A1, THBS2, S100A8, and S100A9) in distinguishing patients with CRC, but not other diseases, from healthy controls

We apologize for not providing more information to support the specificity of the biomarkers for the diagnosis of CRC rather than other diseases. We appreciate the reviewer for the constructive suggestions about the association of the biomarkers with other cancers and even other medical diseases beyond cancer. We agree with the reviewer that the four proteins (COL12A1, THBS2, S100A8, and S100A9), involving extracellular matrix organization processes, could associate with some diseases in addition to cancer. According to the reviewer's suggestion, in the revision, to explore whether the four proteins were specific to CRC diagnosis rather than other cancers, we further enrolled a multi-cancer (including seven cancer types) plasma independent cohort composed of a total of 115 plasma samples, including 95 plasma samples from treatment-naive patients with various cancer types (including colorectal cancer (CRC, N = 20), lung cancer (LC, N = 15), malignant lymphoma (ML, N = 10), bladder cancer (BLCA, N = 10), breast carcinoma (BRCA, N = 15), gastric cancer (GC, N = 10), esophageal cancer (EC, N = 15)), and 20 plasma samples from healthy controls (HCs, N = 20).

The proteomic measurement showed the four proteins (COL12A1, THBS2, S100A8, and S100A9) were consistently up-regulated in the CRC patients compared with HCs, while the four proteins didn't exhibited the consistency of up-regulation in other cancers (including GC, EC, BRCA, LC, BLCA, and ML), suggesting the consistency of up-regulation of the four proteins only in CRC but not in other cancers. Specifically, for the GC, S100A8 and THBS2 showed a higher expression in the GC patients compared with HCs, while S100A9 and COL12A1 showed no significant difference in the GC; for the EC, S100A8 and S100A9 were up-regulated in the EC, but COL12A1 showed no obvious change and THBS2 was even down-regulated in the EC; for the BRCA, COL12A1 and S100A9 showed an elevation in the BRCA patients compared with HCs, while S100A8 and THBS2 showed no significant change between BRCA patients and HCs; for the LC, S100A8 and THBS2 showed a higher expression in the LC patients compared with HCs, while COL12A1 and S100A9 showed no significant change between LC patients and HCs; for the BLCA, only S100A8 was up-regulated in the BLCA patients compared with HCs, while THBS2 was down-regulated in the BLCA, and other two proteins (COL12A1 and S100A9) showed no significant change between BLCA patients and HCs; for the ML, only THBS2 was up-regulated in the ML patients compared with HCs, while COL12A1 was down-regulated in the ML, and other two proteins (S100A8 and S100A9) showed no significant change between ML patients and HCs (**Figure RL1A**). These results demonstrated that although the four proteins (COL12A1, THBS2, S100A8, and S100A9) showed association with cancers and some proteins were up-regulated in the particular cancers, the consistency of up-regulation of the four proteins was only observed in CRC while the various difference (part up-regulation or even down-regulation) of the four proteins was observed in other cancers. We summarized the differential expression of the four proteins in these cancers in the **Figure RL1B**. To further verify the predictive efficacy of the four proteins in distinguishing patients with CRC, but not other cancers, from HCs, we also performed the receiver operating characteristic (ROC) analysis to evaluate the predictive effect. The ROC analysis revealed the four proteins combined prediction achieved good performance with an AUC of 0.910 in distinguishing CRC patients from HCs, while the four proteins had poor performance with AUC values no more than 0.685 in distinguishing patients with other cancers from HCs (GC: 0.685, EC: 0.663, BRCA: 0.673, LC: 0.670, BLCA: 0.440, ML: 0.655) (**Figure RL1C, see also Figure S2I in the revision**).

Overall, in the revision, we validated the consistency of up-regulation of the four proteins (COL12A1, THBS2, S100A8, and S100A9) was only observed in CRC but not in other cancers; and the four proteins could achieve good performance in the diagnosis of CRC, but not other cancers.

In addition, to explore the predictive efficacy of the four proteins to distinguish patients with some non-malignant conditions, according to the reviewer's comments, we also searched for the public datasets related to ulcerative colitis (an inflammatory bowel disease as the reviewer mentioned) and infection disease (such as SARS-CoV-2 infection), which could be obtained from Gene Expression Omnibus (GEO) with the accession number GSE11223 and GSE207015, respectively. The GSE11223 dataset was composed of ulcerative colitis patients (UC, N = 129) and healthy controls (HCs, N = 73), and the GSE207015 dataset was composed of SARS-CoV-2 infected patients (COVID-19, N = 124) and non-infected healthy controls (N = 70). Then, the differential expression analysis showed S100A8 and S100A9 showed an elevation in UC compared with HCs, while COL12A1 and THBS2 showed no significant change between UC and HCs; as for the SARS-CoV-2 infection, we observed the four proteins (COL12A1, THBS2, S100A8, and S100A9) showed no obvious change between the SARS-CoV-2 infected patients and non-infected HCs (**Figure RL1A and 1B**). To further investigate whether the four proteins could distinguish the patients with these non-malignant conditions, we performed the receiver operating characteristic (ROC) analysis. The results revealed the four proteins combination could not distinguish the patients with the non-malignant diseases (such as inflammatory diseases and infections) from healthy controls with AUC values no more than 0.660 (UC: 0.659, SARS-CoV-2 infection: 0.612) (**Figure RL1C, see also Figure S2I in the revision**).

Taken together, in the revision, we validated that the combination of four proteins could well distinguish the CRC patients from HCs, but could not distinguish the patients with other cancers and the non-malignant diseases (such as autoimmune diseases and infections), demonstrating the relative specificity of the four proteins for the CRC diagnosis. It is worth exploring the four proteins in the diagnosis of various diseases in the future.

Figure RL1. The predictive efficacy of the four proteins in distinguishing patients with different cancer types and some non-malignant diseases from healthy controls. (A) Barplot showing the differential expression of the four proteins (COL12A1, THBS2, S100A8, and S100A9) in different diseases (including colorectal cancer, gastric cancer, esophageal cancer, breast carcinoma, lung cancer, ulcerative colitis, bladder cancer, malignant lymphoma, and COVID-19) compared with their healthy controls. **(B)** The differential expression of the four proteins in different diseases. **(C)** The ROC curves of the four proteins to distinguish patients with different diseases from healthy controls.

(2) about the clinical validity of the four proteins on a wider population scale of CRC patients, but not limited to the RAS wild-type CRC patients or metastatic CRC patients

We appreciate the reviewer for the constructive suggestions, and we agree with the reviewer that the four proteins were indeed screened in the cohort composed of metastatic CRC patients without *RAS* mutations, which reflected the possible limitation of the clinical validity. In the revision, to explore whether the four proteins could also be applied on a wider population scale of CRC patients, we included a CRC plasma cohort composed of patients of CRC (N = 20), of

which 50% (N = 10) patients with CRC had *RAS* mutation and 50% (N = 10) patients with CRC had no tumor metastases, and healthy controls (HC, N = 20). To eliminate the possible limitations that the four proteins could not distinguish CRC patients with no metastases and *RAS* mutation, we performed the differential analysis and ROC analysis based on the plasma proteomic data, respectively in total 20 CRC patients (**Figure RL2A**), CRC patients with *RAS* mutation (**Figure RL2B**), and CRC patients with no metastases (**Figure RL2C**). The differential analysis identified the four proteins (COL12A1, THBS2, S100A8, and S100A9) showed significant up-regulation in the total 20 CRC patients, and the ROC analysis demonstrated the four proteins showed a well distinguish of CRC patients from HCs with an AUC of 0.948 (**Figure RL2A, see also Figure 2H in the revision**). As for the CRC patients with *RAS* mutation, we also found the four proteins showed a consistent up-regulation in the CRC patients with *RAS* mutation; meanwhile, the four proteins achieved a high prediction with an AUC of 0.985 (**Figure RL2B, see also Figure 2H in the revision**). As for the CRC patients with no metastases, the differential analysis also showed a consistently significant up-regulation of the four proteins in the CRC patients with no metastases, with achieving a good performance with an AUC of 0.950 in distinguishing CRC patients with no metastases from HCs (**Figure RL2C, see also Figure 2H in the revision**).

In addition, to further verify whether the four proteins (COL12A1, THBS2, S100A8, and S100A9) showed consistent up-regulation in the tissue samples respectively from CRC patients with no metastases and *RAS* mutation, we also searched the public cohort from the Clinical Proteomic Tumor Analysis Consortium (CPTAC), which composed of a total of 96 tumor and matched normal adjacent tissues (NATs) pairs from CRC patients (Cell., 2019, PMID: 31031003). In the CPTAC CRC cohort, we found the four proteins showed a consistent up-regulation in the tissue samples from CRC patients, which suggested that the four proteins were indeed up-regulated in the tumor samples from CRC patients with no metastases and *RAS* mutation. The receiver operating characteristic (ROC) analysis demonstrated that the four proteins combined prediction could achieve a high prediction with an AUC of 0.917 for the CRC diagnosis, suggesting the stability and universality of the four proteins in CRC diagnosis (**Figure RL2D, see also Figure 2I in the revision**). In the CPTAC CRC cohort, 42% (N = 40) patients with

CRC had *RAS* mutation, and 90% (N = 86) patients with CRC had no tumor metastases. To further demonstrate the predictive efficiency of the four proteins was not limited to the patients with *RAS* wild-type metastatic colorectal cancer, we further stratified 40 CRC patients with *RAS* mutation, of which ROC analysis showed a well distinguish from tumor and NATs with an AUC of 0.912 (**Figure RL2E, see also Figure 2I in the revision**). In addition, we also stratified 86 non-metastatic CRC patients, of which ROC analysis also showed a good prediction with an AUC of 0.931 (**Figure RL2F, see also Figure 2I in the revision**). Therefore, in the revision, we validated the predictive effect of the four proteins on a wider population scale of CRC patients, but not limited to the *RAS* wild-type CRC patients or metastatic CRC patients.

Overall, in this revision, we validated that the combination of four proteins could well distinguish the CRC from HC, but not limited to the *RAS* wild-type CRC patients or metastatic CRC patients, validating the clinical validity of the four proteins on a wider population scale of CRC patients. In addition, we also validated that the four proteins could not distinguish the patients with other cancer types and the non-malignant diseases, demonstrating the relative specificity of the four proteins for the CRC diagnosis. These results demonstrated the robustness of the four proteins in distinguishing CRC patients from healthy controls.

We appreciate the reviewer's valuable suggestions that it is indeed more appropriate to apply the four proteins in the consecutive patients in a colonoscopy cohort for the CRC diagnosis, which inspire us to perform a screening endoscopy in a cohort of consecutive patients in the future. In this study, we toned down the description about the biomarkers for cancer screening and diagnosis in the revision. For example, in the revision, we corrected "diagnostic biomarkers (for CRC diagnosis)" as "the four proteins (in distinguishing CRC patients from HCs)" in the manuscript thoroughly. We also added the discussion about clinical measurement and implication of these potential proteins. Finally, in the revision, we updated the prediction efficacy of the four proteins in distinguishing various diseases from healthy controls, and the clinical validity of the four proteins on a wider population scale of CRC patients, but not limited to the *RAS* wild-type CRC patients or metastatic CRC patients. Please see **Figures 2 and S2; "The plasma proteomic biomarkers to distinguish CRC patients from healthy controls"** in the

“Result” section (line 304–389 in Page 11–13), and the “Discussion” section (line 709–711 and line 714–720 in Page 23) in the revision; the composition of the multi-cancer plasma independent cohort, and the information of the public datasets GSE11223 and GSE207015 were also updated in the “Methods” section (line 884–892 in Page 30, and line 1203–1207 in Page 41). In addition, the raw mass spectrometry proteomics data generated in the revision have also been deposited in the iProX repository under the project ID IPX0005221000 (<https://www.iprox.cn/page/PSV023.html?url=1674226568474Xemk>, password: ykEi).

Figure RL2. The predictive efficacy of the four biomarkers in the plasma and tissue

validation cohorts on a wider CRC population scale. (A, B, and C) Upper: The boxplots showing the up-regulation of the four proteins (COL12A1, THBS2, S100A8, and S100A9) in the total CRC patients (A), the CRC patients with *RAS* mutation (B), the CRC patients with no metastases (C) in the Fudan plasma validation cohort on a wider CRC population scale (two-sided Wilcoxon rank-sum test, adj *P*-value < 0.05). Boxplots show median (central line), upper and lower quartiles (box limits), 1.5×interquartile range (whiskers). Bottom: The ROC curves showing the predictive efficacy of the four proteins to distinguish CRC patients from HCs (A), the CRC patients with *RAS* mutation from HCs (B), the CRC patients with no metastases from HCs (C). (D, E, and F) Upper: The boxplots showing the up-regulation of the four proteins (COL12A1, THBS2, S100A8, and S100A9) in the total CRC patients (D), the CRC patients with *RAS* mutation (E), the CRC patients with no metastases (F) in the CPTAC tissue validation cohort on a wider CRC population scale (two-sided Wilcoxon rank-sum test, adj *P*-value < 0.05). Boxplots show median (central line), upper and lower quartiles (box limits), 1.5×interquartile range (whiskers). Bottom: The ROC curves showing the predictive efficacy of the four proteins to distinguish the total CRC patients in the CPTAC cohort from HCs (D), the CRC patients with *RAS* mutation from HCs (E), the CRC patients with no metastases from HCs (F).

Q2. Proposed (RRAS2, MMP8, FBLN1, RPTOR, and 22 IMPDH2) for the initial response prediction & suggested potential biological mechanisms and biomarkers for the therapy response

Plasma analyses of 89 therapy-naive CRC patients with initial therapy response was evaluated based on the RECIST 1.1 criteria, classifying patients into sensitive (S) and non-sensitive (NS) groups.

Unsupervised consensus clustering analysis was performed on 1,500 of the most variable proteins, resulting in the identification of three proteomic subtypes: G-I, G-II, and G-III.

Key findings:

- Patients in the G-III proteomic subtype tended to benefit from cetuximab therapy, with**

an increase in CD8+ Tem cells and activation of T cell receptor signaling.

· Patients in the G-I proteomic subtype were unlikely to benefit from cetuximab therapy and exhibited activation of the RRAS and RRAS2-mediated ECM pathway.

· Key proteins involved in these pathways were identified and combined to create a predictive model for initial response to cetuximab therapy, which showed robustness and stability across validation cohorts.

· CD8+ Tem cells were identified as potential markers for cetuximab sensitivity, validated across different data types and cohorts.

Comment:

R2 had asked (Q10) about other clinical parameters that may associate with response to anti-EGFR therapy, and whether these were accounted for. This is inadequately answered, in particular, given the association with Right and Left RAS-wildtype CRC with anti-EGFR therapy from phase 3 trials, this must be addressed both as a table showing how proteomic subtypes relate to left and right cancers and whether the predictive abilities remain after adjusting for left/right status.

The findings in relation to pathways related to response/non-response are by nature speculative or observational. The authors should discuss this in relation to current known mechanisms of response/non-response and also discuss why they think it did not predict longitudinal response, just initial response.

Response to Q2:

We appreciate the reviewer for the summary of key findings about the proteomic subtypes and potential biological mechanisms and biomarkers for the initial therapy response. According to the reviewer's suggestions, we divided the response into three parts to answer: (1) about the association of clinical parameters with proteomic subtypes; (2) about the discussion of the potential response mechanisms; (3) about the discussion of the biomarkers for initial response prediction and longitudinal response prediction.

(1) about the association of clinical parameters with proteomic subtypes

Firstly, as the reviewer mentioned, clinical parameters associated with response to anti-EGFR therapy should be accounted for the analysis, such as the primary tumor locations (right-sided and left-sided) and *RAS* mutation status. As clinical trials and researches reported, patients with *RAS* mutation or right-sided metastatic CRC could not benefit from cetuximab therapy, and only the patients with both *RAS* wild-type and left-sided metastatic CRC were candidates for cetuximab therapy (J Clin Oncol., 2008, PMID: 18202412; Ann Oncol., 2015, PMID: 25115304; JAMA Surg., 2018, PMID: 28975237; J Natl Compr Canc Netw., 2019, PMID: 31117039; J Natl Compr Canc Netw., 2021, PMID: 33724754). Therefore, according to the National Comprehensive Cancer Network® (NCCN®) stated the NCCN Clinical Practice Guidelines in Oncology (NCCN Guidelines®) for available at NCCN.org, cetuximab was recommended for the patients with left-sided *RAS* wild-type metastatic colorectal cancer. The same strategy applied for the CRC patients with same condition was also reported in the previous publications (Sci Transl Med., 2015, PMID: 25632036; J Clin Oncol., 2023, PMID: 36252154; Nat Commun., 2023, PMID: 37666855). Therefore, all the CRC patients receiving cetuximab therapy included in this study were left-sided *RAS* wild-type metastatic colorectal cancer. The *RAS* mutation and right-sided metastatic CRC was the exclusion criteria. We apologize for not clarifying the information about the patients' enrollment. In the revision, we added this exact inclusion criteria in the "**Clinical Sample Acquisition**" in the "**Methods**" section.

Secondly, as for the association of proteomic subtypes with clinical parameters, in the last revision, we performed Fisher's exact test on the categorical variables (including age, gender, degree of tumor differentiation, Eastern Cooperative Oncology Group (ECOG) performance-status score, and therapy response) and one way-ANOVA analysis on the continuous variables (including biochemical indicators: lactate dehydrogenase (LDH) level, white blood cell (WBC) count, lymphocyte number (LYMPHN), hemoglobin (HB), and platelet count) among three subtypes (G-I, G-II, and G-III). As a result, we found only therapy response had an obvious association between proteomic subtypes (Fisher's exact test, $P = 0.014$); as for other clinical parameters, including grade, degree of tumor differentiation, ECOG, and some biochemical indicators, there was no significant association with the proteomic subtypes ($P > 0.05$) (Table 2

in the last revision). These results demonstrated that the proteomic subtype could reflect the strong association with therapy response, which was irrespective of other clinical parameters.

Therefore, in this study, the primary tumor locations and RAS mutation status were strictly followed the NCCN Guidelines (only patients with left-sided *RAS* wild-type metastatic colorectal cancer were treated with cetuximab therapy), clarifying the inclusion criteria of the patients with left-sided *RAS* wild-type metastatic colorectal cancer in our cohort; in addition, other clinical parameters were also verified by statistical analysis, showing no association with the proteomic subtypes, thus supporting significant association of proteomic subtypes with therapy response. In the revision, we added this exact inclusion criteria in the “**Clinical Sample Acquisition**” in the “**Methods**” section (line 871–873 in Page 30).

(2) about the discussion of the potential response mechanisms

As the reviewer’s summarized, in this study, we identified three proteomic subtypes featured with distinct molecular features and therapy response, of which patients in the G-III proteomic subtype tended to benefit from cetuximab therapy, with an increase in CD8+ Tem cells and activation of T cell receptor signaling; while patients in the G-I proteomic subtype were unlikely to benefit from cetuximab therapy and exhibited activation of the RRAS and RRAS2-mediated ECM pathway.

As reported, tumor immune cell infiltration was associated with the cetuximab sensitivity, of which CD8+ T cells, one of the primary tumor-infiltrating immune cells that delivered antitumor responses, could improve the efficacy of cetuximab combined therapy in CRC (Cancer Lett., 2023, PMID: 37354983). In this study, we identified elevation of CD8+ Tem and activation of T cell receptor signaling in the cetuximab sensitive group, and further validated the representative signatures (CD44 and GZMK) of the CD8+ Tem through immunohistochemistry (IHC) in the tissue samples, showing agreement with the reported mechanism; in addition, we also identified two proteins RPTOR and IMPDH2 positively associated with both expression of CD8+ Tem and better prognosis. We proposed that the potential function of RPTOR and IMPDH2 in regulation

of tumor immune cell infiltration, and their association with the cetuximab sensitivity is deserved to be further explored in the future.

As for the mechanism related to cetuximab resistance, known mechanisms of resistance to cetuximab therapy included (i) upstream mutations in the extracellular domain of EGFR that directly confer resistance to antibody blockade (Nat Med., 2012, PMID: 22270724), (ii) downstream pathways activated by EGFR, such as RAS-RAF-MAPK-ERK, PI3K-PTEN-AKT, and JAK/STAT pathways mainly through mutations of *KRAS*, *NRAS*, *BRAF* and *MAP2K1* (J Clin Oncol., 2016, PMID: 26438111; J Clin Oncol., 2023, PMID: 37315390; Br J Cancer., 2009, PMID: 19603018; Nature., 2015, PMID: 26416732), and amplifications of *MET* (Cancer Discov., 2013, PMID: 23729478), and *ERBB2* (Sci Transl Med., 2011, PMID: 21900593)). In this study, we identified MAPK signaling activation mediated by RRAS and RRAS2 associated with cetuximab resistance. In addition, besides MAPK signaling, we identified the downstream ECM pathway showed a significant association with cetuximab resistance. We proposed that the potential function of RRAS and RRAS2 in regulation of ECM pathway, and the potential drugs to improve the cetuximab sensitivity were worth being developed in the future.

Overall, our findings uncovered high level of CD8+ Tem and T cell receptor signaling associated with cetuximab sensitivity, while RRAS and RRAS2 in regulation of MAPK and ECM pathway associated with cetuximab resistance, which provided a rich resource for exploring the cetuximab sensitivity and resistance at the proteome level and contributed to develop the potential drug to improve the cetuximab sensitivity. In the revision, we extended the discussion of potential response mechanisms in the “**Discussion**” section (**line 730–731 and line 737–753 in Page 24–25**).

(3) about the discussion of the biomarkers for initial response prediction and longitudinal response prediction

In this study, we combined the key proteins involved in the featured pathways associated with therapy response, including FBLN1, MMP8, and ITGA5 (associated with cetuximab resistance), as well as RPTOR and IMPDH2 (associated with cetuximab sensitivity), and constructed a

predictive model for the initial response to the first cetuximab therapy, which also showed robustness and stability across validation cohorts. In the last revision, according to the reviewer's suggestion, to explore the predictive efficacy of the biomarkers for initial response in the longitudinal treatment, we applied the predictive model for the initial response in the longitudinal cohort covering multi-course treatment. As a result, we found the predictive model composed of the key proteins (RRAS2, MMP8, FBLN1, RPTOR, and IMPDH2) associated with initial response, didn't achieve a good distinguish between sensitive and non-sensitive patients during the longitudinal cetuximab treatment (Figures S4A and S4B in the last revision). Therefore, our finding demonstrated that the biomarkers and predictive model for the initial response to the first cetuximab therapy could not well predict the longitudinal response in the multi-course cetuximab therapy, showing the heterogeneity of different biomarkers for the initial response prediction and longitudinal response prediction respectively in the first course and multi-course cetuximab treatment. The similar phenomenon was also observed in the reported researches. For example, as Tang et al. reported in the research related to serum immune proteomics in predicting response to preoperative chemotherapy of gastric cancer, the pre-treatment serum biomarker level should have greater clinical significance than the post-treatment samples (Cell Rep Med., 2023, PMID: 36724786). One potential reason for the attenuation of the prediction efficiency of biomarkers for initial response prediction in longitudinal response prediction was that, the biomarkers for the initial response prediction were identified and differentially expressed between the sensitive and non-sensitive groups in the therapy-naïve (pre-treatment) samples, but this difference was not significant between the sensitive and non-sensitive groups in the samples after cetuximab treatment (post-treatment). Another possibility was that the difference of the proteins or pathways related to therapy response in the pre-treatment samples possibly changed during the treatment, for instance, some sensitive features could be attenuated while some resistant features could be reserved or even enhanced during the multi-course treatment. In the revision, we added the discussion of biomarkers for the initial and multi-course cetuximab therapy in the **"Discussion"** section (line 764–780 in Page 26).

Overall, in the revision, we added this exact inclusion criteria in the “**Clinical Sample Acquisition**” in the “**Methods**” section (line 871–873 in Page 30), and added the discussion of potential response mechanisms and biomarkers for the initial and multi-course cetuximab therapy in the “**Discussion**” section (line 730–731 and line 737–753 in Page 24–25; line line 764–780 in Page 26).

Q3. Longitudinal dynamics of biomarkers in relation to therapy response of CRC patients with cetuximab

Small cohort:

22 CRC patients, 105 plasma samples, covering seven-course cetuximab treatment.

subset of signatures (including IDH3G, MDN1, 912 KLC4, MYL9, SBF1, and HTRA3) markers suggested

Update: 31 CRC patients, 77 plasma samples, covering two course cetuximab treatment

- **Initial diagnostic and predictive models did not effectively distinguish between sensitive and non-sensitive patients during cetuximab treatment.**
- **Protein dynamic changes were observed in patients' responses to cetuximab treatment during multiple courses.**
- **Sensitive and non-sensitive biomarkers of cetuximab therapy were identified based on protein level fluctuations during treatment.**
- **Pathway enrichment analysis identified pathways associated with these differentially expressed proteins (DEPs), highlighting differences in glycolysis, signaling by Rho GTPases, ECM proteoglycans.**

Comment: Details of the protein level dynamic fluctuations need to be provided.

Given the small number of patients, a supplementary figure showing how the levels of these proteins changed over time in relation to resist response over time in each of these patients will allow the reader/reviewer to appreciate the robustness and dynamic range of this set of protein markers.

Since this marker is being proposed to provide early readouts of response/non-response,

acquired resistance amongst patients with initial response, this information will help clinical readers to interpret the results.

Response to Q3:

We appreciate the reviewer for the constructive suggestions about the dynamic fluctuations of the biomarkers for the longitudinal response prediction to improve the interpretation of our findings. In this study, we defined two panels of proteins as sensitive biomarkers (N = 29) and non-sensitive biomarkers (N = 10) of cetuximab therapy in CRC patients, respectively; K-means analysis demonstrated sensitive biomarkers presented a gradual downward trend with the increase of sampling time during cetuximab treatment, and non-sensitive biomarkers showed a gradual upward trend with the increasing courses during cetuximab treatment. Furthermore, we identified a subset of signatures that accurately discriminates SSG and SNSG (named as S-sig and NS-sig). The S-sig include MDN1, KLC4, and IDH3G, while the NS-sig include SBF1, HTRA3, and MYL9. According to the reviewer's comments, to improve the interpretation of the dynamic change of the signatures, we provided a supplementary figure showing how the levels of these proteins (including MDN1, KLC4, IDH3G, SBF1, HTRA3, and MYL9) changed over sampling time in relation to therapy response during therapy in the revision. As shown in the **Figure RL2**, the S-sig, including MDN1, KLC4, and IDH3G, exhibited higher expression in the sensitive (S) points compared with the non-sensitive (NS) points over the sampling time during therapy, specifically presented a gradual downward trend with the increase of sampling time in relation to response; while the NS-sig, including SBF1, HTRA3, and MYL9, exhibited higher expression in the NS points compared with the S points over the sampling time during therapy, specifically presented a gradual upward trend with the increase of sampling time in relation to response. In the revision, we added the dynamic change of these signatures in the **"Result"** section of the revised manuscript (**line 643–651 in Page 22**), and updated the supplementary figure in the **Figure S4G**.

Figure RL2. The dynamic fluctuation of the biomarkers for the longitudinal response prediction over sampling time in relation to therapy response during therapy. The x axis represents the sampling time: “1” means the first sampling after receiving one course treatment; “2” means that the second sampling after receiving two course treatments; “3” is defined as the third sampling after receiving three course treatments; “4” is defined as the fourth sampling after receiving four course treatments; “5” is defined as the fifth sampling after receiving five course treatments; “6” is defined as the sixth sampling after receiving six course treatments; “7” is defined as the seventh sampling after receiving seven course treatments. The y axis represents the protein intensity (\log_{10} transformed).

REVIEWERS' COMMENTS

Reviewer #4 (Remarks to the Author):

I think, in general the authors have addressed most of the comments and I commend and thank the authors for addressing my 3 categories of comments

Minor comments

(1) diagnostic biomarkers They now only have (N = 10) patients with CRC with no tumor metastases, and limited healthy controls (HC, N = 20). Thus, the authors have appropriately toned down the description about the biomarkers for cancer screening and diagnosis in the revision. I am satisfied with the use of the four proteins (in distinguishing CRC patients from HCs)"

(2) Regarding initial markers of response/resistance to anti-EGFR, the findings of high level of CD8+ Tem and T cell receptor signaling associated with cetuximab sensitivity, while RRAS and RRAS2 in regulation of MAPK and ECM pathway associated with cetuximab resistance. A sentence is needed in the discussion to assist the reader to understand that these findings may suggest that non-tumor, immune related events could contribute towards responses/resistance to anti-EGFR therapy to remind the readers of the difference of their approach (systemic circulating markers from patient blood) versus experimental systems or tumor-only profiling where it is direct anti-tumor effect of drug without interaction with potential effector cells.

(3) Regarding the question of the two panels of proteins as sensitive biomarkers (N = 29) and non-sensitive biomarkers (N = 10) of cetuximab therapy especially the subset of proteomic markers S-sig include MDN1, KLC4, and IDH3G, while the NS-sig include SBF1, HTRA3, and MYL9 that relate to dynamic markers of acquired resistance to therapy. The summary figure in RL2 is not enough, since eventually, we will need to monitor these markers and their dynamics in individual patients (just as average CEA across a cohort is meaningless, it is the intra-patient variability that is helpful)

Presumably, since there are now 6 markers and 31 patients (77 samples) or 22 patients (105 samples)

It would be good to show a supplementary chart per patient to understand the intra-patient variability of these protein markers across time/treatment for the patient, with an arrow to mark out when resistance emerged in that patient.

If indeed these are helpful markers then for each time point that each patient begins to develop acquired resistance, the sensitive markers should drop and the resistance markers should fall.

For how many patients, this occurs and the consistency of the pattern will thus be apparent to the reader in interpreting these findings and the usefulness of the marker.

RESPONSE TO REVIEWERS' COMMENTS

Reviewer #4 (Remarks to the Author):

I think, in general the authors have addressed most of the comments and I commend and thank the authors for addressing my 3 categories of comments

Response:

Thanks for the reviewer's satisfaction with the last round revision. We deeply thank the reviewer for giving us the chance for publication in Nature Communications.

Minor comments

Q1. Diagnostic biomarkers: They now only have (N = 10) patients with CRC with no tumor metastases, and limited healthy controls (HC, N = 20). Thus, the authors have appropriately toned down the description about the biomarkers for cancer screening and diagnosis in the revision. I am satisfied with the use of the four proteins (in distinguishing CRC patients from HCs)"

Response to Q1:

We appreciate for the reviewer's satisfaction with the last round revision. We deeply thank the reviewer for giving us the chance for publication in Nature Communications.

Q2. Regarding initial markers of response/resistance to anti-EGFR, the findings of high level of CD8+ Tem and T cell receptor signaling associated with cetuximab sensitivity, while RRAS and RRAS2 in regulation of MAPK and ECM pathway associated with cetuximab resistance. A sentence is needed in the discussion to assist the reader to understand that these findings may suggest that non-tumor, immune related events could contribute towards responses/resistance to anti-EGFR therapy to remind the readers of the difference of their approach (systemic circulating markers from patient blood) versus experimental systems or tumor-only profiling

where it is direct anti-tumor effect of drug without interaction with potential effector cells.

Response to Q2:

We appreciate for the reviewer's constructive suggestions, which deepen the interpretation of our findings that CD8+ Tem and T cell receptor signaling associated with cetuximab sensitivity, while RRAS and RRAS2 in regulation of MAPK and ECM pathway associated with cetuximab resistance. As the reviewer's summarized, our findings suggested that non-tumor, immune related events could contribute towards responses to anti-EGFR therapy, which would remind us of the difference of the approach (systemic circulating markers from patient blood) versus experimental systems or tumor-only profiling where it is direct anti-tumor effect of drug without interaction with potential effector cells. Finally, we appreciate the reviewer's valuable reminder, and we have added the understanding of our findings in the third paragraph of "**Discussion**" section in the revision as follows (highlighted in red text) (**line 762–766 in Page 26**).

"Besides tumor diagnosis, the resistant or sensitive mechanisms for cetuximab therapy remain a major problem to be unsolved. We found these diagnostic biomarkers could not be regarded as indicators for the therapy response prediction. In this study, to search for the biomarkers for predicting the therapy response, we determined the molecular features associated with therapy response by consensus clustering analysis and identified the potential predictive biomarkers by logistic regression strategy. We identified three proteomic subtypes featured with different bioprocesses and associated with various therapeutic responses. As main findings of this study, among the three proteomic subtypes, G-III subtype was featured with autophagy mediated aggregation of CD8+ Tem, and prone to sensitive to cetuximab therapy. Consistently, as reported, tumor immune cell infiltration was associated with the cetuximab sensitivity⁶⁸. Furthermore, the immune microenvironment estimated by xCell analysis on the plasma and tissue proteomic data revealed the association of CD8+ Tem with cetuximab sensitivity, which was further validated by IHC measurement of the representative signatures of CD8+ Tem in the tissue samples. In addition, we explored the clinical implication of CD8+ Tem, and proposed that CD8+ Tem could be regarded as the potential marker for the cetuximab sensitivity, which was validated by the single-cell transcriptome data from CRC patients (GSE108989). In this

study, we identified two proteins RPTOR and IMPDH2 positively associated with both expression of CD8+ Tem and better prognosis. We proposed that the potential function of RPTOR and IMPDH2 in regulation of tumor immune cell infiltration, and their association with the cetuximab sensitivity is deserved to be further explored in the future. While G-I subtype was featured with RRAS and RRAS2-mediated an axis of resistant mechanism including MAPK, RHO, and ECM pathways, and prone to be resistant to cetuximab therapy. As reported, known mechanisms of resistance to cetuximab therapy mainly included (i) upstream mutations in the extracellular domain of EGFR that directly confer resistance to antibody blockade⁶⁹, (ii) downstream pathways activated by EGFR, such as RAS-RAF-MAPK-ERK, PI3K-PTEN-AKT, and JAK/STAT pathways mainly through mutations of *KRAS*, *NRAS*, *BRAF*, and *MAP2K1*^{70, 71, 72, 73}, and amplifications of *MET* and *ERBB2*^{74, 75}. Importantly, in this study, we identified MAPK signaling activation mediated by RRAS and RRAS2 associated with cetuximab resistance. In addition, besides MAPK signaling, we identified the downstream ECM pathway showed a significant association with cetuximab resistance. We proposed that the potential function of RRAS and RRAS2 in regulation of ECM pathway, and the potential drugs to improve the cetuximab sensitivity were worth being developed in the future. Interestingly, the key proteins (RRAS2, MMP8, FBLN1, RPTOR, and IMPDH2) involved in these represented resistant/sensitive pathways and associated with clinical prognosis, had a good combined prediction of the initial response to first cetuximab treatment, further validating the association with cetuximab therapeutic response. These findings provided a solid reference for further investigating the sensitive or resistant mechanism, **which also suggested that the non-tumor, immune related events could contribute towards responses to anti-EGFR therapy. This revealed the difference of the approach (systemic circulating markers from patient blood) versus experimental systems or tumor-only profiling where it is direct anti-tumor effect of drug without interaction with potential effector cells.** In addition, although we observed the consistent alteration of the proteins and pathways uncovered both in plasma samples and tumor tissues of CRC patients, we couldn't completely exclude the contribution of other tissues or cells in plasma. It is deserved to explore the components secreted by circulating immune cells associated with the therapy response in the future."

Q3. Regarding the question of the two panels of proteins as sensitive biomarkers (N = 29) and non-sensitive biomarkers (N = 10) of cetuximab therapy especially the subset of proteomic markers S-sig include MDN1, KLC4, and IDH3G, while the NS-sig include SBF1, HTRA3, and MYL9 that relate to dynamic markers of acquired resistance to therapy.

The summary figure in RL2 is not enough, since eventually, we will need to monitor these markers and their dynamics in individual patients (just as average CEA across a cohort is meaningless, it is the intra-patient variability that is helpful)

Presumably, since there are now 6 markers and 31 patients (77 samples) or 22 patients (105 samples)

It would be good to show a supplementary chart per patient to understand the intra-patient variability of these protein markers across time/treatment for the patient, with an arrow to mark out when resistance emerged in that patient.

If indeed these are helpful markers then for each time point that each patient begins to develop acquired resistance, the sensitive markers should drop and the resistance markers should fall.

For how many patients, this occurs and the consistency of the pattern will thus be apparent to the reader in interpreting these findings and the usefulness of the marker.

Response to Q3:

We appreciate the reviewer for the more valuable suggestions about the dynamic fluctuations of the biomarkers for the longitudinal response prediction in this round revision. We apologize for not presenting the dynamic change of these biomarkers in individual patient. We agree with the reviewer that the average of the expression of these biomarkers across a cohort is not enough, and the intra-patient variability is helpful. In this study, we defined two panels of proteins as sensitive biomarkers (N = 29) and non-sensitive biomarkers (N = 10) of cetuximab therapy in CRC patients, respectively; K-means analysis demonstrated sensitive biomarkers presented a gradual downward trend with the increase of sampling time during cetuximab treatment, and non-sensitive biomarkers showed a gradual upward trend with the increasing

courses during cetuximab treatment. Furthermore, we identified a subset of signatures that accurately discriminates SSG and SNSG (named as S-sig and NS-sig). The S-sig include MDN1, KLC4, and IDH3G, while the NS-sig include SBF1, HTRA3, and MYL9.

According to the reviewer's comments, to improve the interpretation of the dynamic change of the signatures, we provided a supplementary figure showing how the levels of these proteins (S-sig: MDN1, KLC4, and IDH3G; NS-sig: SBF1, HTRA3, and MYL9) of individual patient changed over sampling time in relation to therapy response during therapy in the revision. Here, we analyzed the proteome data of the S-sig and NS-sig proteins in the 22 CRC patients covering multiple courses of cetuximab treatments (up to seven courses), which could clearly and systematically present the dynamics of the S-sig and NS-sig proteins associated with response over the multiple courses of cetuximab treatment in individual patient. As shown in the **Figure RL1**, across the 22 patients with multiple-course cetuximab treatment, we observed the S-sig proteins (including MDN1, KLC4, and IDH3G) exhibited a significant drop when resistance emerged, and maintained a gradual downward trend in the subsequent courses of cetuximab treatment; while the NS-sig proteins (including SBF1, HTRA3, and MYL9) exhibited a significant rise when resistance emerged, and maintained a gradual upward trend in the subsequent courses of cetuximab treatment. Among the 22 patients, we observed a consistent expression pattern of these signature proteins across the multiple courses of cetuximab treatment, indicating the effectiveness of these biomarkers for predicting cetuximab response of individual patient. In addition, we also observed the variability of these proteins with longitudinal response among intra-patients, suggesting the potential heterogeneity of the biomarkers for response prediction among different individual patients. We thank the reviewer help us further interpretate the dynamic change of these biomarkers in individual patient. Finally, in the revision, we added the dynamic change of these signatures in the "**Results**" section (**line 646–656 in Page 22**) and "**Discussion**" section (**line 795–800 in Page 27**) of the revised manuscript, and updated the supplementary figure in the **Figure S5**.

Figure RL1. The dynamic fluctuation of the biomarkers for the longitudinal response prediction across multiple courses during cetuximab treatment. S-sig: MDN1, KLC4, and IDH3G; NS-sig: SBF1, HTRA3, and MYL9. The arrow indicates the resistance emerged.

The x axis represents the sampling time: "1" means the first sampling after receiving one course treatment; "2" means that the second sampling after receiving two course treatments; "3" is defined as the third sampling after receiving three course treatments; "4" is defined as the fourth sampling after receiving four course treatments; "5" is defined as the fifth sampling after receiving five course treatments; "6" is defined as the sixth sampling after receiving six course treatments; "7" is defined as the seventh sampling after receiving seven course treatments. The y axis represents the protein intensity (\log_{10} transformed).